

# Evaluating uncertainty and predictive performance of probabilistic models devised for grade estimation in a porphyry copper deposit

Raymond Leung, Alexander Lowe, and Arman Melkumyan

Rio Tinto Centre, Faculty of Engineering, The University of Sydney NSW 2006, Australia

**Correspondence:** Raymond Leung (raymond.leung@sydney.edu.au)

**Abstract.** Probabilistic models are used extensively in geoscience to describe random processes as they allow prediction uncertainties to be quantified in a principled way. These probabilistic predictions are valued in a variety of contexts ranging from geological and geotechnical investigations to understanding subsurface hydrostratigraphic properties and mineral distribution. However, there are no established protocols for evaluating the uncertainty and predictive performance of univariate

probabilistic models, and few examples for researchers and practitioners to lean on. This paper aims to bridge this gap by developing a systematic approach that targets three objectives. First, geostatistics are used to check if the probabilistic predictions are reasonable given validation measurements. Second, image-based views of the statistics help facilitate large-scale simultaneous comparisons for multiple models across space and time, spanning multiple regions and inference periods. Third, variogram ratios are used to objectively measure the spatial fidelity of models. In this study, the model candidates include or-

dinary kriging and Gaussian Process, with and without sequential or correlated random field simulation. FLAGSHIP statistics are proposed to examine the fidelity, likelihood, accuracy, goodness, synchronicity, histogram, interval tightness and precision of the model predictive distributions. These statistics are standardised, interpretable and amenable to significance testing. The proposed methods are demonstrated using extensive data from a real copper mine in a grade estimation task, and accompanied by an open-source implementation. The experiments are designed to emphasise data diversity and convey insights, such

as the increased difficulty of future-bench prediction (extrapolation) relative to in-situ regression (interpolation). This work presents a holistic approach that enables modellers to evaluate the merits of competing models and employ models with greater confidence by assessing the robustness and validity of probabilistic predictions under challenging conditions.

## 1 Introduction

Probabilistic models are useful for describing a wide range of stochastic processes and natural phenomena in the geosciences.

For instance, Monte Carlo techniques have been used in the field of landslide hazard assessment to account for estimation uncertainties and spatial variability of geological, geotechnical, geomorphological and seismological parameters by treating the target quantities as statistical distributions (Refice and Capolongo, 2002). In the field of subsurface and hydrostratigraphic modelling, borehole and geophysical data are often combined to improve lithological and structural understanding in a study area (Tacher et al., 2006). One difficulty is maintaining consistency between different pieces of information while taking

into account various spatial and geological factors when assigning uncertainties to such interpretation. Facing this challenge,



probabilistic models have been used to quantify and communicate such uncertainties in a more principled way. In Madsen et al. (2022), a realisation of the subsurface is created from a 3D geological model to estimate the uncertainty. It requires geostatistical simulation of each hydrostratigraphic layer using boundary points specified by geologists. Another area of research focuses on dynamic stochastic models and bayesian inference in very high dimensional space. For instance, Bacci et al. (2023) considered

different approaches for sampling distribution, while others applied bayesian fusion to multiple data sources to minimise uncertainty (Seillé et al., 2023). Although these ideas are innovative and compelling, they lie outside the scope and general concerns of the present study, which focuses instead on model evaluation. Nonetheless, this brief survey shows there is intense interest in using probabilistic models to describe stochastic processes in the geoscientific research community. Accompanying such rapid development is the need to clarify the performance of models to determine how reasonable they are within the

context of their stated aims. For a discourse on the topic of errors and uncertainties in the geosciences, readers are referred to (Pérez-Díaz et al., 2020). The rest of this introduction will provide a background to approaches that are more closely aligned with this work.

In engineering geology, geostatistical approaches based on kriging (Oliver et al., 2015) are often seen as the methods of choice for spatial interpolation in applications such as soil sampling where the available measurements are sparse. In machine

learning, Gaussian Processes (GP) has been advanced as an alternative kernel-based approach for performing regression over a regionalised variable using Bayesian inference techniques. Unlike the kriging families of models, which require sequential simulations to provide credible uncertainty estimates and overcome excessive smoothing, the GP solution inherently provides covariance and mean estimates in the form of a posterior distribution that maximises the marginal likelihood (Williams and Rasmussen, 2006). Empirically, GP probabilistic models have been found to be at least as effective as, if not superior to ordi-

nary kriging (Christianson et al., 2023). However, there has not been large-scale systematic comparison published in literature to establish their efficacy to the best of our knowledge. As motivation, this work seeks to close this gap. The main contribution of this paper is a multipronged method for evaluating the uncertainty and predictive performance of probabilistic models, specifically ones devised for grade estimation in a porphyry copper deposit. Although this application features challenges that are unique to mining—such as high sampling cost and pit-level causality—at its core, assessing the quality of extrapolation is

of general interest in a multitude of problems. In open-pit mining, there is a strong emphasis on predicting the geochemical properties of an orebody beyond the active mining area where assay data have already been gathered from existing blastholes. Thus, the ability to predict the grade distribution in the bench-below—which has not yet been excavated—or adjacent areas within the current bench is highly valued. This, however, does not trivialise the importance of in-situ regression (grade modelling within the current bench) as it facilitates precision mining and tracking of material movement. Improved knowledge of

the region can improve grade-control and reduce incidents like ore dilution where low-grade waste is excavated and transferred inadvertently to high-grade stockpiles (Leung et al., 2023b). A practical application of probabilistic modelling is depicted in Fig. 1. There is certainly an incentive to refine models of the local geology incrementally by exploiting production data (Leung et al., 2022), extend and predict with a quantifiable degree of confidence the grade distribution in future benches, as this is necessary for robust mine planning. These probabilistic predictions can affect scheduling decisions and optimisation of

mining operations downstream (Seiler et al., 2022; Samavati et al., 2020). In this context, the spatial domain is intrinsically

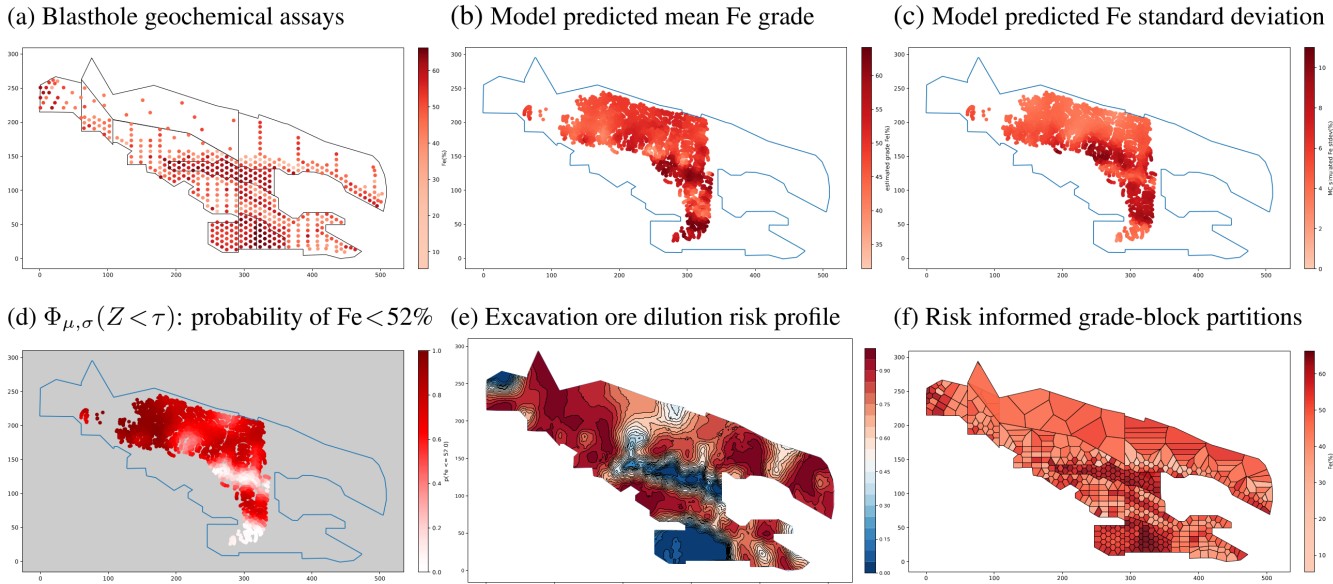

**Figure 1.** A motivating example. For open-pit mining at iron ore deposits, (a) sparse assay measurements are taken from blastholes to facilitate ore grade probabilistic modelling. (b) and (c) show the estimated mean Fe concentration and standard deviation in a local region soon to be excavated. The value of having a probabilistic model is that it provides a reliable and objective description of ore/waste distribution in spite of sampling errors and epistemic uncertainty. This allows operators to assess risks, such as ore dilution in (d), if a volume of low-grade material is dug up at a [red] location and transported to a high-grade destination. (e) A high-fidelity probabilistic model makes informed decision making possible. Its applications include high precision large-scale tracking of material movement, as well as (f) grade-block partitioning and reconfiguration during mine planning and the ability to reroute material to different destinations on demand (Leung et al., 2023b).

three-dimensional, whereas it is common for surficial geochemistry modelling (Chlingaryan et al., 2024) in environmental monitoring to be two-dimensional. The objective of this paper is to assemble a suite of relevant global and local accuracy and uncertainty-based measures and recommend a systematic geostatistical approach for evaluating the performance of univariate probabilistic models.

For deterministic model evaluation, readers might be accustomed to using measures such as the root mean squared error (RMSE). A major deficiency with RMSE is that it treats errors as independent and ignores spatial correlation. When uncertainty is modelled explicitly, model evaluation takes on a different meaning; performance is viewed through a different lens. Associated with each prediction is an implied probability that the true value lies within some interval according to the distribution. This perspective bestows meaning and captures the essence of the uncertainty-based measures. In terms of organisation, Sec. 2 provides concrete definitions of the model candidates considered in this work. This encompasses simple and ordinary kriging (with and without sequential Gaussian simulations) and different forms of Gaussian Process regression (with and without sequential or correlated random field simulations). Section 3 introduces the proposed measures. These measures include classical histogram distances, a spatial fidelity measure derived from variograms, and a reframing of established uncer-





tainty measures based on the notion of 'synchronicity'. Section 4 describes the experiments, dataset and its geological setting. Section 5 presents results and extensive analysis covering a dozen inference periods and approximately ten domains. The experiments are designed to mimic the staged progression of mining operations in an open-pit mine and highlight the opportunity to propagate local geological knowledge, principally acquired from assay data collected from production drilled holes, to project into a future bench.[1] The decision to examine multiple geological domains is to reflect the diverse geochemical characteristics through their grade distributions. Section 6 summarises the main findings from this study and recommends a standard procedure (known as FLAGSHIP) for evaluating uncertainty and predictive performance given a number of probabilistic models.

## 2 Geostatistical modelling

A fundamental viewpoint of probabilistic models is that the target attribute at each point $\boldsymbol{x} \in \mathbb{R}^d$ is described by a probability distribution rather than a single value. Ascribing to the theory of random functions, the observed value is considered as a random realisation of a stochastic process, where the pattern of variation with respect to $\boldsymbol{x}$ can only be described in a statistical sense—typically by the mean and correlation structure in the signal. In this work, the attribute represents the concentration/grade of copper in an ore deposit and the points $\boldsymbol{x}$ correspond to locations in 3D space. The following sections provide a concise overview of Gaussian Processes and kriging based on the description and notations used in (Shekaramiz et al., 2019).

### 2.1 Gaussian Processes

The general problem involves predicting a target attribute (or output) $y_* \in \mathbb{R}$ at some test locations $\boldsymbol{x}_* \in \mathbb{R}^d$ given a training set $D = \{(\boldsymbol{x}_i, y_i)\}_{i=1:n}$ where $y_i$ is known. The asterisk indicates the variable belongs to the test set, where $y_* = f(\boldsymbol{x}_*)$ for some unknown random function $f$. Concatenating the input observations together, the training data may be written in matrix-vector notation as $D = \{\mathbf{X}, \boldsymbol{y}\}$ with $\mathbf{X} = [\boldsymbol{x}_1, \ldots, \boldsymbol{x}_n] \in \mathbb{R}^{d \times n}$ and $\boldsymbol{y} \in \mathbb{R}^n$. In Gaussian Process (GP) regression, a Gaussian prior is placed over the function $f$. Using the observations in $D$, Bayesian inference infers $p(f \,|\, D)$ and converts this into a posterior via $p(\boldsymbol{y}_* \,|\, \mathbf{X}_*, \{\mathbf{X}, \boldsymbol{y}\}) = \int p(\boldsymbol{y}_* \,|\, f, \mathbf{X}_*) p(f|D) df$. The GP consists of a collection of random variables, $f(\boldsymbol{x}) \sim \mathrm{GP}(m(\boldsymbol{x}), \kappa(\boldsymbol{x}, \boldsymbol{x}'))$, which are individually and jointly Gaussian distributed. Hence, a GP is completely specified by a mean function $m(\boldsymbol{x}) = \mathbb{E}[f(\boldsymbol{x})]$ and covariance function $\kappa(\boldsymbol{x}, \boldsymbol{x}') = \mathbb{E}[(f(\boldsymbol{x}) - m(\boldsymbol{x}))(f(\boldsymbol{x}') - m(\boldsymbol{x}'))^T]$. GP regression defines a joint distribution, $p(\boldsymbol{f} \,|\, \mathbf{X}) = \mathcal{N}(\boldsymbol{f} \,|\, \boldsymbol{\mu}, \mathbf{K})$, over a set of $n$ data points, where $\mathbf{K}_{ij} = \kappa(\boldsymbol{x}_i, \boldsymbol{x}_j)$ and $\boldsymbol{\mu} = [m(\boldsymbol{x}_1), \ldots, m(\boldsymbol{x}_n)]^T$. Here, $\kappa$ represents a positive semi-definite kernel (covariance function) that models spatial correlation in the data. It conforms to the general expectation that $f(\boldsymbol{x}_i)$ and $f(\boldsymbol{x}_j)$ will be more similar if $\boldsymbol{x}_i$ and $\boldsymbol{x}_j$ are close together.

---

[1] As a term of reference, the scope of a *future bench* includes any undeveloped region in the vicinity of a densely-drilled and sampled region in an open-pit mine, whether it is physically adjacent to or below the current bench.




For noisy observations, the training data is assumed to satisfy $y_i = f(\boldsymbol{x}_i) + \epsilon$, where $\epsilon \sim \mathcal{N}(0, \sigma_y^2)$.[2] The joint density governing the observed and test data is given by

$$\begin{bmatrix} \boldsymbol{y} \\ \boldsymbol{f}_* \end{bmatrix} \sim \mathcal{N}\left(0, \begin{bmatrix} \mathbf{K}_y & \mathbf{K}_* \\ \mathbf{K}_*^T & \mathbf{K}_{**} \end{bmatrix}\right) \tag{1}$$

and the posterior predictive density is given by

$$p(\boldsymbol{f}_* | \mathbf{X}_*, \mathbf{X}, \boldsymbol{f}) = \mathcal{N}(\boldsymbol{f}_* | \boldsymbol{\mu}_*, \boldsymbol{\Sigma}_*) \tag{2}$$

The equations that describe the prediction outcomes, viz., the posterior mean, $\boldsymbol{\mu}_*$, and covariance, $\boldsymbol{\Sigma}_*$, are given by

$$\boldsymbol{\mu}_* = \mu(\mathbf{X}_*) + \mathbf{K}_*^T \mathbf{K}_y^{-1}(\boldsymbol{f} - \mu(\mathbf{X})) \tag{3}$$

$$\boldsymbol{\Sigma}_* = \mathbf{K}_{**} - \mathbf{K}_*^T \mathbf{K}_y^{-1} \mathbf{K}_* \tag{4}$$

$$\mathbf{K}_y = \mathrm{cov}(\boldsymbol{y} | \mathbf{X}) = \mathbf{K} + \sigma_y^2 \mathbf{I} \tag{5}$$

The training data covariance matrix $\mathbf{K} \in \mathbb{R}^{n \times n}$ is independent of test locations, whereas $\mathbf{K}_* \in \mathbb{R}^{m \times n}, \mathbf{K}_*[i,j] = \kappa(\boldsymbol{x}_i, \boldsymbol{x}_{*j})$ and $\mathbf{K}_{**} \in \mathbb{R}^{m \times m}, \mathbf{K}_{**}[i,j] = \kappa(\boldsymbol{x}_{*i}, \boldsymbol{x}_{*j})$ both depend on the test/query locations.

Returning to the covariance function $\kappa$, common choices include the exponential and squared exponential kernels. For this work, a Matérn 3/2 kernel (Melkumyan and Ramos, 2011), $\kappa_{\text{Matérn 3/2}}(\theta)$, is used drawing on past experience in orebody grade modelling. GP training describes the process of fitting the kernel hyperparameters, $\theta = [a^2, l_x, l_y, l_z, \sigma_v^2]$, to the training data. This involves maximising the log marginal likelihood (LML) with respect to $\theta$:

$$\log p(\boldsymbol{y} | \mathbf{X}, \theta) = -\frac{1}{2} \boldsymbol{y}^T \left[ \mathbf{K} + \sigma_y^2 \mathbf{I} \right]^{-1} \boldsymbol{y} - \frac{1}{2} \log \left| \mathbf{K} + \sigma_y^2 \mathbf{I} \right| - \frac{N}{2} \log 2\pi \tag{6}$$

where $|\cdot|$ denotes the determinant. The marginal likelihood in Eq. 6 contains three terms that represent (from left to right) the data fit, complexity penalty (to include the Occam's razor principle) and normalisation constant. The first two terms in Eq. 6 depend on the values of the hyperparameters. As the marginal likelihood is a non-convex function of the hyperparameters, only local maxima can be obtained—usually via gradient descent from multiple starting points (Melkumyan and Ramos, 2009).

## 2.2 Kriging

The central proposition in kriging is to estimate the value of a continuous attribute $Z$ at any unsampled location $\boldsymbol{x}_*$ using only the available observations $D = \{(\boldsymbol{x}_i, z(\boldsymbol{x}_i))\}_{i=1:n}$ in the studied area. The objective is to obtain $Z^*(\boldsymbol{x}_*)$, an unbiased linear estimator of the random variable $Z(\boldsymbol{x}_*)$, via

$$Z^*(\boldsymbol{x}_*) - \mu(Z(\boldsymbol{x}_*)) = \sum_{i=1}^{\tilde{n}} \lambda_i(\boldsymbol{x}_i)[z(\boldsymbol{x}_i) - \mu(Z(\boldsymbol{x}_i))] \tag{7}$$

---

[2]This observation model naturally excludes multimodal data. This situation is thankfully circumvented by grouping the training data into self-similar clusters or by geological domains, as is done customarily within the field.





where $\lambda_i(\boldsymbol{x}_i)$ denotes the kriging weights assigned to $z(\boldsymbol{x}_i)$, the observed values at location $\boldsymbol{x}_i$ from the training data, and $\mu(Z(\boldsymbol{x}))$ denotes the expected value of the random variable $Z(\boldsymbol{x})$. The number of terms involved in the summation, $\tilde{n}$, can vary from location to location. In general, only the $n(\boldsymbol{x}_*)$ terms closest to $\boldsymbol{x}_*$ are retained. Kriging seeks to minimise the estimation error $\sigma_E^2(\boldsymbol{x}_*) = \text{Var}(Z^*(\boldsymbol{x}_*) - Z(\boldsymbol{x}_*))$ subject to $\mathbb{E}[Z^*(\boldsymbol{x}_*) - Z(\boldsymbol{x}_*)] = 0$. The random variable may be resolved in terms of a residual component, $R(\boldsymbol{x})$, and a trend component, $\mu(Z(\boldsymbol{x}))$, and written as $Z(\boldsymbol{x}) = R(\boldsymbol{x}) + \mu(Z(\boldsymbol{x}))$ based on (7). Kriging models this residual component as a zero-mean stationary Gaussian Process such that $R(\boldsymbol{x}) \sim \text{GP}(0, \text{Cov}(R(\boldsymbol{x}), R(\boldsymbol{x} + h))$.

In this work, two types of kriging known as Simple Kriging (SK) and Ordinary Kriging (OK) are considered. In *simple kriging*, the mean $\mu(Z(\boldsymbol{x}))$ is assumed to be known and constant such that $\mu(Z(\boldsymbol{x})) = m \ \forall \boldsymbol{x}$ throughout the modelled region. *Ordinary kriging* permits variation in the mean which is unknown by limiting the domain of stationarity to local neighbourhoods, $N(\boldsymbol{x})$. As such, $\mu(Z(\boldsymbol{x})) = m(\boldsymbol{x})$ where $m(\boldsymbol{x}') = \text{const} \ \forall \boldsymbol{x}' \in N(\boldsymbol{x})$. In SK, the prediction error variance in (8) is minimised by the kriging weights in (9) under the stationary mean assumption.

$$\sigma_E^2(\boldsymbol{x}_*) = \sum_{i=1}^{n(\boldsymbol{x}_*)} \sum_{j=1}^{n(\boldsymbol{x}_*)} \lambda_i(\boldsymbol{x}_*)\lambda_j(\boldsymbol{x}_*)C_R(\boldsymbol{x}_i - \boldsymbol{x}_j) + C_R(0) - 2\sum_{i=1}^{n(\boldsymbol{x}_*)} \lambda_i(\boldsymbol{x}_*)C_R(\boldsymbol{x}_i - \boldsymbol{x}_*) \tag{8}$$

where $C_R(h) = \mathbb{E}[R(\boldsymbol{x}_*)R(\boldsymbol{x}_* + h)]$

$$\boldsymbol{\lambda}_{\text{SK}}(\boldsymbol{x}_*) = \text{argmin}_{\boldsymbol{\lambda}} \sigma_E^2(\boldsymbol{x}_*) = \mathbf{K}_{\text{SK}}^{-1}\boldsymbol{k}_{\text{SK}} \tag{9}$$

where $\mathbf{K}_{\text{SK}} = \begin{bmatrix} C(\boldsymbol{x}_1 - \boldsymbol{x}_1) & \dots & C(\boldsymbol{x}_1 - \boldsymbol{x}_{n(\boldsymbol{x}_*)}) \\ \vdots & \vdots & \vdots \\ C(\boldsymbol{x}_{n(\boldsymbol{x}_*)} - \boldsymbol{x}_1) & \dots & C(\boldsymbol{x}_{n(\boldsymbol{x}_*)} - \boldsymbol{x}_{n(\boldsymbol{x}_*)}) \end{bmatrix}$

$$\boldsymbol{k}_{\text{SK}} = [C(\boldsymbol{x}_1 - \boldsymbol{x}_*), \dots, C(\boldsymbol{x}_{n(\boldsymbol{x}_*)} - \boldsymbol{x}_*)]^T$$

$$\boldsymbol{\lambda}_{\text{SK}} = [\lambda_1(\boldsymbol{x}_*), \dots, \lambda_{n(\boldsymbol{x}_*)}(\boldsymbol{x}_*)]^T \text{ s.t. } \sum_i \lambda_i(\boldsymbol{x}_*) = 1$$

In the ordinary kriging case, one solves for the augmented system in (10)

$$\begin{bmatrix} \mathbf{K} & -\mathbf{1}_{n(\boldsymbol{x}_*)} \\ \mathbf{1}_{n(\boldsymbol{x}_*)}^T & 0 \end{bmatrix} \begin{bmatrix} \boldsymbol{\lambda}_{\text{OK}} \\ \zeta \end{bmatrix} = \begin{bmatrix} \boldsymbol{k}_{\text{OK}} \\ 1 \end{bmatrix} \tag{10}$$

where $\mathbf{K}_{ij} = C(\boldsymbol{x}_i - \boldsymbol{x}_j), \boldsymbol{k}_{\text{OK},i} = C(\boldsymbol{x}_i - \boldsymbol{x}_*), \boldsymbol{\lambda}_{\text{OK},i} = \lambda_i(\boldsymbol{x}_*)$ and $\zeta$ is a Lagrange multiplier.

This will result in a slightly higher $\sigma_{\text{OK}}^2$ than $\sigma_{\text{SK}}^2$ due to greater uncertainty associated with estimating $m(\boldsymbol{x}')$. The minimum prediction variance are given respectively by (11) and (12).

$$\sigma_{\text{SK}}^2(\boldsymbol{x}_*) = \min \sigma_E^2(\boldsymbol{x}_*) = C(0) - \boldsymbol{\lambda}_{\text{SK}}^T \boldsymbol{k}_{\text{SK}} \tag{11}$$

$$\sigma_{\text{OK}}^2(\boldsymbol{x}_*) = C(0) - \sum_{i=1}^{n(\boldsymbol{x}_*)} \boldsymbol{\lambda}_{\text{OK}}^T \boldsymbol{k}_{\text{OK}} + \zeta \tag{12}$$



These procedures are commonly referred as solving the kriging equations. Notably, the kriging weights $\boldsymbol{\lambda}_{\text{SK}}$ and variance $\sigma^2_{\text{SK}}$ (similarly for $\boldsymbol{\lambda}_{\text{OK}}$ and $\sigma^2_{\text{OK}}$) depend only on the locations and covariance function, $C$, not the observed values $z_i$ in $D$. The training data, $D$, is fitted to experimental variograms (Oliver et al., 2015) to deduce the effective range of spatial dependence in the random process. This in turn informs how quickly the signal correlation attenuates under a variogram model. In this paper, a Matérn covariance function is used to compute $C(\boldsymbol{x}, \boldsymbol{x}') \overset{\triangle}{=} C(\boldsymbol{x} - \boldsymbol{x}')$.

**2.3 Sequential Gaussian simulation**

It is well known that kriging regression causes oversmoothing (Olea and Pawlowsky, 1996) and this phenomenon can be described as a deficit of variance, $\text{Var}[Z(\boldsymbol{x}_*)] - \text{Var}[Z^*(\boldsymbol{x}_*)] = \sigma^2_K(\boldsymbol{x}_*) > 0$. This effect increases as the estimated location $\boldsymbol{x}_*$ gets further away from the known data $\boldsymbol{x} \in D$ (Journel et al., 2000). Stated differently, although kriging correctly estimates the data variance between known points—and this ability carries over to replicating the variability between a test point $\boldsymbol{x}_* \notin D$ and

160 known point $\boldsymbol{x} \in D$— it does not reproduce the spatial variability between a pair of test points $(\boldsymbol{x}_{*i}, \boldsymbol{x}_{*j})_{i \neq j} \notin D$. Sequential Gaussian simulation (SGS) remedies this situation by imparting structure (conditional dependence between known and test points) incrementally and augmenting the set of known points with newly estimated test points. The full procedure is described in (Bai and Tahmasebi, 2022; Asghari et al., 2009). This involves the following steps: 1) apply normal score transformation $g$ to the sampled data $z_i \in D$ s.t. $y_i = g(z_i)$; 2) fit the Gaussian-distributed data to a variogram model; 3) for each simulation $s$ (from

165 1 to $N_S$), use a permutation function $\pi_s$ to generate a random path that traverses all test points $\{\boldsymbol{x}_{*,j}\}_j$ that require estimation (McLennan, 2002); 4) for each test point $\boldsymbol{x}_{*,\pi_s(j)}$ along the designated path, search within a local neighbourhood of $\boldsymbol{x}_{*,\pi_s(j)}$ and find the $\tilde{n}$ closest points among all sampled data and previously simulated points; 5) using the $\tilde{n}$ points, compute the kriging mean and variance estimates, $\hat{\mu}_{*,\pi_s(j)}$ and $\hat{\sigma}^2_{*,\pi_s(j)}$, for point $\boldsymbol{x}_{*,\pi_s(j)}$; 6) randomly sample a value $\tilde{y}^{(s)}_{*,\pi_s(j)}$ from the normal distribution $\mathcal{N}(\hat{\mu}_{*,\pi_s(j)}, \hat{\sigma}_{*,\pi_s(j)})$; 7) append $(\boldsymbol{x}_{*,\pi_s(j)}, \tilde{y}_{*,\pi_s(j)})$ to the set of previously simulated points; 8) continue until all

170 test points have been visited in simulation $s$; 9) back transform the simulated values to the original space s.t. $\tilde{z}^{(s)}_{*,j} = g^{-1}(\tilde{y}^{(s)}_{*,j})$; 10) when all $N_S$ simulations are completed, estimate the mean and variance for each $\boldsymbol{x}_{*,j}$ using $\{\tilde{z}^{(s)}_{*,j}\}_{s=1:N_S}$.

SGS works according to the chain rule, which states that the full joint distribution can be written as a product of a marginal and univariate conditional distributions as shown in (13)

$$f(\boldsymbol{z}_{*,1:m} \,|\, \mathbf{X}_*, D) = f(z_{*1}; \boldsymbol{x}_{*1}, D) \cdot \prod_{i=2}^{m} f(z_{*i} \,|\, \boldsymbol{x}_{*,1:i}, \boldsymbol{z}_{*,1:i-1}, D) \tag{13}$$

where observed data $D = \{\boldsymbol{x}_i, z_i\}_{i=1:n}$, unknown points $\boldsymbol{x}_{*1:i} = (\boldsymbol{x}_{*,1}, \boldsymbol{x}_{*,2}, \dots, \boldsymbol{x}_{*,i})$ and the simulated/updated values $\boldsymbol{z}_{*,1:k} = (z_{*,1}, z_{*,2}, \dots, z_{*,k})$. This implies a random realisation $f(\boldsymbol{z}_{*,1:m} \,|\, \mathbf{X}_*, D)$ can be obtained, if a realisation of the univariate conditional distribution can be obtained for all subsets of conditional model parameters in any traversal order (Hansen, 2021). For simplicity, the permutation $\pi_s$ is omitted from the subscripted indices to avoid clutter.

**2.4 Spatially correlated random fields**

A random field refers to a family of random variables, $\boldsymbol{Z}_* = [Z(\boldsymbol{x}_{*1}), Z(\boldsymbol{x}_{*2}), \dots, Z(\boldsymbol{x}_{*m})]$, indexed by $\boldsymbol{x}_{*i} \in \mathbb{R}^d$. The objective of computing a GP spatially correlated random field (CRF) is to generate a random realisation of the stationary process





**Table 1.** Model candidates for probabilistic copper grade estimation

|   | Abbreviation | Description | Cross-reference |
|---|---|---|---|
| 1 | SK | Simple Kriging | §2.2 (9) (11) |
| 2 | SK-SGS | Simple Kriging + Sequential Gaussian Simulation | §2.2, §2.3 |
| 3 | OK | Ordinary Kriging | §2.2 (10) (12) |
| 4 | OK-SGS | Ordinary Kriging + Sequential Gaussian Simulation | §2.2, §2.3 |
| 5 | GP(L) | Gaussian Process Regression (local neighbourhood mean) | §2.1 (3) (4) |
| 6 | GP-SGS | Gaussian Process + Sequential Gaussian Simulation | §2.1, §2.3 |
| 7 | GP(G) | Gaussian Process Regression (with stationary/global mean) | §2.1 (3) (4) |
| 8 | GP-CRF | Gaussian Process Spatially Correlated Random Field | §2.1, §2.4 |

such that $\boldsymbol{Z}_* \sim \mathcal{N}(\boldsymbol{\mu}, \boldsymbol{\Sigma})$ follows a joint Gaussian distribution with mean vector $\boldsymbol{\mu}$ and covariance $\boldsymbol{\Sigma}$ as defined by a GP. Since the matrix $\boldsymbol{\Sigma}$ is symmetric and non-negative definite, it may be factorised via Cholesky decomposition to obtain an invertible lower triangular matrix, $\mathbf{L}$, such that $\mathbf{L}\mathbf{L}^T = \boldsymbol{\Sigma}$. Subsequently, a random realisation $\tilde{z}^{(s)} \in \mathbb{R}^m$ is readily obtained from (14)

$$\tilde{\boldsymbol{z}}_*^{(s)} = \mathbf{L}\boldsymbol{w} + \boldsymbol{\mu}, \quad \boldsymbol{w}[i] \sim \mathcal{N}(0,1) \tag{14}$$

where $\boldsymbol{w} \in \mathbb{R}^m$ represents an uncorrelated random vector drawn from the standard Normal distribution (Yang et al., 2022). As in sequential simulation, the mean function and standard deviation will be computed based on $N_S$ random realisations.

### 2.5 Model candidates

Altogether, eight model candidates (see Table 1) will be considered and compared during performance analysis. The kriging candidates (SK, SK-SGS, OK, OK-SGS) are self-explanatory. In GP(L), the mean estimates are conditioned on a local neighbourhood $N(\boldsymbol{x}_*)$ whereas for GP(G), a global (constant) mean is assumed. Another difference is that kriging models employ an isotropic kernel—specifically a Matérn covariance function with two parameters $r$ and $\nu$ for the range and shape—whereas GP models employ a fixed Matérn 3/2 kernel ($\nu = 3/2$) with heterogeneous length scales $(l_x, l_y, l_z)$ plus two extra parameters $(a^2, \sigma_v)$ for amplitude and noise. In general, the $\tilde{n}$ nearest neighbours, $\{\boldsymbol{x}_k \,|\, \boldsymbol{x}_k \in N(\boldsymbol{x}_*)\}$, are found using a search ellipsoid in a rotated space that aligns with feature orientation in the modelled domain. A final remark is that while normal score transformation (nst) is always applied in SGS and CRF simulation, for the candidates SK, OK, GP(L) and GP(G), modelling is performed on both raw data, and normal score transformed data. The latter will be annotated with 'nst' for clarity and requires extra care. Suppose the nst transformed values are denoted $y = g(z)$. Having obtained $\mathrm{Var}[g(Z)] \equiv \hat{\sigma}_Y^2$, the variance in $Z$, $\hat{\sigma}_Z^2$, must be computed using Taylor series approximation as shown in (15). In contrast, the mean is given by $\hat{\mu}_Z = g^{-1}(\mathbb{E}[Y])$.

$$\mathrm{Var}[g(Z)] = g'(\mu)[g'(\mu) - \mu g''(\mu)]\mathrm{Var}[Z] + \text{higher order terms} \implies \hat{\sigma}_Z^2 \approx \frac{\mathrm{Var}[g(Z)]}{g'(\mu)^2} \tag{15}$$





## 3 Geostatistical measures

This section describes the performance measures used for model evaluation. The first category are histogram distance measures that reflect global accuracy in the mean grade estimates. The second category are variogram-based measures which capture spatial correlation; these are meant to reflect spatial fidelity (local variability) in the model predicted mean. The third category

are uncertainty-based measures which assess the goodness of probabilistic models using both the mean and standard deviation estimates, $\hat{\mu}(\boldsymbol{x}_*)$ and $\hat{\sigma}(\boldsymbol{x}_*)$, and the groundtruth (actual grade) $\mu_0(\boldsymbol{x}_*)$.

### 3.1 Histogram-based measures

The general goal is to measure discrepancies between the mean prediction and groundtruth histograms. For simplicity, let $\boldsymbol{p}$ and $\boldsymbol{q}$ be the probability mass functions (pmf) for the model predicted mean and groundtruth vectors, $\hat{\boldsymbol{\mu}} = [\hat{\mu}(\boldsymbol{x}_{*i})]_{i=1:m}$ and

210 $\boldsymbol{\mu}_0 = [\mu_0(\boldsymbol{x}_{*i})]_{i=1:m}$, respectively. As an overview, the four chosen measures are motivated by hypothesis testing, information theory, set theory and the Monge-Kantorovich optimal transportation / distribution morphing problem (Chizat et al., 2018).

#### 3.1.1 Probabilistic symmetric Chi-square measure

The probabilistic symmetric $\chi^2$ histogram distance (Deza and Deza, 2009) is a symmetric variant of the regular $\chi^2$ distance. It represents twice the *triangular discrimination* defined by Topsøe (2000)

$$215 \quad h_{\text{psChi}} = 2 \sum_x \frac{|p(x) - q(x)|^2}{p(x) + q(x)} \tag{16}$$

#### 3.1.2 Jensen-Shannon divergence

The Jensen-Shannon divergence, $h_{\text{JS}}$ in (17), represents a symmetric and smoothed form of the KL divergence (Nielsen, 2019).

$$h_{\text{JS}} = \frac{1}{2} \left[ \sum_x p(x) \log \left( \frac{2p(x)}{p(x) + q(x)} \right) + \sum_x q(x) \log \left( \frac{2q(x)}{p(x) + q(x)} \right) \right] \tag{17}$$

$$= \frac{1}{2} [KL(\boldsymbol{p} \,||\, \boldsymbol{m}) + KL(\boldsymbol{q} \,||\, \boldsymbol{m})], \quad \text{where } KL(\boldsymbol{p} \,||\, \boldsymbol{q}) = \sum_x p(x) \log(p(x)/q(x)) \text{ and } \boldsymbol{m} = (\boldsymbol{p} + \boldsymbol{q})/2$$

The second line expresses $h_{\text{JS}}$ in terms of the Kullback-Leibler divergence which is also known as relative entropy. Using base-2 logarithm, this measure satisfies the bounds $0 \le h_{\text{JS}} \le 1$, attaining zero when $\boldsymbol{p} = \boldsymbol{q}$.

#### 3.1.3 Ruzicka distance

The Ruzicka distance, $h_{\text{Ruz}}$, is defined by Ruzicka similarity $S_{\text{Ruz}}$. Given two probability mass functions, $\boldsymbol{p}$ and $\boldsymbol{q}$,

$$h_{\text{Ruz}} = 1 - S_{\text{Ruz}} = 1 - \frac{\sum_x \min\{p(x), q(x)\}}{\sum_x \max\{p(x), q(x)\}} \tag{18}$$

$S_{\text{Ruz}}$ may be interpreted as the intersection between $\boldsymbol{p}$ and $\boldsymbol{q}$ over the union of $\boldsymbol{p}$ and $\boldsymbol{q}$ and abbreviated as IoU (Cha, 2008). It generalises the Jaccard similarity index from $\{0,1\}^m$ to $\mathbb{R}^m$. The Ruzicka distance is bounded by $0 \le h_{\text{Ruz}} \le 1$.




### 3.1.4 Wasserstein distance

The Wasserstein distance, $W_1$, also called the Earth-mover's distance, EM distance or Kantorovich optimal transport distance, is a similarity metric that may be interpreted as the minimum energy cost of moving and transforming a pile of dirt in the shape of one probability distribution into the other (Rubner et al., 1998). The cost is quantified by the distance and amount of probability mass being moved. It might be preferred over JS divergence, as the Kantorovich-Mallows-Monge-Wasserstein metric represents the Lipschitz distance between probability measures and has to be K-Lipschitz continuous. When the measures are uniform over a set of discrete elements, the problem is also known as minimum weight bipartite matching. Formally, the $k$-Wasserstein distance between probability distributions $P$ and $Q$ is defined as an infinum over joint probabilities

$$h_{\text{EM}}(P,Q) \equiv W_k(P,Q) = \inf_{\gamma \in \prod(P,Q)} \mathbb{E}_{(x,y)\sim\gamma}[d(x,y)] \tag{19}$$

where $\prod(P,Q)$ is the set of all joint distributions whose marginals are $P$ and $Q$. In general, it requires solving a linear assignment problem. However, in one-dimension, it may be computed simply using order statistics. In particular,

$$W_k(P,Q) = \left( \frac{1}{m} \sum_i^m |\tilde{\boldsymbol{p}}_{(i)} - \tilde{\boldsymbol{q}}_{(i)}|^k \right)^{1/k} \tag{20}$$

where $\tilde{\boldsymbol{p}}_{(i)}$ and $\tilde{\boldsymbol{q}}_{(i)}$ refer to the $i^{\text{th}}$ element in the sorted sequence of $\hat{\boldsymbol{\mu}} = [\hat{\mu}(\boldsymbol{x}_{*j})]_{j=1:M}$ and $\boldsymbol{\mu}_0 = [\mu_0(\boldsymbol{x}_{*j})]_{j=1:M}$. Therefore, it does not require quantisation or conversion of $\hat{\boldsymbol{\mu}}$ and $\boldsymbol{\mu}_0$ into histograms or pmf.

### 3.2 Variogram-based measures

A variogram ratio statistic is proposed as a basis for measuring the loss of spatial fidelity in a model. This partly stems from the widespread use of semi-variograms in geostatistics. The variogram curve $\gamma(d)$, see Oliver et al. (2015), measures the inverse correlation between points as a function of their separating distance, $d$. A key feature of this curve is the *sill* which refers to the maximum height of $\gamma(d)$ at large distances as samples become uncorrelated. Hence, when two variograms are compared, and the sill associated with a model is lower than the sill for a reference, it is indicative of smoothing or a reduction in spatial variability. Formally, this may be represented by the ratio between two variogram curves as shown in (21)

$$r_{\text{model}}(d) = \frac{\gamma_{\text{model}}(d)}{\gamma_{\text{reference}}(d)} \tag{21}$$

Percentile statistics such as the median or lower/upper quantiles can be computed from $r_{\text{model}}(d)$ to indicate the average loss in spatial fidelity (equivalently, attenuation in signal power). This allows a visual diagnostic tool to be converted into a quantitative measure. In practice, the sill may also be raised, particularly when the result from one sequential simulation is examined. Thus, a ratio that increases far beyond 1 is also undesirable as it signifies noise amplification. For this reason, the following convex function (symmetrical about $R = 1$) is proposed as a proxy measure for spatial fidelity.

**Spatial Fidelity** $\quad F(R) = \sqrt{1 - |\min\{R,2\} - 1|}$, where $R = \text{median } r_{\text{model}}(d)$ $\tag{22}$

This needs to be interpreted with care in respect of a suitable reference, such as verification data (actual groundtruth values at the predicted locations) or the training data from which the kriging variogram or GP kernel hyperparameters are learned.





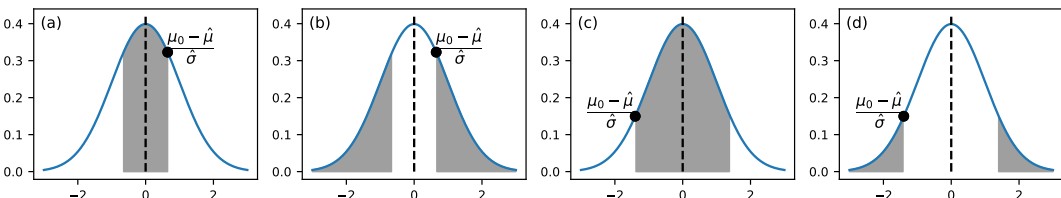

**Figure 2.** Illustration of coverage probability in (a,c) and likelihood the model is correct in (b,d)

## 3.3 Uncertainty-based measures

All models (listed in Table 1) will provide an estimate $(\hat{\mu}_j, \hat{\sigma}_j)$ at inference location $\mathbf{x}_{*j}$ and be compared with the groundtruth $\mu_{0,j}$. Under the Gaussian assumption, the second-order statistics $\{(\hat{\mu}_j(\boldsymbol{x}_{*j}), \hat{\sigma}_j(\boldsymbol{x}_{*j}))\}_{j=1:m}$ are sufficient for characterising

the conditional distribution $p(\boldsymbol{f}_* \mid \mathbf{X}_*, \mathbf{X}, \boldsymbol{f})$. Since it is clear that we are dealing with predicted points, henceforth the $\boldsymbol{x}_*$ notation will be dropped. To assess how reasonable these estimates are given verification data $\{\mu_{0,j}\}_{j=1:m}$, it is useful to convert $(\mu_{0,j}, \hat{\mu}_j, \hat{\sigma}_j)$ into Z scores via $z_j = (\mu_{0,j} - \hat{\mu}_j)/\hat{\sigma}_j$. In Fig. 2, the black dots represent true values. Panels (a-b) illustrates the case where the model mean *underestimates* the true value ($\hat{\mu} < \mu_0$) while panels (c-d) illustrates the case where the true value is *overestimated* ($\hat{\mu} > \mu_0$). The shaded area in (a) represents the coverage probability, $p$. Its complement, $1-p$, describes

the likelihood that the model is correct once validation measurement is revealed. This corresponds to the area under the tail sections in (b). To distinguish overestimation from underestimation, we define a signed scoring function $s$ called *synchronicity*.

**Synchronicity** $\quad S \equiv s(\hat{\mu}, \hat{\sigma} \mid \mu_0) = \begin{cases} 2 \times [1 - \Phi(z)] & \text{if } \mu_0 \geq \hat{\mu} \quad \text{[underestimating]} \\ -2\,\Phi(z) & \text{otherwise} \quad \text{[overestimating]} \end{cases}$ (23)

$$= 2 \times [\mathbb{I}(\hat{\mu} \leq \mu_0) \cdot (1 - \Phi(z)) - (1 - \mathbb{I}(\hat{\mu} \leq \mu_0)) \cdot \Phi(z)]$$ (24)

where $z = (\mu_0 - \hat{\mu})/\hat{\sigma}$ and $\Phi(z) = \frac{1}{\sqrt{2\pi}} \int_{-\infty}^{z} e^{-t^2/2} dt$ denotes the CDF of the standard normal distribution. This may be written

more compactly using an indicator function in (24). The likelihood is simply the magnitude of $s$, as shown in (25).

**Likelihood** $\quad L \equiv l(\hat{\mu}, \hat{\sigma} \mid \mu_0) = |s(\hat{\mu}, \hat{\sigma} \mid \mu_0)|$ (25)

### 3.3.1 Goodness of model predicted uncertainty

One criterion for assessing local uncertainty prediction accuracy is based on the Deutsch (1997) goodness statistic. By construction, there is a probability $p$ (identical to the coverage probability described in Sec. 3.3) that the true value of the random

variable falls within a symmetric $p$-probability interval (PI) bounded by the $p_L = (1-p)/2$ and $p_U = (1+p)/2$ quantiles, $Q_L \equiv Q_{(1-p)/2}$ and $Q_U \equiv Q_{(1+p)/2}$, of the estimated conditional distribution function (Fouedjio and Klump, 2019). As a special case, when $p = 0.5$, $Q_L$ and $Q_U$ correspond to the lower and upper quartiles, $Q_{0.25}$ and $Q_{0.75}$, respectively. Given validation measurements $\{\mu_{0,j}\}_{j=1:m}$ at inference locations $\{\boldsymbol{x}_{*j}\}_{j=1:m}$, one is interested in $\bar{\kappa}(p)$, the fraction of true values





that are bounded by the PI interval with probability $p$. Concretely, the expression for $\bar{\kappa}(p)$ in (26) computes the empirical mean

over all test locations as a function of $p$.

$$\bar{\kappa}(p) = \frac{1}{m} \sum_{j=1}^{m} \kappa_j(p), \tag{26}$$

$$\kappa_j(p) = \begin{cases} 1 & \text{if } \hat{Q}_{(1-p)/2}(j) < Y_j < \hat{Q}_{(1+p)/2}(j) \\ 0 & \text{otherwise} \end{cases} \tag{27}$$

In practice, when the random process (fluctuations about the posterior mean function) is modelled as Gaussian, symmetrical intervals are obtained following Z score transformation, so effectively $\hat{Q}_{(1-p)/2}(j) = -\hat{Q}_{(1+p)/2}(j)$. The mean proportion $K$

is given by the integral $K = \int_0^1 \bar{\kappa}(p) dp$.

### 3.3.2 Accuracy of the estimated distribution

The average of $[\bar{\kappa}(p) \geq p]$ over $p$ is known as *distribution accuracy*, $A_\xi$, as shown in (28).

$$\textbf{Accuracy} \quad A_\xi = \int_0^1 \mathbb{I}_\xi(p) dp, \quad \mathbb{I}_\xi(p) \equiv \mathbb{I}(\bar{\kappa}(p), \xi) = \begin{cases} 1 & \text{if } \bar{\kappa}(p) \geq (1-\xi)p \\ 0 & \text{otherwise} \end{cases} \quad \text{for a slack variable } \xi \in [0, 0.1] \tag{28}$$

### 3.3.3 Precision of the estimated distribution

Precision measures the narrowness of the model estimated distribution. It is only defined for accurate probability distributions. A $p$-probability interval that recalls more than $p\%$ of true values is accurate but not precise. Optimal precision means the $p$-PI contains the true values exactly $p\%$ of time. On this basis, the precision of the estimated distribution is defined by Deutsch (1997) as

$$\textbf{Precision} \quad P = 1 - 2 \int_0^1 \mathbb{I}_0(p) \left[ \bar{\kappa}(p) - p \right] dp \tag{29}$$

The precision is only meaningful when there is accuracy. In other words, when the estimated proportions $\bar{\kappa}(p)$ are consistently above the expected proportions $p$. This can be checked from the accuracy plot of $\bar{\kappa}(p)$ vs $p$ in the bottom half of Fig. 3.

### 3.3.4 Prediction uncertainty goodness statistic

The closeness between the estimated and theoretical proportions is quantified by $G$ in (30)

$$\textbf{Goodness} \quad G = 1 - \int_0^1 \left[ 3\mathbb{I}_0(p) - 2 \right] \left[ \bar{\kappa}(p) - p \right] dp \tag{30}$$

The $G$ statistic indicates the closeness of points to the bisector of the $\kappa$-accuracy plot. Unlike the accuracy and precision, this also considers instances where $\bar{\kappa}(p) < p$. $G = 1$ when $\bar{\kappa}(p) = p \, \forall p \in [0, 1]$. $G = 0$ when none of the true values are contained





in any PIs. The choice of weights indicates that $\bar{\kappa}(p) < p$ is more consequential. The penalty for $\bar{\kappa}(p) < p$ (when observed proportions are below expectation) is twice that for $\bar{\kappa}(p) > p$.

### 3.3.5 Width of prediction uncertainty

For models with similar goodness statistics, one would prefer a model where the $p$-probability interval is as narrow as possible. A model (or conditional cumulative distribution function) that consistently provides narrow and accurate PIs should be preferred over another that provides wide and accurate PIs. Different notions of spread such as entropy, variance or inter-quartile range can be used. Goovaerts (2001) proposed using the interval width in (31) to measure the average tightness of the p-PIs subject to containment of the true value

$$\bar{W}(p) = \frac{1}{m\bar{\kappa}(p)} \sum_{j=1}^{m} \kappa_j(p) \left[ \hat{Q}_{(1+p)/2}(j) - \hat{Q}_{(1-p)/2}(j) \right] \tag{31}$$

### 3.3.6 Prediction uncertainty tightness statistic

The average width of $\bar{W}(p)$ over $p$ can be defined in an analogous manner to $A$. However, it is more difficult to interpret since it is highly dependent on the data. To make the tightness scale more meaningful, the average uncertainty interval is normalised by the process standard deviation $\sigma_Y$ observed in the validation measurements or groundtruth as shown in (32).

**Interval tightness** $\quad I = \frac{1}{\sigma_Y} \int\limits_0^1 \bar{W}(p)dp$ $\tag{32}$

In general, both $G$ and $I$ need to be taken in account when assessing probabilistic models, because uncertainty cannot be artificially reduced at the expense of accuracy (Deutsch, 1997).

### 3.3.7 Connections

The calculation of $\kappa_j(p)$ can be reframed in terms of $s(\hat{\mu}_j, \hat{\sigma}_j \mid \mu_{0,j})$ or $l(\hat{\mu}_j, \hat{\sigma}_j \mid \mu_{0,j})$. Instead of searching for $Q_{(1-p)/2}$ and
$Q_{(1+p)/2}$ for various $p$, there exists a critical value $p^*$ at which $z_{0,j} = (\mu_{0,j} - \hat{\mu}_j)/\hat{\sigma}_j$ lies just on the edge of $[q_L(j, p^*), q_U(j, p^*)]$. This is precisely the purpose of $l(\hat{\mu}_j, \hat{\sigma}_j \mid \mu_{0,j})$ which converts each input $(\hat{\mu}_j, \hat{\sigma}_j; \mu_{0,j})$ into a Z score $z_{0,j}$ and maps either $q_L(j, p^*)$ or $q_U(j, p^*)$ to $1 - p^*$. Since the interval grows with $p$, $\kappa_j(p) = 1$ for all $p \geq p^*$, where $p^* = 1 - l(\hat{\mu}_j, \hat{\sigma}_j \mid \mu_{0,j})$. In the next subsection, we will reinforce the general concepts with an example and demonstrate the efficacy of the uncertainty-based statistics using synthetic data where the groundtruth is known.

## 3.4 Illustration

From left to right, the panels in Fig. 3 illustrate three scenarios. The columns labelled (a), (b) and (c) correspond to an optimistic, the preferred, and conservative settings, respectively. If we restrict our attention to (a), the probabilistic predictions and uncertainty interpretations ($\kappa$ accuracy plots) occupy the top and bottom halves, respectively. For the predictions, the green curve represents the groundtruth. Each prediction at location $x_*$ consists of the mean and uncertainty which are represented by





a dot and vertical bar that signifies $\pm\hat{\sigma}$. This length is somewhat arbitrary; its purpose is to emphasize that we have a predictive distribution. The current choice, $[\hat{\mu}-\hat{\sigma}, \hat{\mu}+\hat{\sigma}]$, corresponds to $[Q_{(1-p)/2}, Q_{(1+p)/2}]$ where $p \approx 0.68$. In blue and black, we have two noisy models. They differ in terms of how much their mean predictions gravitate toward the actual mean, $\mu_0(x_*)$. Model 1 is simulated using a uniform distribution, so its estimated means are more spread out. Model 2 is simulated from a normal distribution, so its mean predictions, $\hat{\mu}(x_*)$ tend to be concentrated around $\hat{\mu}_0(x_*)$ but its tail values extend further out. Both

models in (a) are considered over-confident, as only a small fraction of the $p$ probability intervals contain the actual mean. This can be seen in the $\kappa$ accuracy plots which show the observed truth containment ratios, described by $\bar{\kappa}(p)$ in (26), consistently below the expected proportions ($p$) for most values of $p$. The ideal situation is depicted in (b) where $\bar{\kappa}(p)$ is close to $p$ and the models (especially model 2) live up to expectations. The models in (c) are considered pessimistic because $\bar{\kappa}(p)$ far exceeds $p$, as can be seen from the $\kappa$ accuracy plots.

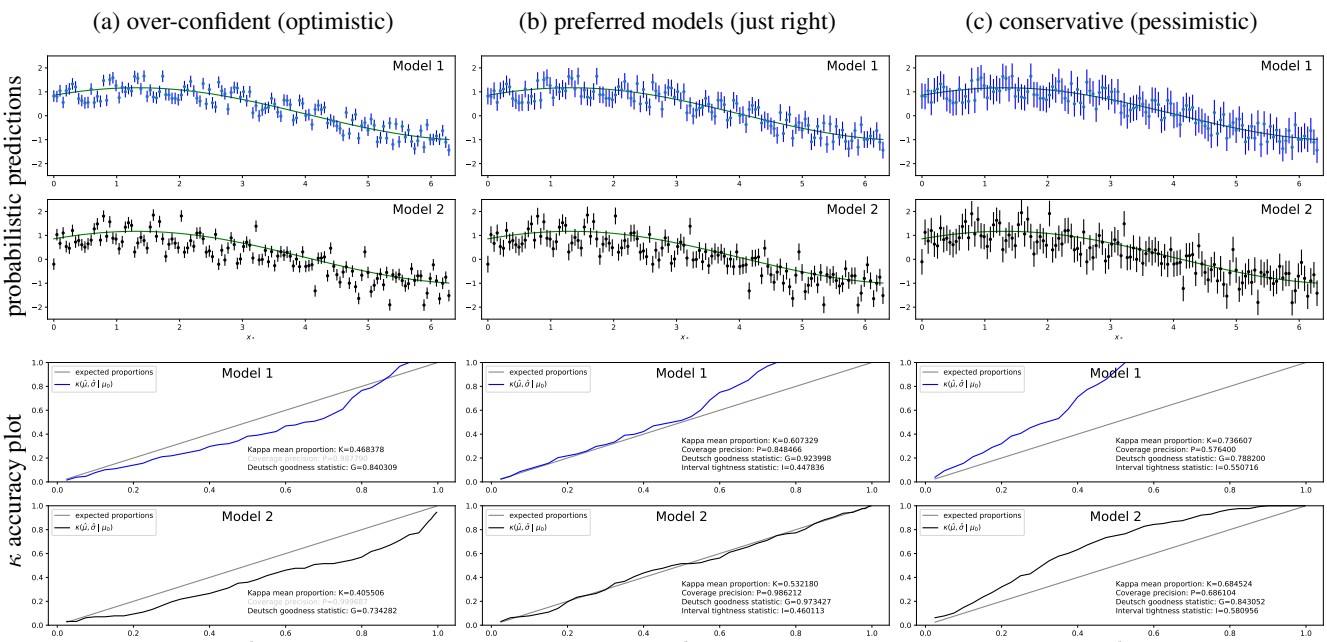

**Figure 3.** Evaluating the uncertainty and predictive performance of two synthesized models. Top: probabilistic predictions. Bottom: $\kappa$ accuracy plots. From left to right, (a), (b) and (c) show what can be expected from the optimistic, preferred and conservative models.

Inspection of Table 2 confirms that these observations are reflected in the statistics. The likelihood ($L$) and proportion ($K$) are lowest in the over-confident scenario (a) where the $p$-probability intervals captured fewer true values than what were expected. As this situation is remedied in the preferred scenario (b), the precision ($P$) and goodness ($G$) statistics have markedly improved. For instance, $G$ increased from 0.734 to 0.973 for model 2. The interval tightness measure ($I$) is highest/worst in the pessimistic scenario (c) while the precision and goodness statistics have also suffered. For instance, $P$ plummeted to 0.576





**Table 2.** Uncertainty-based statistics for the example in Figure 3

| Model | Setting | $\hat{\mu}$ distribution[†] | $\hat{\mu}$ bias | $\hat{\sigma}$ amplitude[†] | Likelihood($L$) | Proportion($K$) | Precision($P$) | Goodness($G$) | Tightness($I$) |
|-------|---------|------------------|---------|-------------------|-----------------|-----------------|----------------|---------------|----------------|
| a.1 | optimistic | uniform | 0 | 0.15 | 0.430 | 0.468 | – | 0.840 | 0.336 |
| b.1 | preferred | uniform | 0 | 0.225 | 0.576 | 0.607$^\uparrow$ | 0.848$^\checkmark$ | 0.923$^\uparrow$ | 0.447 |
| c.1 | pessimistic | uniform | 0 | 0.35 | 0.711 | 0.736$^\uparrow$ | 0.576$_\downarrow$ | 0.788$_\downarrow$ | 0.550 |
| a.2 | optimistic | normal | -0.2 | 0.15 | 0.369 | 0.405 | – | 0.734 | 0.331 |
| b.2 | preferred | normal | -0.2 | 0.225 | 0.494 | 0.532$^\uparrow$ | 0.986$^\checkmark$ | 0.973$^\uparrow$ | 0.460 |
| c.2 | pessimistic | normal | -0.1 | 0.35 | 0.656 | 0.684$^\uparrow$ | 0.686$_\downarrow$ | 0.843$_\downarrow$ | 0.580 |

[†] $\hat{\mu}$ distribution controls the central tendency and spread of the predicted mean, $\hat{\sigma}$ amplitude scales the prediction interval to emulate different model behaviours.

and 0.686 for models 1 and 2, respectively. These findings are consistent with our expectations. They reveal the strengths and weaknesses of models and show promise for large scale model evaluation using real-world data.

## 4 Experiments

This section describes the geological setting, data attributes, design and implementation of the experiments.

### 4.1 Geological setting

The data used in our experiments were obtained from the Bingham Canyon (Kennecott) open-pit copper mine located in Utah. It is classified as a porphyry skarn-hosted copper deposit which describes a copper orebody formed from hydrothermal fluids that originate from a magma chamber. Predating or associated with those fluids are multiple intrusions and vertical dikes of diorite to quartz monzonite composition with porphyritic textures. This basically refers to the appearance of large crystals set in a finegrained or glassy groundmass on the surface of igneous rocks which gives rise to its name. Metasomatism further
explains how the rocks undergo compositional and mineralogical transformations associated with chemical reactions triggered by the reaction of fluids which invade the protolith (Lesher and Spera, 2015). Detailed description of its geomorphology and mineralogical properties can be found in (Porter et al., 2012; Redmond and Einaudi, 2010). A major orebody characteristic is that successive envelopes of hydrothermal alteration typically enclose a core of disseminated ore minerals in a complex system of hairline fractures and veins known as *stockwork* (see Redmond and Einaudi, 2010, Figs. 2, 3, 9 and 10). This mineralisation
produces pit maps with >1%, >0.7%, >0.35% and >0.15% copper gradation extending from the inner to the outer zones.

### 4.2 Data attributes

The input used for modelling consists of the location $x$ and grade $y$ of blasthole assay measurements taken roughly $20 \pm 5$m apart. The sampling, assaying techniques and geological interpretations are elaborated in (Hayes and McInerney, 2022). This data is grouped spatiotemporally, resulting in 11 geological domains and 12 inference periods. Each geological domain is




represented by a four digit code LGPR which represent the limb zone, grade zone, porphyry zone and rock type, respectively. These are determined by geologists based on stratigraphy, lithology and other relevant information that control mineralisation and ore/waste boundaries; see (Porter et al., 2012, p.136–137) and (Redmond and Einaudi, 2010, p.49–60). The spatial structure of these domains can be seen in Fig. 4. An important property is that geochemical diversity is localised and reflected through the grade distribution in these domains. This can be seen in Fig. 5. From a modelling perspective, variations in the grade

distribution (in terms of skewness, dispersion and shape) across different domains are useful as they mitigate selection bias and allow the robustness of models to be properly tested.

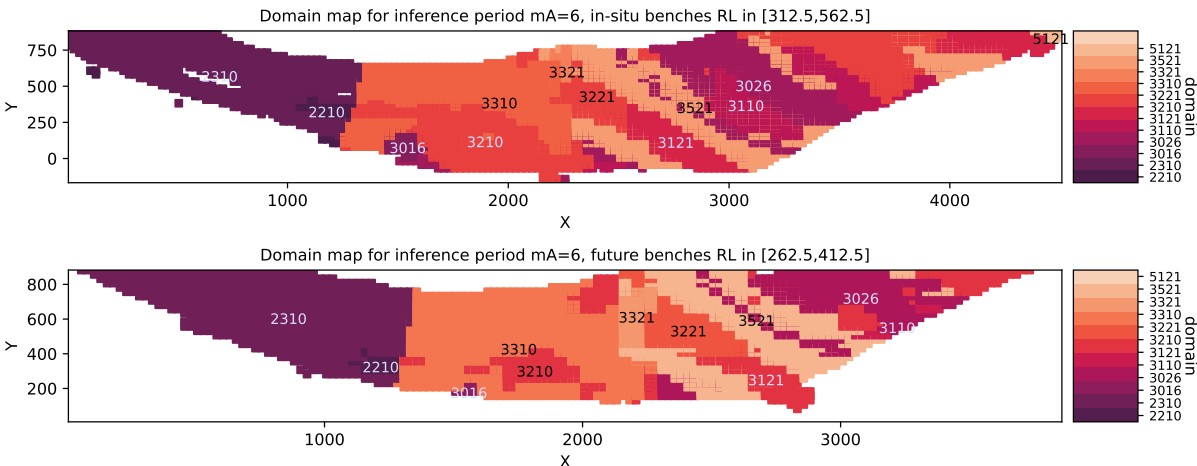

**Figure 4.** The orebody is partitioned into different geological domains to facilitate copper grade modelling. The panels show correlated spatial structure and subtle changes at different RL elevations. Top image shows blocks associated with active mining operations, i.e. benches that may benefit from in-situ regression. Bottom image shows blocks associated with future-bench prediction which require extrapolation. It should be noted that the actual spatial coordinates have been shifted so that the minimum coordinates of the study area are close to the origin in the Cartesian coordinates system. This applies also to the RL elevation (ft) to anonymise the data due to commercial sensitivity.

## 4.3   Experimental design

The experiments were designed to emulate the staged observations and progression of mining operations in a real mine. For *future-bench prediction*, each inference period ($m_A$) signals the intent that the probabilistic models will use data gathered

prior to the month of $m_A$ to predict into new locations relevant to mine planning for the next three months (for instance, the months of April, May and June if $m_A$=4). These new locations represent regions below or adjacent to the current bench, thus blasthole measurements will not be available since these benches have not yet been drilled or developed. A less technically challenging problem is *in-situ regression* which requires interpolating the grade within current benches or operating areas where excavation activities might be planned; the distinction is that this does not require extrapolation into new territories. The

number of blasthole samples available for training ($n$) and inference locations that require grade prediction ($m$) are both highly variable; some statistics are shown in Table 3.





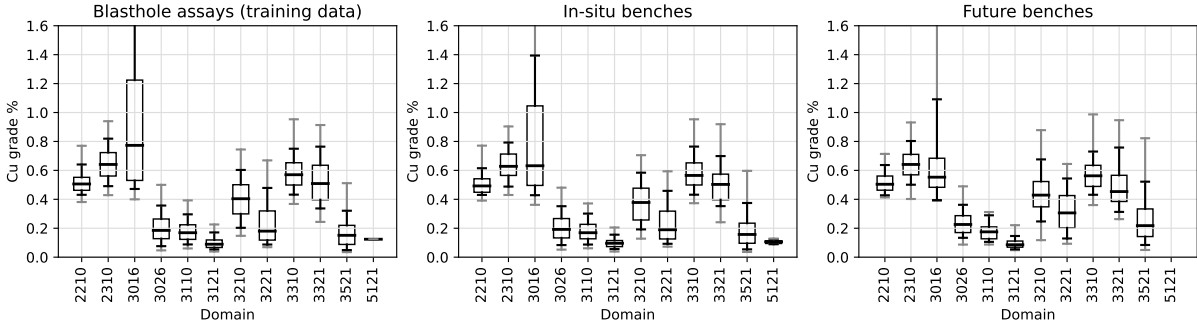

**Figure 5.** Copper grade distribution across different domains for inference period $m_A$=6. In each box-plot, the outer (faint) and inner (dark) whiskers represent the 2.15/97.85 and 8.87/91.13 percentiles, whereas the horizontal bar and box edges represent the median and lower/upper quartiles, respectively. From an economic perspective, porphyry orebodies can be mined profitably from Cu concentrations as low as 0.15–0.3%. The left, middle and right plots pertain to blasthole training data, blocks that require in-situ regression, and future-bench prediction, respectively. As expected, in-situ distributions more closely resemble the training data than future-bench distributions.

**Table 3.** Sample size statistics for in-situ interpolation and future-bench extrapolation

| | Number of blastholes samples for training[†] | | Number of locations requiring prediction[†,‡] | |
|---|---|---|---|---|
| | In-situ | Future-bench | In-situ | Future-bench |
| Lower quartile | 133 | 481 | 96 | 26 |
| Median | 694 | 1112 | 680 | 138 |
| Upper quartile | 2338 | 2293 | 1968 | 401 |
| Total count ⋆ | 148302 | 160459 | 125004 | 29689 |

[†] All figures refer to quantiles per domain, per inference period unless otherwise stated

[‡] Includes only instances where validation measurements are available. ⋆ Sum over all domains and inference periods

## 4.4 Implementation

The **eup3m.git** repository provides a Python implementation of all the algorithms described in this paper.

The **run_experiments.py** script executes one single experiment at a time given an inference period / month ($m_A$) and geological domain ($g_D$) as input, using the standard configuration parameters specified in **rtcma_utils.py**. A bash script is used to run a complete set of experiments asynchronously on a machine with 30 CPUs, iterating over $m_A$ and $g_D$ to produce the full results. Each individual experiment has a model construction phase and performance analysis phase (refer to the eight model candidates in Table 1 and statistical measures described in Sec. 3). The measures implemented in **rtcma_evaluation_metrics.py** utilise the **stats**, **spatial** and **special** libraries in **scipy**. The GP approaches implemented in **gstatsim3d_gaussian_process.py** utilise the **scipy.linalg** and **scikit-learn** packages. The kriging approaches implemented in **gstatsim3d_kriging.py** utilise the **scikit-gstat** package and extends existing functionalities in **GStatSim** to support 3D data and an irregularly spaced inference grid. For sequential simulation, the selection of





random paths is domain and inference-period dependent, however, the sequence remains the same for the SK-SGS, OK-SGS and GP-SGS models in each simulation run, $s$. To achieve consistent and reproducible results, a SHA256 hash is computed for each ($m_A$, $g_D$) pair to initialise the state of a random generator, then $N_S$ values are drawn to obtain $N_S = 128$ random seeds which will subsequently determine the order $\pi_s$ in which the $\{\boldsymbol{x}_{*,j}\}_{j=1:m}$ points are visited as described in Sec. 2.3.

## 5 Results

The ensuing analysis is organised in two parts to target two related objectives. The first is to familiarise with the proposed statistics and assess how they respond to real data. To reinforce concepts and develop real insight, we will devote our attention to in-depth analysis of two domains in one inference period, examining histogram distances, variogram-ratios visually and uncertainty-based measures quantitatively. The second objective is to systematically evaluate the uncertainty and predictive performance of the chosen probabilistic models and interpret the results across all domains and inference periods.

### 5.1 Analysis 1: Specific domains within a single inference period

The analysis throughout Sec. 5.1 pertains to future-bench prediction. Two geological domains (2310 and 3521) with vastly different geochemical characteristics were selected for analysis from one inference period ($m_A = 4$). The copper concentration reported in the blasthole training data and groundtruth (for estimated locations in future benches) are depicted in the left and right columns in Fig. 6, respectively. The reason for including these is to show explicitly the known data $\{\boldsymbol{x}_i, y_i\}_{i=1:n}$ used for fitting variograms, learning kriging weights or GP kernel hyperparameters (on the left) and actual grades for predicted points $\{\boldsymbol{x}_{*j}\}_{j=1:m}$ (on the right) where verification measurements are available. Focusing on domain 2310 first, Fig. 7 shows the mean grade predicted by all eight models. A couple of observations stem from these results. First, techniques that rely on (or assume) a stationary mean, such as SK and GP(G), tend to produce predictions that are too smooth compared with the groundtruth in Fig. 6. Second, simulations—whether it is SGS or CRF—can improve the spatial fidelity of the predictions. As expected, this quality restorative property recedes as the number of simulation rounds ($s$) increases, albeit at different rates for OK-SGS and GP-SGS; this is illustrated in Fig. 8. These behaviours are amplified in domain 3521 which exhibit high grade mineralisation in the northwestern tip. The same observations on oversmoothing and the benefits of simulation can be seen more clearly in Figs. 9 and 10. In particular, SK and GP(G) both significantly underestimate the Cu peaks.

Since the model candidates produce probabilistic predictions, it would be apt to include results for variance predictions in Figs. 11–14. As the kriging variance in (11)–(12) does not depend on the observed values, its legitimacy is often questioned. In the mining geology context, the kriging variance largely reflects the epistemic uncertainty—for example, due to a lack of training data or spatial sampling—rather than the aleatoric uncertainty which is concerned with inherent grade variability within the deposit. In Figs. 11 and 13, it can be seen the kriging standard deviations lack any meaningful spatial variation that the GPs convey. Nevertheless, the kriging variance is comparable to the average GP variance in these examples when normal score transformation is applied. This, to some extent, corroborates with the findings in (Heuvelink and Pebesma, 2002) which



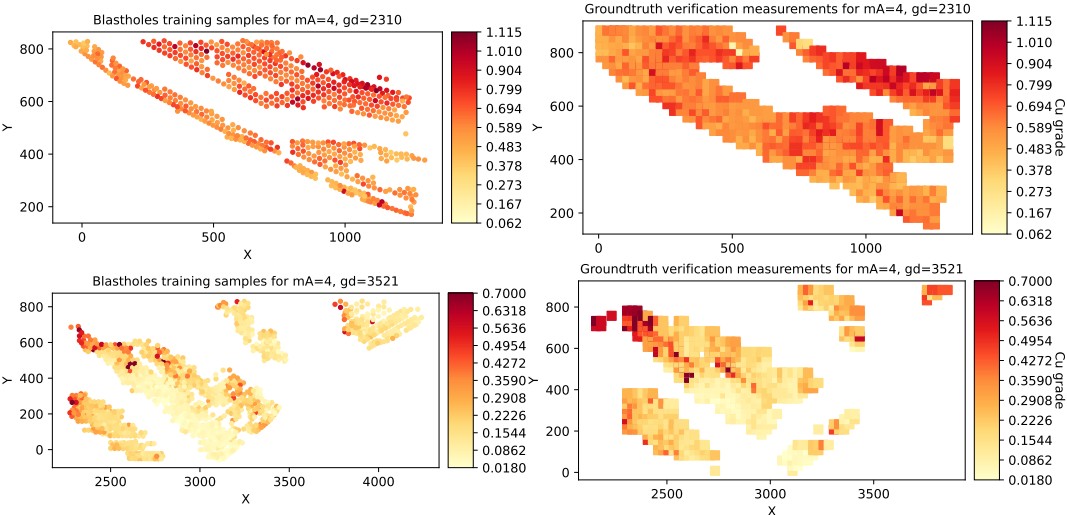

**Figure 6.** Visualisation of copper grade for blasthole training data (left) and groundtruth at predicted locations (right) for two domains 2310 (top) and 3521 (bottom) in inference period $m_A = 4$.

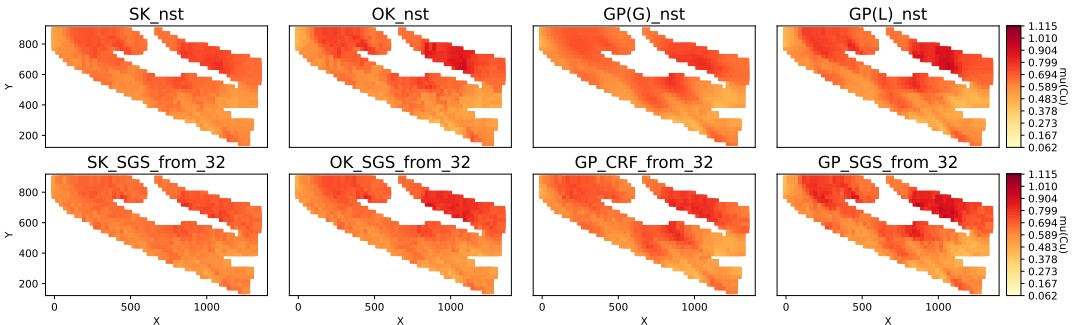

**Figure 7.** Mean copper grade predicted by models for domain $g_D = 2310$ and inference period $m_A = 4$.

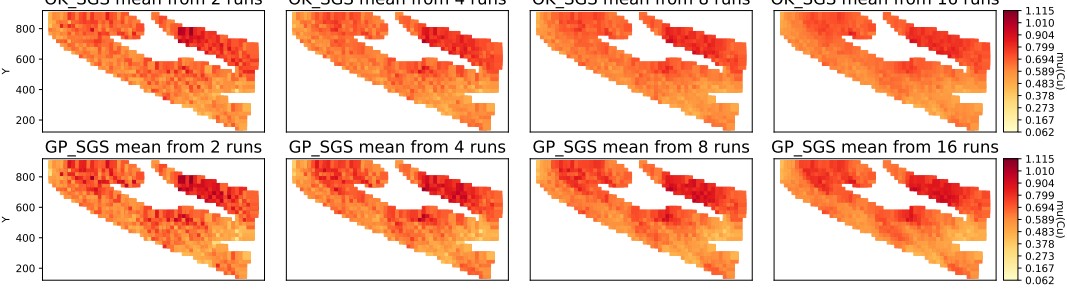

**Figure 8.** Mean copper grade estimated from $s$ simulations ($s = 2, 4, 8, 16$) for $g_D = 2310$ and $m_A = 4$.



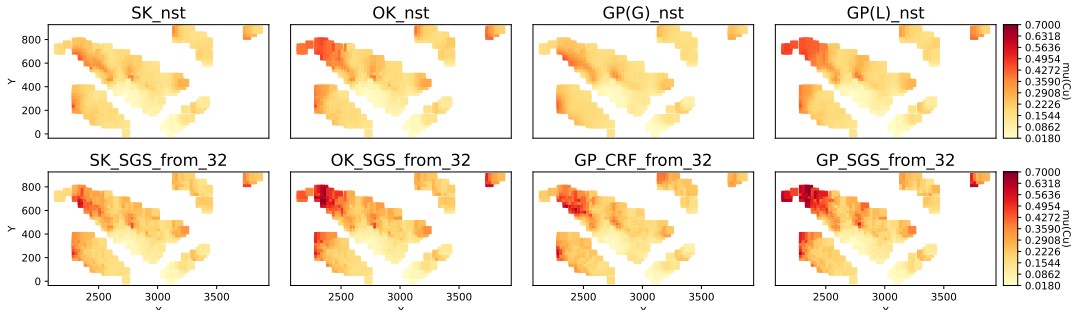

**Figure 9.** Mean copper grade predicted by models for domain $g_D = 3521$ and inference period $m_A = 4$.

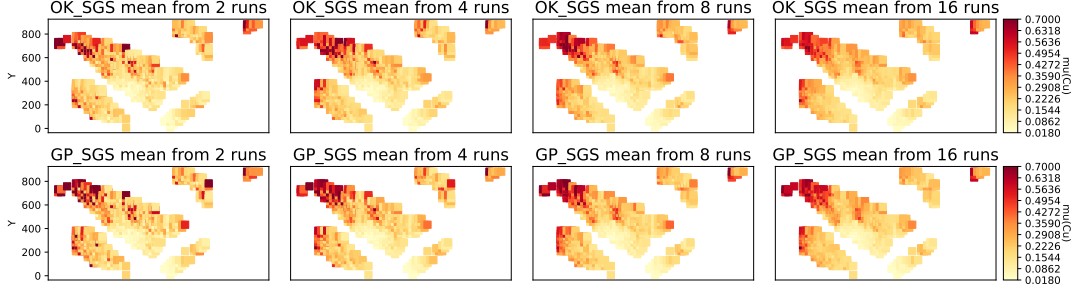

**Figure 10.** Mean copper grade estimated from $s$ simulations ($s = 2, 4, 8, 16$) for $g_D = 3521$ and $m_A = 4$.

suggest the kriging variance estimates might be reasonable under Gaussian conditions. In Fig. 13, simulations can be seen to reveal *heterogeneity* in the uncertainty and potentially improve on stand-alone GP or kriging variance estimates.

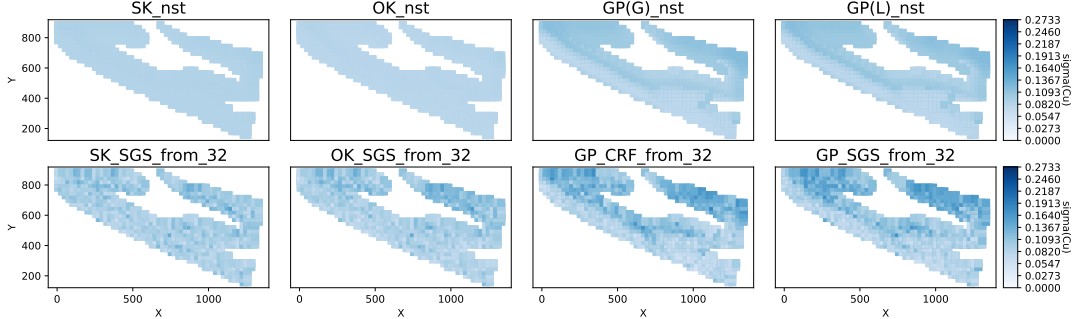

**Figure 11.** Copper grade standard deviation predicted by models for domain $g_D = 2310$ and inference period $m_A = 4$.




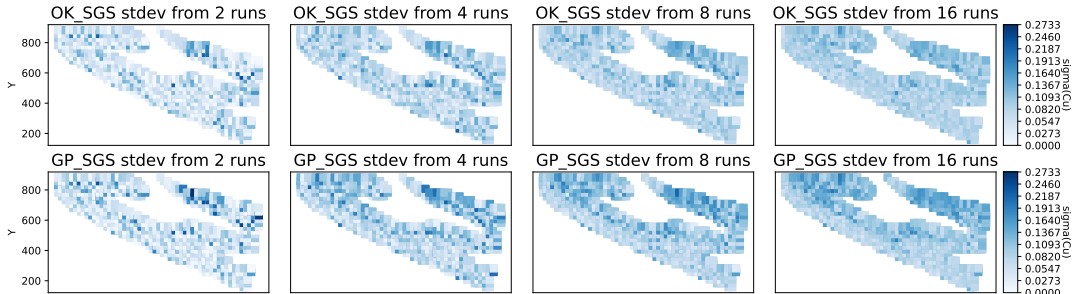

**Figure 12.** Copper grade standard deviation estimated from $s$ simulations ($s = 2, 4, 8, 16$) for $g_D = 2310$ and $m_A = 4$.

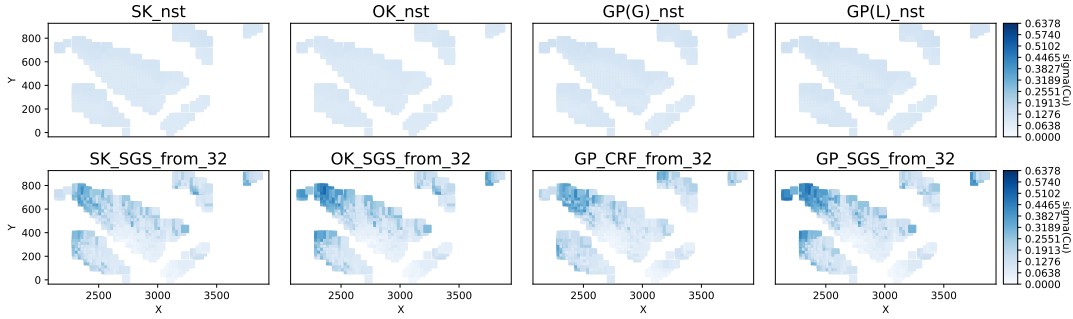

**Figure 13.** Copper grade standard deviation predicted by models for domain $g_D = 3521$ and inference period $m_A = 4$.

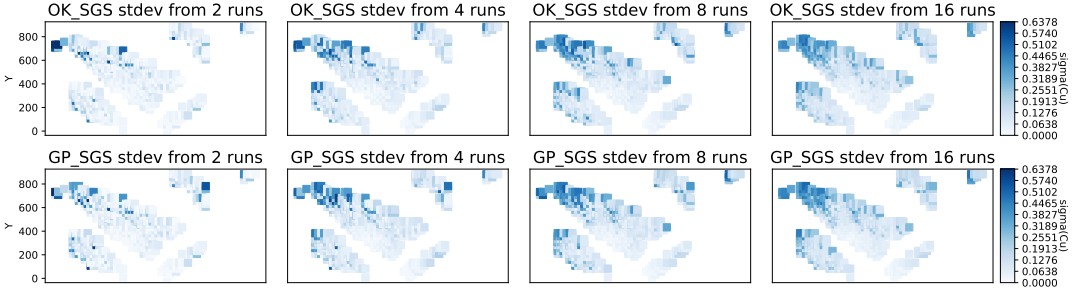

**Figure 14.** Copper grade standard deviation estimated from $s$ simulations ($s = 2, 4, 8, 16$) for $g_D = 3521$ and $m_A = 4$.

### 5.1.1 Histograms

To assess global accuracy, histograms are rendered in Figs. 15 and 16. In these bar graphs, the groundtruth and mean model predictions are represented by black hollow and blue filled columns, respectively. Visually, GP-SGS and OK-SGS can be seen to provide a closer approximation to the groundtruth probability mass function (pmf) whereas the range is more compressed in the case of SK and GP(G), resulting in inadequate coverage of the tail(s).






The probabilistic symmetric $\chi^2$, Jensen-Shannon, Ruzicka and Wasserstein histogram distances (described in Sec. 3.1.1–3.1.4) are computed and presented in Table 4. Although the trend varies somewhat depending on the domain, a common observation is a general improvement in the histogram rank (equivalently, reduction in histogram distances) when SGS or CRF is coupled with GP. Overall, the computed histogram distances are consistent with our graph-based interpretations.

In Fig. 17, the cross-plots show that $h_{\mathrm{psChi}}$ and $h_{\mathrm{JS}}$ are linearly correlated ($\rho > 0.99$), while $h_{\mathrm{Ruz}}$ is strongly correlated with both $h_{\mathrm{psChi}}$ and $h_{\mathrm{JS}}$ $\rho \in [0.84, 0.97]$). Since the Jensen-Shannon divergence can be interpreted as information difference between two distributions and it is bounded, we suggest it should be included in general assessment alongside the Wasserstein histogram measure, $h_{\mathrm{EM}}$, as the latter is not dependent on quantisation and less sensitive to sample size.

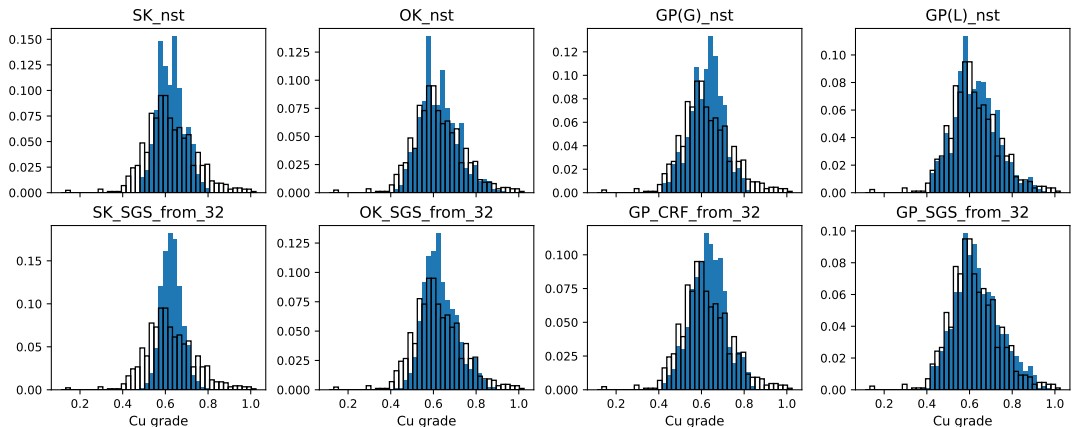

**Figure 15.** Copper grade histograms for $g_D = 2310$ and $m_A = 4$. Black hollow: groundtruth. Blue: model predictions

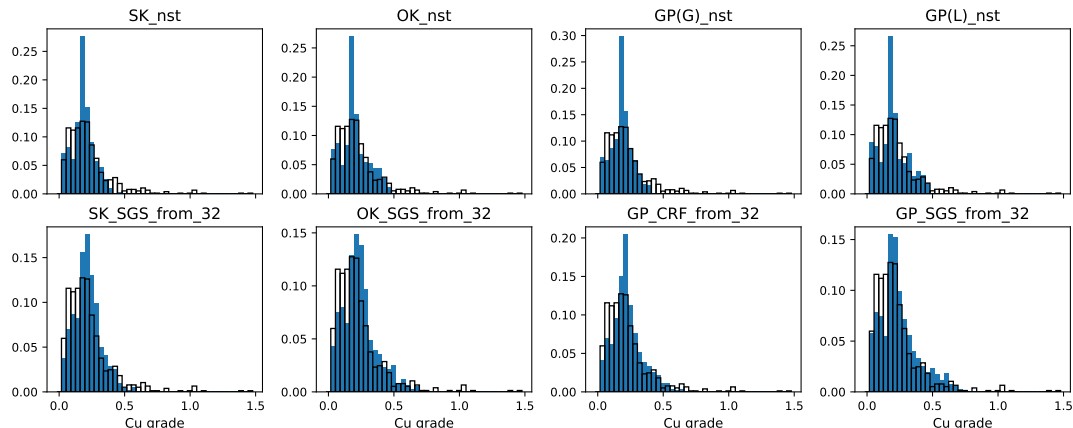

**Figure 16.** Copper grade histograms for $g_D = 3521$ and $m_A = 4$. Black hollow: groundtruth. Blue: model predictions





**Table 4.** Histogram distances for mean model predictions relative to groundtruth

|  | Domain 2310 | | | | | Domain 3521 | | | | |
| Model | $h_{\text{psChi}}$ | $h_{\text{JS}}$ | $h_{\text{Ruz}}$ | $h_{\text{EM}}$ | rank* | $h_{\text{psChi}}$ | $h_{\text{JS}}$ | $h_{\text{Ruz}}$ | $h_{\text{EM}}$ | rank* |
|---|---|---|---|---|---|---|---|---|---|---|
| SK_nst | 0.5594 | 0.1225 | 0.4422 | 0.0414 | 27 | 0.3723 | 0.0791 | 0.3480 | 0.0502 | 30 |
| OK_nst | 0.2071 | 0.0422 | 0.2792 | 0.0199 | 15 | 0.3368 | 0.0702 | 0.3656 | 0.0345 | 29 |
| GP(L)_nst | 0.1273 | 0.0266 | 0.2261 | 0.0137 | 8 | 0.3335 | 0.0695 | 0.3690 | 0.0335 | 27 |
| GP(G)_nst | 0.3303 | 0.0693 | 0.3578 | 0.0331 | 24 | 0.3900 | 0.0829 | 0.3480 | 0.0522 | 31 |
| SK-SGS (from 32) | 0.8774 | 0.1845 | 0.5629 | 0.0535 | 30 | 0.2198 | 0.0458 | 0.3210 | 0.0408 | 24 |
| OK-SGS (from 32) | 0.3119 | 0.0650 | 0.3356 | 0.0302 | 22 | 0.1929 | 0.0380 | 0.2949 | 0.0354 | 17 |
| GP-SGS (from 32) | 0.0948 | 0.0192 | 0.1851 | 0.0182 | 6 | 0.1902 | 0.0376 | 0.2835 | 0.0357 | 16 |
| GP-CRF (from 32) | 0.2763 | 0.0580 | 0.3194 | 0.0311 | 21 | 0.1876 | 0.0391 | 0.2854 | 0.0344 | 14 |

\* Ranks are not consecutive as distances computed for $\{\text{SGS/CRF (from } s)\}_{s=2,4,8,16,64}$ have been omitted for clarity.

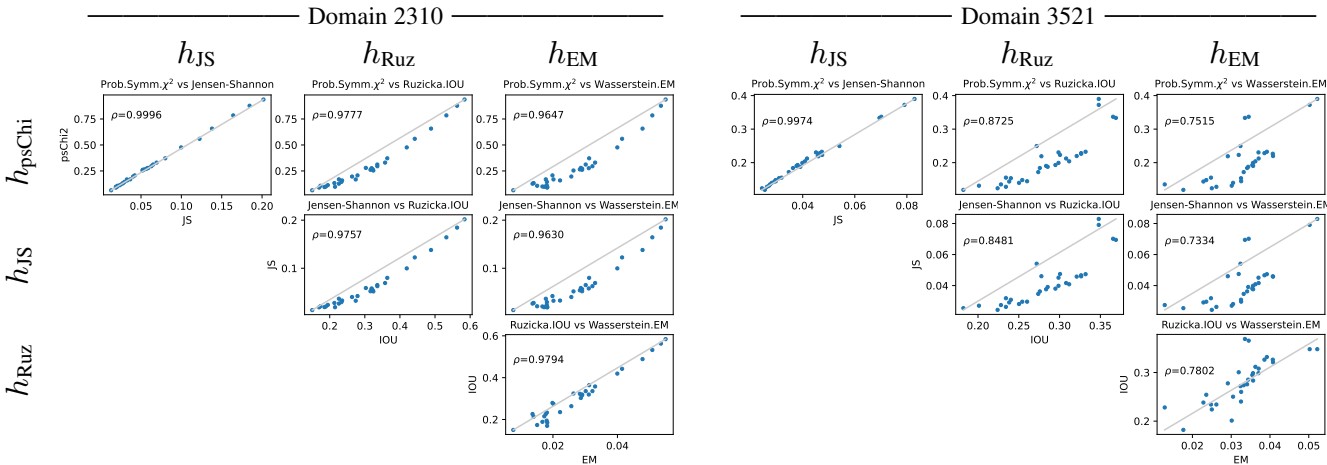

**Figure 17.** Histogram distances cross-plots for $g_D = 2310$ and $g_D = 3521$ in $m_A = 4$

### 5.1.2 Variograms

Variograms are presented separately for models in the SK, OK, GP(L) and GP(G) family along with their SGS/CRF counterparts in Fig. 18. To be clear, the north-west, north-east, south-west and south-east quadrants in each half represent the SK/SK-SGS, OK/OK-SGS, GP(G)/GP-CRF and GP(L)/GP-SGS families, respectively. Within a given domain, the variogram plots can be compared directly between families since the scales are the same. Two reference curves—black-solid for the *groundtruth* $\{\boldsymbol{x}_{*j}, y_{*j}\}_{j=1:m}$ and black-dashed for *blasthole* training data $\{\boldsymbol{x}_i, y_i\}_{i=1:n}$—are included to indicate the range of

spatial variability the models should strive to achieve. Additionally, a grey curve representing a black-box long-range prediction model and a lilac curve representing GP(L) are included in each plot to provide a benchmark.





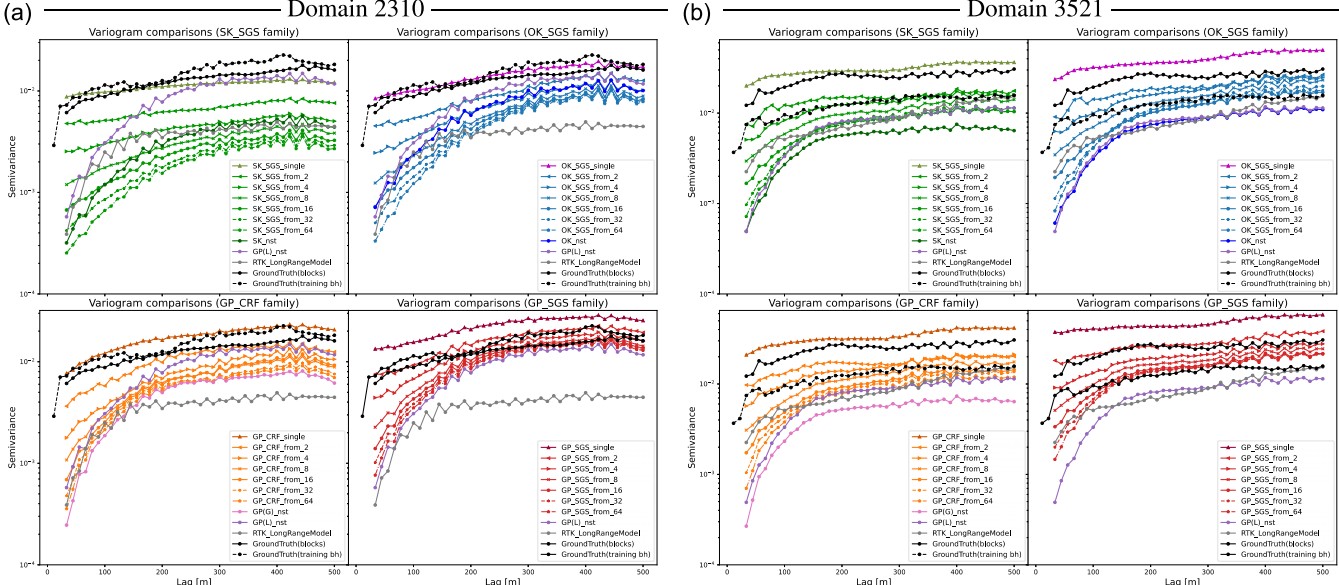

**Figure 18.** Copper grade variograms for $g_D = 2310$ (left) and $g_D = 3521$ (right). The quadrants group together models in the following families: (north-west) SK/SK-SGS, (north-east) OK/OK-SGS, (south-west) GP(G)/GP-CRF, (south-east) GP(L)/GP-SGS.

Focusing on domain 2310 first, the north-west quadrant shows that simple kriging is not competitive with GP(L), in fact, SK-SGS generally performs far worse than the long-range model which itself is inferior to all other models. In the south-west quadrant, the pink curve representing GP(G) outperforms the long-range model. Henceforth, we use the notation $\gamma_\aleph$ to denote

the variogram for a model/reference $\aleph$. With sequential Gaussian simulation, the average variogram for a single realisation (orange ▲) matches $\gamma_{\text{blastholes}}$ and from two realisations, $\gamma_{\text{GP-CRF from 2}}$ (orange ◄) matches $\gamma_{\text{groundtruth}}$ for the most part. As the number of simulations ($s$) increases, $\gamma_{\text{GP-CRF from } s}$ becomes smoother and approaches $\gamma_{\text{GP(G)}}$. The patterns for ordinary kriging are very similar, except $\gamma_{\text{OK-SGS from } s} < \gamma_{\text{GP-CRF from } s}$ for smaller lags.

### 5.1.3 Insights from the variograms

In the north-east quadrant of Fig. 18(a), one observes the blue curves representing $\gamma_{\text{OK-SGS from } s}$ are generally above $\gamma_{\text{long-range}}$ but, more importantly, underneath the lilac curve representing $\gamma_{\text{GP(L)}}$. This indicates GP(L) has the highest spatial fidelity among the base candidates. In the south-east quadrant, it can be seen that sequential simulations can further propel $\gamma_{\text{GP-SGS}}$ above $\gamma_{GP(L)}$. This finding is highly significant. It shows that while GP(L) can handle mean regression for a non-stationary process through the use of local neighbourhoods, the local variance estimates based on these neighbourhoods do not adequately

capture the covariance (full inter-sample dependencies) of the underlying process. The loss, in terms of longer-range spatial correlation, can be replenished through sequential Gaussian simulation (see Sec. 2.3 and chain rule in (13)) which effectively propagates conditional information through random paths.





**Table 5.** Variogram ratios ($R$) and spatial fidelity ($F$) statistics for $g_D = 2310$ and $g_D = 3521$ in $m_A = 4$.

|  | Domain 2310 | | Domain 3521 | |
|---|---|---|---|---|
|  | $R$ | $F$ | $R$ | $F$ |
| SK_nst | 0.2775 | 0.5267 | 0.2328 | 0.4824 |
| OK_nst | 0.5945 | 0.7710 | 0.3311 | 0.5754 |
| GP(L)_nst | 0.7568 | 0.8699 | 0.3451 | 0.5874 |
| GP(G)_nst | 0.4525 | 0.6726 | 0.2217 | 0.4708 |
| SK-SGS (from 4) | 0.3415 | 0.5843 | 0.5171 | 0.7190 |
| OK-SGS (from 4) | 0.5894 | 0.7677 | 0.6882 | 0.8295 |
| GP-SGS (from 4) | 1.0774 | 0.9605 | 0.8046 | 0.8969 |
| GP-CRF (from 4) | 0.6391 | 0.7994 | 0.6105 | 0.7813 |
| SK-SGS (from 32) | 0.1736 | 0.3998 | 0.3275 | 0.5722 |
| OK-SGS (from 32) | 0.4565 | 0.6756 | 0.4705 | 0.6859 |
| GP-SGS (from 32) | 0.8481 | 0.9209 | 0.5964 | 0.7722 |
| GP-CRF (from 32) | 0.4929 | 0.7020 | 0.3941 | 0.6277 |

The lessons are similar for domain 3521 in Fig. 18(b), except GP(L) on its own is close to but not necessarily better than the long-range model. We believe this is due to $\gamma_{\text{blastholes}} < \gamma_{\text{groundtruth}}$, viz., the training data is smoother than the groundtruth. This perhaps makes the goal of matching the spatial variability in the groundtruth unattainable and the task of future-bench prediction more difficult. The key observation is that SGS is needed to elevate the performance of OK and GP into the target band encompassed by $\gamma_{\text{blastholes}}$ and $\gamma_{\text{groundtruth}}$. A related observation is that GP-SGS pushes the curves higher than OK-SGS.

### 5.1.4 Practicality

It takes considerable effort to visually compare variograms even for eight models in a single domain. This becomes cumbersome and error-prone when there are over a hundred ($g_D$, $m_A$) combinations to assess, as is the case in our later experiments. The variogram ratio ($R$) and spatial fidelity ($F$) statistics formulated in Sec. 3.2 provide a practical measure of model quality, taking into account local spatial correlation in the regression results. These statistics are reported in Table 6. The key finding is that the spatial fidelity of the base models can be boosted with limited rounds of sequential simulation. As a case in point, consider domain 3521 in Table 6. With four rounds of SGS, the $F$ value has increased from 0.575 to 0.829, and 0.587 to 0.896, for OK and GP(L), respectively. Although spatial fidelity drops off with further rounds of simulations (as illustrated graphically in Table 6), the benefit is sustained for GP-SGS and GP-CRF even after 32 iterations—$F$ remain at 0.772 and 0.627, respectively, which are substantially higher than 0.575 and 0.587.




### 5.1.5 Prediction accuracy and uncertainty intervals

This section examines the accuracy and interval of the predictive distributions both qualitatively and quantitatively. In domains
2310 and 3521, we observed that all model curves have similar shapes in the accuracy and interval plots, $\bar{\kappa}(p)$ and $\bar{W}(p)/\sigma_Y$.
Therefore, it suffices to illustrate the general behaviour through one model family, viz. ordinary kriging and OK-SGS. Results
depicted in Fig. 19 are typical of all models within the respective domains. The main finding that can be distilled from these
plots is that the models are very accurate in domain 2310, as evident from its high precision (closeness between the observed
and expected proportions, $\bar{\kappa}(p)$ and $p$). Based on the interpretations of Sec. 3.4, the fact that $\bar{\kappa}(p) > p$ for much of $p$ (note: $p$
denotes the expected groundtruth capture probability) which is reflected by lower precision ($P$) and goodness ($G$) suggests the
models are conservative in domain 3521.

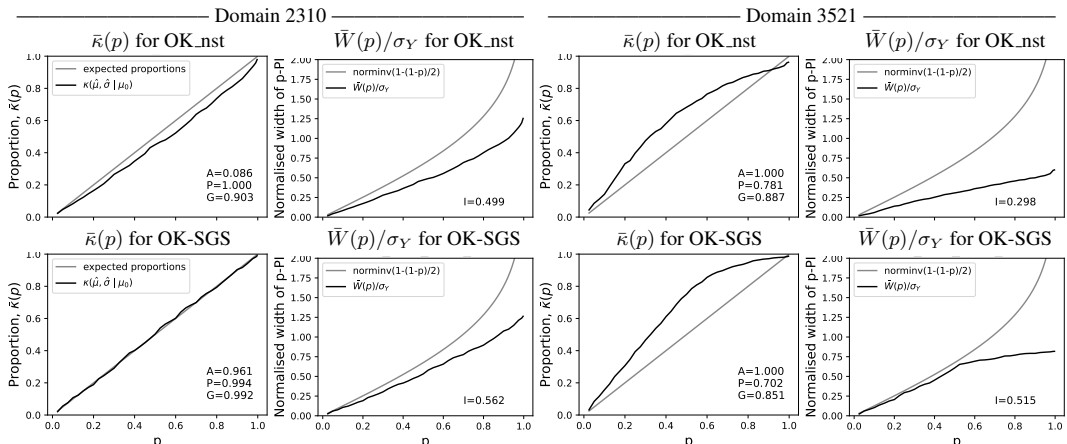

**Figure 19.** Copper grade predictive distribution accuracy and uncertainty ++interval plots. Selected results for $g_D = 2310$ and $g_D = 3521$ in $m_A = 4$.

A detailed reading of the statistics shown in Table 6 show that the results are mixed and no dominant model can be established
between these two domains. Regarding the individual indicators, the following comments can be made. The accuracy, $A_\xi$, is
quite sensitive even when a slack variable $\xi = .05$ is employed—confer with (28). It can change rapidly from 0 to 1, see
domain 2310 OK_nst vs GP(G)_nst, due to the hard constraint it imposes on the observed-vs-expected proportion comparison.
As the number of simulation runs increases, the precision and goodness statistics ($P$ and $G$) both get slightly worse due to
stochasticity, this pattern runs contrary to the likelihood trend ($L$). Philosophically, the notion of $p$-probability intervals, which
is built on the notion of groundtruth capture and what a model promises and actually delivers, might not be a great criterion to
judge models on. On current evidence, these $p$-PI statistics seem to be quite limited in their capacity at differentiating models.
However, we reserve final judgement on their utility as the scope of our analysis is quite limited at this point. This issue will
be revisited in part two of our analysis (Sec. 5.2).





**Table 6.** Uncertainty-based statistics for $g_D = 2310$ and $g_D = 3521$ in $m_A = 4$. Specifically, $L$, $A$, $P$, $G$ and $I$ denote the likelihood, accuracy, precision, goodness and interval tightness of the probabilistic predictions.

| | Domain 2310 | | | | | Domain 3521 | | | | |
| | $L$ | $A_{.05}$ | $P$ | $G$ | $I$ | $L$ | $A_{.05}$ | $P$ | $G$ | $I$ |
|---|---|---|---|---|---|---|---|---|---|---|
| SK_nst | 0.4817 | 0.6172 | 1.0000 | 0.9631 | 0.5182 | 0.6231 | 1.0000 | 0.7480 | 0.8686 | 0.2918 |
| OK_nst | 0.4517 | 0.0859 | 0.9999 | 0.9030 | 0.4992 | 0.6075 | 1.0000 | 0.7808 | 0.8866 | 0.2984 |
| GP(L)_nst | 0.4963 | 0.9062 | 0.9953 | 0.9857 | 0.5752 | 0.6353 | 1.0000 | 0.7272 | 0.8618 | 0.3071 |
| GP(G)_nst | 0.5184 | 1.0000 | 0.9628 | 0.9812 | 0.5548 | 0.6294 | 1.0000 | 0.7361 | 0.8634 | 0.2965 |
| SK-SGS (from 32) | 0.4967 | 0.9297 | 0.9987 | 0.9913 | 0.5652 | 0.6506 | 1.0000 | 0.6978 | 0.8484 | 0.4246 |
| OK-SGS (from 32) | 0.5010 | 0.9609 | 0.9936 | 0.9925 | 0.5617 | 0.6485 | 1.0000 | 0.7022 | 0.8509 | 0.5152 |
| GP-SGS (from 32) | 0.5071 | 0.7383 | 0.9753 | 0.9771 | 0.6531 | 0.6411 | 1.0000 | 0.7172 | 0.8585 | 0.5311 |
| GP-CRF (from 32) | 0.5265 | 0.9961 | 0.9462 | 0.9723 | 0.6208 | 0.6255 | 0.9727 | 0.7477 | 0.8730 | 0.4779 |
| SK-SGS (from 128) | 0.5116 | 0.9805 | 0.9754 | 0.9863 | 0.5659 | 0.6728 | 1.0000 | 0.6535 | 0.8266 | 0.4273 |
| OK-SGS (from 128) | 0.5142 | 0.9688 | 0.9707 | 0.9844 | 0.5661 | 0.6698 | 1.0000 | 0.6598 | 0.8298 | 0.5187 |
| GP-SGS (from 128) | 0.5232 | 0.9609 | 0.9530 | 0.9759 | 0.6475 | 0.6650 | 0.9961 | 0.6693 | 0.8346 | 0.5370 |
| GP-CRF (from 128) | 0.5434 | 1.0000 | 0.9127 | 0.9563 | 0.6085 | 0.6592 | 0.9961 | 0.6810 | 0.8404 | 0.4491 |

Looking at the base models in Table 6, what is clear is that GP models produce higher likelihood scores ($L$) than kriging models in both domains. For example, the likelihood scores for GP(L) and GP(G) [$L = 0.496, 0.518$] are higher than those for SK and OK [$L = 0.481, 0.451$] in domain 2310; and the same can be said for domain 3521. The likelihood scores also show

SGS/CRF improve prediction performance and this effect increases with more simulation runs. Significantly, this improvement is geared toward bringing the predictions closer to the groundtruth ($\mu_0$), rather than mere containment of the groundtruth within a prediction interval (Fouedjio and Klump, 2019). This distance-based interpretation follows from (24)–(25) where likelihood $L$ is defined in terms of synchronicity $s(\hat{\mu}, \hat{\sigma} \,|\, \mu_0)$ which is driven by Z scores, $z = (\mu_0 - \hat{\mu})/\hat{\sigma}$.

### 5.1.6 Synchronicity as a visualisation tool

It is worth highlighting the potential of the synchronicity measure, $S \stackrel{\Delta}{=} s(\hat{\mu}, \hat{\sigma} \,|\, \mu_0)$, for model evaluation from a spatial perspective. By construction, $S > 0$ (resp. $S < 0$) when the predicted mean underestimates (resp. overestimates) the true grade. These instances are rendered in red (resp. blue) in Figs. 20 and 21. Larger deviations from the groundtruth are indicated by a darker shade. Following this convention, these figures can serve effectively as local distortion maps. Specifically, the blue cluster in the northwest corner in Fig. 20 show areas where the OK model had under-performed by way of overestimating the

Cu grade. The relative strength of the GP(G) model is highlighted by lighter patches at the corresponding location. Moving over to the label marked 'A' in the east, GP(L) can be seen to provide a better estimation than GP(G) whereby the intensity of the red patches (underestimation) is reduced. There is perhaps no where more obvious than in Fig. 21, where the problem of underestimation at the northwestern tip is conspicuous in all base models and the magnitude of the prediction error is significantly reduced through SGS/CRF simulation.





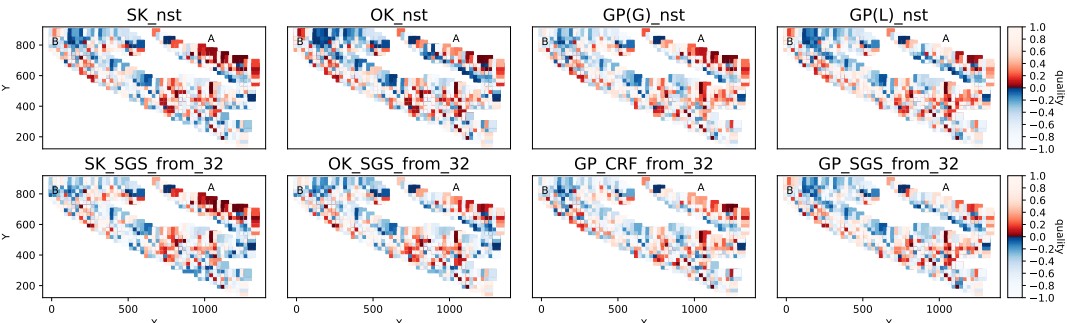

**Figure 20.** Synchronicity of grade predictions w.r.t. the groundtruth for $g_D = 2310$ and $m_A = 4$.

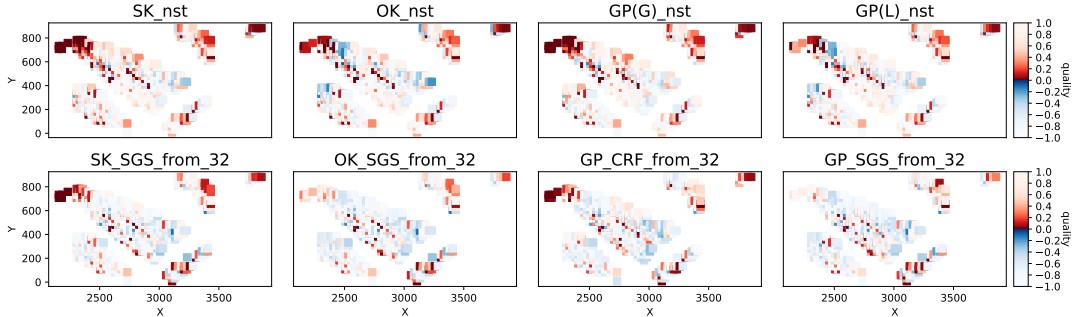

**Figure 21.** Synchronicity of grade predictions w.r.t. the groundtruth for $g_D = 3521$ and $m_A = 4$.

## 5.2 Analysis 2: Performance of probabilistic models across all domains and inference periods

Consolidating on the preliminary analysis, this section now looks at the broader picture across all inference periods and domains. This is prompted by a desire to minimise selection bias and determine the stability of models under varying conditions. A key motivation is to obtain statistically significant results so that findings arising from random chance can be effectively ruled out. Readers can expect to see qualitative and quantitative analysis on future-bench prediction performance, including a statistical comparison with in-situ regression in Sec. 5.3. The chief strategy advocated in this paper is to view various statistics from an image perspective, whereby models and conditions (inference period and domain) are represented by the vertical and horizontal axes, respectively. This takes inspiration from microplates (Piletska et al., 2012), a standard screening tool used in clinical diagnostic testing such as enzyme-linked immunosorbent assay (ELISA), whereby antigen-antibody interactions are detected within a 2D array. This has been used in biochemistry to study enzyme diversity in soils (Marx et al., 2001). A similar setup that exploits this attention mechanism is equally well suited to large-scale simultaneous comparisons in geostatistics.





**Table 7.** Histogram distance summary statistics for future-bench prediction over domains and inference periods.

| Model family | Abbrev | $h_{JS}$ mean | $h_{EM}$ mean |
|---|---|---|---|
| Simple kriging | SK / SK-SGS | 0.4607 | 0.1678 |
| Ordinary kriging | OK / OK-SGS | 0.3524 | 0.1126 |
| Gaussian process (global mean) | GP(G) / GP-CRF | 0.2937 | 0.0753 |
| Gaussian process (local mean) | GP(L) / GP-SGS | 0.2802 | 0.0691 |

### 5.2.1 Histogram distances

As an example, the Jensen-Shannon and Wasserstein histogram distances, $h_{JS}$ and $h_{EM}$, are visualised as images in the left and right halves of Fig. 22, respectively. In these arrays, the rows represent models which are grouped along family lines into four categories: SK, OK, GP(G) and GP(L). The columns represent geological domains (see outer x labels). Furthermore, successive inference periods ($m_A$) are interleaved within each domain (see inner x labels). Looking at $h_{JS}$, these results may be interpreted in two ways. At a macro-level, the GP(G) and GP(L) families, represented by the third and fourth blocks down the y axis, appear much darker than the rest. These indicate lower distortion in the predicted grade histograms relative to the groundtruth. The relevant group statistics are summarised in Table 7. At a granular level, differences between row 0 (SK) and row 1 (SK_nst) illustrate the importance of normal score transformation in simple kriging. Focusing on higher level trends, more pervasive distortion can be seen in the SK and OK families, as evident from the bright pixels in the first and second block.

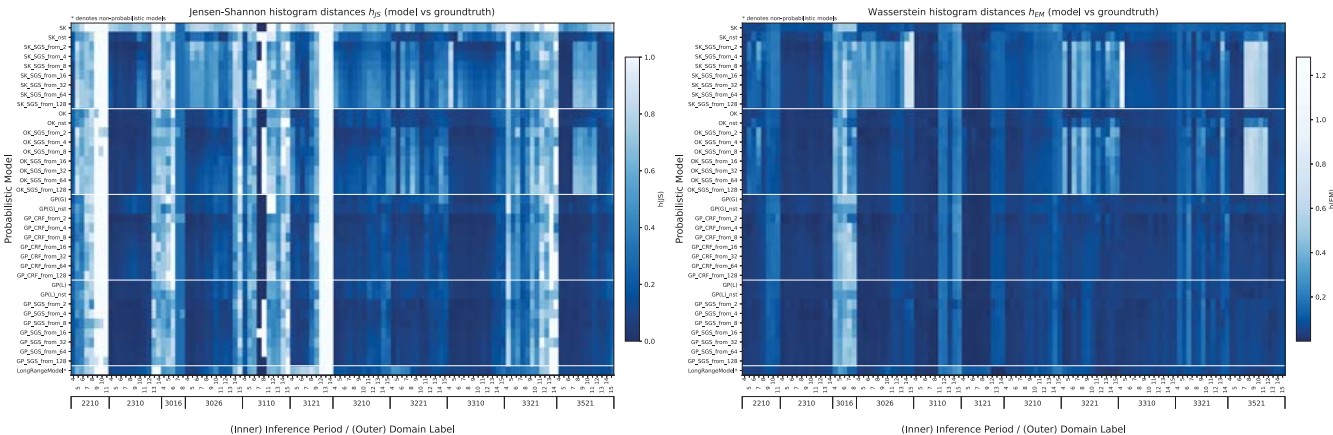

**Figure 22.** View of (left) Jensen-Shannon, (right) EM histogram distances for future-bench prediction across domains and inference periods.

### 5.2.2 Influential factors

An investigation of the bright pixel columns in Fig. 22(left)—instances where SK and OK apparently underperformed—reveals two contributing factors. The first is that histogram distance measures (not just $h_{JS}$, but also $h_{psChi}$ and $h_{Ruz}$) are sensitive to





**Table 8.** Sample size statistics for certain domains ($g_D$) and inference periods ($m_A$) involved in future-bench prediction.

| $g_D$ | $m_A$ | $n_T$ | $n_I$ | $g_D$ | $m_A$ | $n_T$ | $n_I$ | $g_D$ | $m_A$ | $n_T$ | $n_I$ | $g_D$ | $m_A$ | $n_T$ | $n_I$ | $g_D$ | $m_A$ | $n_T$ | $n_I$ |
|---|---|---|---|---|---|---|---|---|---|---|---|---|---|---|---|---|---|---|---|
| 2210 | 4 | 76 | 66 | | 10 | 123 | 5 | | 6 | 86 | 10 | | 15 | 2120 | 10 | | 9 | 208 | 28 |
| | 5 | 66 | 30 | | 11 | 123 | 4 | | 7 | 86 | 2 | 3321 | 4 | 14 | 49 | | 10 | 224 | 36 |
| | 6 | 114 | 37 | 2310 | 13 | 2653 | 19 | | 8 | 86 | 2 | | 5 | 61 | 101 | | 11 | 227 | 12 |
| | 7 | 109 | 16 | | 14 | 2654 | 19 | 3026 | 12 | 2050 | 111 | | 6 | 59 | 104 | | 12 | 242 | 16 |
| | 8 | 130 | 15 | 3016 | 4 | 67 | 28 | | 13 | 2088 | 75 | | 7 | 143 | 108 | | 13 | 238 | 4 |
| | 9 | 121 | 9 | | 5 | 81 | 8 | | 14 | 2113 | 35 | | 8 | 195 | 68 | | 14 | 242 | 4 |

$n_T$ and $n_I$ denote number of training and inference points

discretisation and number of inference points ($n_I$) used in a given groundtruth comparison; this is not a modelling artefact. The second is a divergence between the training data and groundtruth distributions. By way of an example, the first phenomenon is evident from the white patches that appear in the 2210 columns in Fig. 22(left) and this coincides with $n_I \leq 16$ from $m_A = 7$ to $m_A = 11$ in Table 8. This indicates a drop in the efficacy of $h_{JS}$ as the sample size decreases. For domain 2310, the number of inference points is similarly small for $m_A = 13$ and 14; we see a similar drop-off as $h_{JS}$ becomes unreliable. On the contrary, $h_{EM}$ (see corresponding columns in Fig. 22(right)) is quite insensitive to sample size.

For domain 3016, the number of inference points is once again very small (mostly $n_I \leq 10$ in Table 8). However, $h_{JS}$ and $h_{EM}$ are both large; this indicates the degradation in performance is genuine. Looking at the distribution of the training data in Fig. S.6 (see supplementary material), we hypothesize that this is due to the spread (almost uniform distribution) observed in this domain. Prediction is more difficult when the entropy of the measured data is high. This may indicate volatility in the grade distribution (an intrinsic property in certain parts of the deposit) or incorrect domaining (epistemic uncertainty attributable to data sparseness and boundary uncertainty).

Domain 3026 is afflicted by the same issues (discretisation and few inference points) as domain 2210. The bad behaviour observed in $m_A = 14$ and 15 correlate directly with sample size in Table 8. The most striking results for $h_{JS}$ occur in domains 3110, 3121 and 3321. For domain 3110, a slight performance drop-off is observed in $m_A = 11$ and beyond. Examination of the training data distribution and groundtruth grade distribution in Fig. S.6 reveals fundamental differences between the two. The higher grade values present in the groundtruth were beyond anything seen in the training data, thus they are quite unexpected and hard to predict. For domain 3121, the significant elevation in $h_{JS}$ from $m_A = 12$ to $m_A = 14$ is due to the propensity of samples lying outside the grade range observed in the training data. From Fig. S.6, it can be confirmed the training data and groundtruth distributions hardly intersect; hence their JSD similarity is only 23.9%.

For domains 3210, 3221, 3310 and 3521, there is general consensus between $h_{JS}$ and $h_{EM}$. The GP models all performed well with respect to both measures. For domain 3321, the moderately elevated $h_{JS}$ from $m_A = 4$ to $m_A = 6$ is due to the small number of training samples available (see Table 8) and slight mismatch between the training data and groundtruth distributions. The much elevated $h_{JS}$ from $m_A = 9$ and beyond is due to the small number of inference points and sensitivity to histogram discretisation. For domain 3521, a persistent cluster of poor performance is observed for kriging models from $m_A = 7$ to $m_A = 11$.

 

These instances were found to occur when there is a highly positive-skewed, long-tail groundtruth distribution coinciding with
a more narrow training data distribution with a more left-leaning mode; this is illustrated in Fig. S.6. This issue also affects
domain 3221 to some extent. These technically more challenging situations for kriging-based models can be seen clearly and
unambiguously as light colour patches in the Wasserstein image in Fig. 22(right). This makes it an imperative to include $h_{\mathrm{EM}}$
in global accuracy assessment if the confounding effects due to sample size or discretisation are to be suppressed.

### 5.2.3 Spatial fidelity

The same techniques are used to examine variogram ratios and spatial fidelity across all periods and domains. What is different
about Fig. 23 is the appearance of dotted cells $\boxed{\bullet}$. These represent instances where a variogram cannot be reliably computed
(when the number of inference points $n_{\mathrm{I}} < 30$) and are used to avoid confusion with bright pixels (more extreme ratios) which
are undesirable. In Fig. 23(left), variogram ratios in the range $[0,1)$ and $[1,2]$ are rendered in red and purple, respectively, with
the colour intensity transitioning from light to dark as the ratios get closer to 1 which is the ideal.

The results in Fig. 23(left) reinforces the findings described in Sec. 5.1.2 in two important respects. First, SGS/CRF simula-
tion increases the variogram ratios across all domains and inference periods, irrespective of the model family: SK, OK, GP(G)
or GP(L). The effect is strongest when $s = 2$ and decreases with further simulations ($s$). However, the benefits are sustained
the longest in GP(L); notably, GP-SGS($s = 32$) has higher spatial fidelity than the GP(L)_nst base model. Second, the GP(L)
family achieves the highest spatial fidelity among all model candidates, especially when combined with SGS. This can be seen
from Fig. 23(right) where the pixels in the lowest block (corresponding to the GP(L) family) are on average the darkest ($F$
being the closest to 1).[3] This conclusion is supported by the summary statistics in Table 9, where the standard error (SE) further
demonstrates the SK, OK modelling results are more variable.

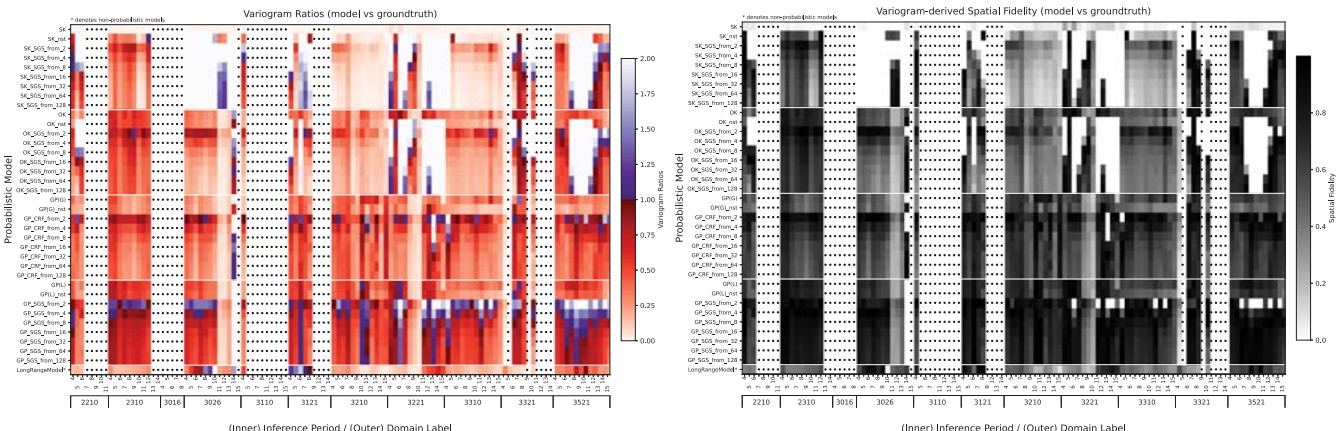

**Figure 23.** View of (left) variogram ratios $R$ and (right) spatial fidelity $F$ for future-bench prediction across domains and inference periods.

---

[3]Apart from $s = 2$ where sequential simulation injects more random fluctuations into the signal before it settles.




**Table 9.** Variogram ratio ($R$) and spatial fidelity ($F$) summary statistics for future-bench prediction over domains and inference periods.

| Model family | Abbrev | $R$ mean (SE) | $F$ mean (SE) |
|---|---|---|---|
| Simple kriging | SK / SK-SGS | 0.2983 (0.0112) | 0.4272 (0.0125) |
| Ordinary kriging | OK / OK-SGS | 0.4787 (0.0102) | 0.6234 (0.0110) |
| Gaussian process (global mean) | GP(G) / GP-CRF | 0.6114 (0.0070) | 0.7675 (0.0055) |
| Gaussian process (local mean) | GP(L) / GP-SGS | 0.7132 (0.0081) | 0.8231 (0.0069) |

### 5.2.4 Accuracy and precision

Turning attention now to uncertainty-based measures, this section examines the accuracy and precision of the predictive distributions across all periods and domains. To keep this brief, we restrict our comments to peculiar cases and general trends. Accuracy is depicted in Fig. 24(left). It turns out that Deutsch's notion of accuracy conveys something important about SGS/CRF. Recall that accuracy relates to groundtruth capture by $p$-probability intervals (see Sec. 3.3.2). It considers what a model promises and actually delivers with respect to the proportion of samples it expects to cover. The prominent white horizontal strips in Fig. 24(left) show that the SGS/CRF models fail to live up to this expectation when $s=2$ and $s=4$. Within the context of this study, at least $s=8$ simulations is required to obtain an accurate probabilistic model for future-bench grade prediction in a porphyry copper deposit. Next, moving on to lesser issues, the vertical strips in domain 2210 (from $m_A=8$ to $m_A=11$) and domain 3121 (from $m_A=12$ to $m_A=14$) coincide with few inference points according to Table 8. In the case of 3121, the relevant periods each had only two samples. For the precision image in Fig. 24(right), the lightly coloured columns in domain 3110, $m_A \in \{7,8\}$, are similarly explained by virtue of having only one test sample. The summary statistics in Table 10 show the GP(L) and GP(G) families achieve the highest overall accuracy across all domains and inference periods while precision is similar across all families (between 0.851 and 0.868).

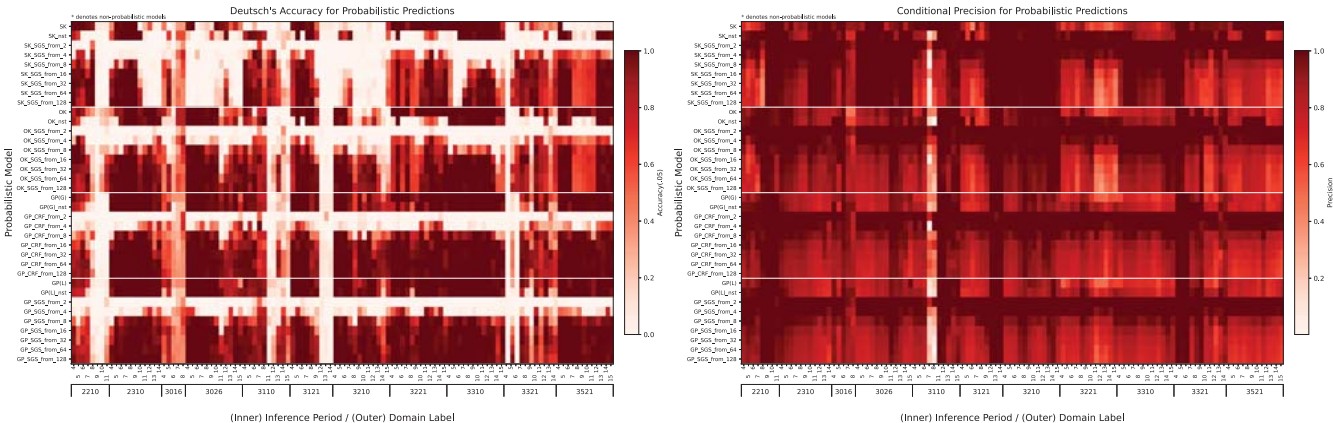

**Figure 24.** View of (left) accuracy $A$ and (right) precision $P$ for future-bench prediction across domains and inference periods.





**Table 10.** Accuracy ($A$) and precision ($P$) summary statistics for future-bench prediction over domains and inference periods.

| Model family | Abbrev | $A$ mean (SE) | $P$ mean (SE) |
|---|---|---|---|
| Simple kriging | SK / SK-SGS | 0.5245 (0.0159) | 0.8680 (0.0058) |
| Ordinary kriging | OK / OK-SGS | 0.7173 (0.0133) | 0.8672 (0.0050) |
| Gaussian process (global mean) | GP(G) / GP-CRF | 0.8084 (0.0119) | 0.8637 (0.0041) |
| Gaussian process (local mean) | GP(L) / GP-SGS | 0.8127 (0.0117) | 0.8510 (0.0048) |

\* Group averages exclude SGS/CRF $s = 2$ and $s = 4$, viz., epochs long before convergence.

### 5.2.5 Likelihood and goodness

The likelihood ($L$) and goodness ($G$) of the predictive distributions across all periods and domains are shown in Fig. 25. An immediate observation is that $L$ and $G$ are generally correlated. Looking at Fig. 25(right), aside from the statistics being more variable for the kriging base models (see first two rows in the SK and OK block), the image is quite unremarkable. Looking at the group statistics in Table 11, the likelihood statistic suggests GP(L) is best and GPs are to be preferred (with $L_{\text{GP(L)}} = 0.543$ and $L_{\text{GP(G)}} = 0.536$) over ordinary kriging (with $L_{\text{OK}} = 0.513$ and $L_{\text{SK}} = 0.452$). The message from the goodness statistic is similar but subtly different, it places GP(L) and GP(G) as equal (with $G_{\text{GP(L)}} = 0.799$ and $G_{\text{GP(L)}} = 0.797$) and ordinary kriging as a close alternative (with $G_{\text{OK}} = 0.786$).

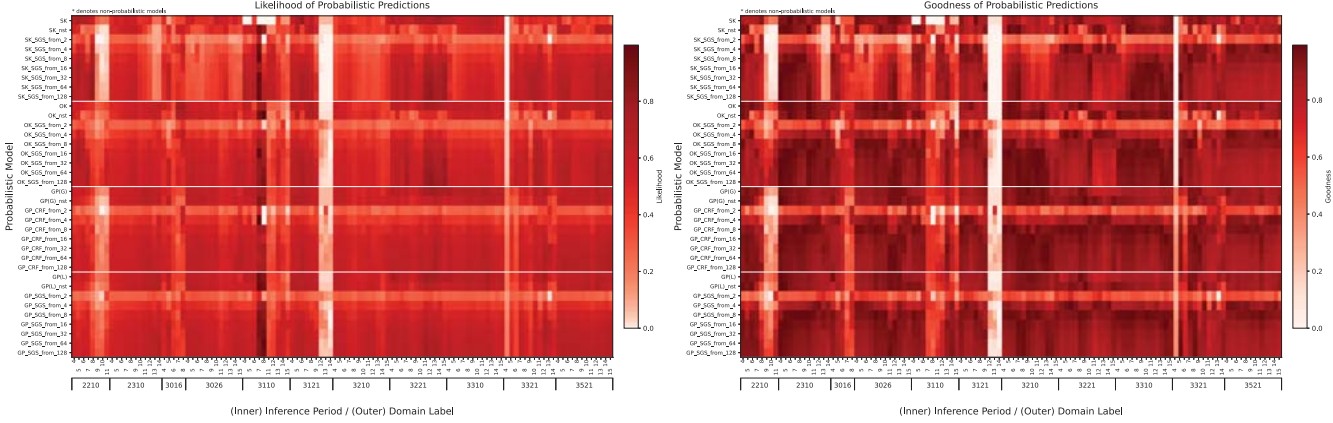

**Figure 25.** View of (left) likelihood $L$ and (right) goodness $G$ for future-bench prediction across domains and inference periods.

### 5.2.6 Interval tightness

The interval tightness ($I$) of the predictive distributions across all periods and domains are shown in Fig. 26. From the extensive white-out regions, where the width of the prediction interval is large, one can reasonably infer that simple kriging produces



**Table 11.** Likelihood ($L$) and goodness ($G$) summary statistics for future-bench prediction over domains and inference periods.

| Model family | Abbrev | $L$ median | $[q_L, q_U]$ | $G$ mean (SE) |
|---|---|---|---|---|
| Simple kriging | SK / SK-SGS | 0.4527 | [0.2346, 0.6219] | 0.7149 (0.0076) |
| Ordinary kriging | OK / OK-SGS | 0.5137 | [0.2795, 0.6764] | 0.7868 (0.0065) |
| Gaussian process (global mean) | GP(G) / GP-CRF | 0.5366 | [0.2855, 0.6990] | 0.7974 (0.0062) |
| Gaussian process (local mean) | GP(L) / GP-SGS | 0.5432 | [0.2996, 0.7053] | 0.7997 (0.0059) |

**Table 12.** Interval tightness ($I$) summary statistics for future-bench prediction over domains and inference periods.

| Model family | Abbrev | $I$ mean (SE) |
|---|---|---|
| Simple kriging | SK / SK-SGS | 0.6898 (0.0096) |
| Ordinary kriging | OK / OK-SGS | 0.6408 (0.0087) |
| Gaussian process (global mean) | GP(G) / GP-CRF | 0.5787 (0.0072) |
| Gaussian process (local mean) | GP(L) / GP-SGS | 0.6180 (0.0073) |

the least confident (most uncertain) predictions. This can be confirmed from the group statistics in Table 12 which also show GP(G) produces the narrowest predictions.

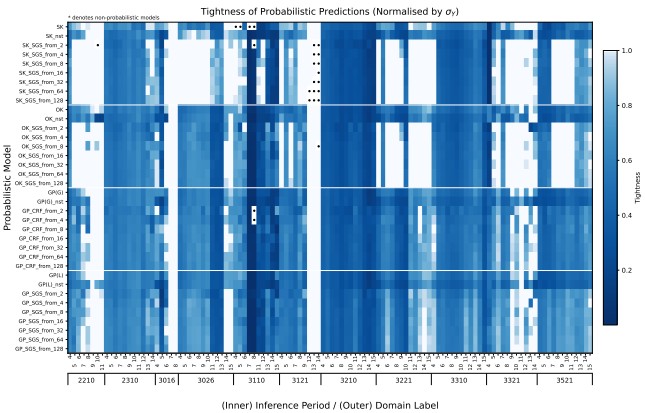

**Figure 26.** View of interval tightness $I$ for future-bench prediction across domains and inference periods.

### 5.2.7 Statistical significance

The dependent t-test is applied to the histogram, fidelity, accuracy, precision, interval tightness, goodness and likelihood scores ($H$, $F$, $A$, $P$, $T$, $G$ and $L$) to establish the significance of the results. In general, the null hypothesis asserts that the mean score for model family $\psi$ (where $\psi \in \{$SK, OK, GP(G)$\}$) is greater than or equal to the mean for the GP(L) family. Thus, the null and alternative hypotheses may be written as $H_0(X, \psi) : \mu_X^\psi \geq \mu_X^{\text{GP(L)}}$ and $H_a(X, \psi) : \mu_X^\psi < \mu_X^{\text{GP(L)}}$. When applied to scores that





**Table 13.** Significance testing of statistical scores for future-bench prediction over all domains and inference periods.

| Family $\psi$ | Histogram $H = h_{\text{EM}}$ | | Spatial Fidelity $F$ | | Accuracy $A$ | | Precision $P$ | |
|---|---|---|---|---|---|---|---|---|
| | $p$ | CI | $p$ | CI | $p$ | CI | $p$ | CI |
| SK/SGS | $< .001$ | $[0.1670, 0.1940]$ | $< .001$ | $[-0.4221, -0.3696]$ | $< .001$ | $[-0.3212, -0.2551]$ | $> 0.99$ | $[0.0221, 0.0396]$ |
| OK/SGS | $< .001$ | $[0.0627, 0.0816]$ | $< .001$ | $[-0.2241, -0.1753]$ | $< .001$ | $[-0.1142, -0.0764]$ | $> 0.99$ | $[0.0128, 0.0247]$ |
| GP(G)/CRF | $< .001$ | $[0.0058, 0.0211]$ | $< .001$ | $[-0.0683, -0.0429]$ | $0.1685$ | $[-0.0132, 0.0045]$ | $> 0.95$ | $[0.0084, 0.0161]$ |
| Reference | $\mu$ | SE | $\mu$ | SE | $\mu$ | SE | $\mu$ | SE |
| GP(L)/SGS | $0.0691$ | $(0.0026)$ | $0.8231$ | $(0.0069)$ | $0.8127$ | $(0.0117)$ | $0.8510$ | $(0.0048)$ |

| Family $\psi$ | Interval $I$ | | Goodness $G$ | | Likelihood $L$ | |
|---|---|---|---|---|---|---|
| | $p$ | CI | $p$ | CI | $p^\dagger$ | CI$^\dagger$ |
| SK/SGS | $< .001$ | $[0.0624, 0.0905]$ | $< .001$ | $[-0.0950, -0.0746]$ | $< .001$ | $[-0.0805, -0.0503]$ |
| OK/SGS | $< .001$ | $[0.0127, 0.0336]$ | $< .001$ | $[-0.0188, -0.0070]$ | $0.0019$ | $[-0.0343, -0.0066]$ |
| GP(G)/CRF | $> 0.99$ | $[-0.0470, -0.0337]$ | $0.1961$ | $[-0.0075, 0.0029]$ | $0.0671$ | $[-0.0239, 0.0032]$ |
| Reference | $\mu$ | SE | $\mu$ | SE | median | $[q_L, q_U]$ |
| GP(L)/SGS | $0.6180$ | $(0.0073)$ | $0.7997$ | $(0.0059)$ | $0.5432$ | $[0.2996, 0.7053]$ |

† The more conservative Welch's t-test is used assuming unequal population variance.

ought to be maximised, viz., $X \in \{F, A, P, G, L\}$, a true $H_a$ indicates the GP(L) family has superior performance. For scores that ought to be minimised, the inequality signs are reversed such that $H_0(Y, \psi) : \mu_Y^\psi < \mu_Y^{\text{GP(L)}}$ and $H_a(Y, \psi) : \mu_Y^\psi \geq \mu_Y^{\text{GP(L)}}$ for $Y \in \{H, I\}$. The p-values are reported in Table 13 along with the 95% confidence intervals for the difference (viz., $X^\psi - X^{\text{GP(L)}}$ or $Y^\psi - Y^{\text{GP(L)}}$) under the alternative hypothesis that the two are unequal.

### 5.2.8  Interpretations

A direct translation of the results in Table 13 is as follows. At a statistical significance (p-value) of 0.05, the alternative hypothesis, $H_a(h_{\text{EM}}, \psi)$ is accepted for all models $\psi \in \{$SK, OK, GP(G)$\}$. This means, in terms of global distortion in the predictive mean, the performance of GP(L) is superior to SK, OK and GP(G). In regard to spatial fidelity, the alternative hypothesis, $H_a(F, \psi)$ is also accepted for all models. Not only is the spatial fidelity of GP(L) higher than SK, OK and GP(G), the confidence intervals indicate that GP(L) is superior by a large margin. To estimate their respective differences, dividing the

CI midpoints [-0.3958, -0.1995, -0.0556] by mean($F$) = 0.8231 for GP(L), one arrives at an average loss in spatial fidelity of 48%, 24% and 6.7% if the SK, OK and GP(G) models are used with SGS/CRF simulation in place of GP(L)/SGS.

In regard to accuracy, $H_a(A, \psi)$ is accepted for SK and OK but rejected for GP(G). This means, the accuracy of the predictive distributions generated by GP(L) is superior to SK and OK, but not significantly different to GP(G) given a p-value of 0.1685, with zero contained in the CI [-0.0132, 0.0045]. The alternative hypothesis $H_a(P, \psi)$, on the other hand, is rejected for all

models. This implies the precision of the GP(L) predictive distributions is not superior to SK, OK and GP(G). However, GP(L) is inferior only by a small margin with a combined CI of [0.008, 0.039]. Because the precision score is conditioned on having an




accurate distribution where only instances of $\bar{\kappa}(p) > p$ are counted [here, $p$ represents proportions as defined in Sec. 3.3.3], the goodness statistic $G$ is generally considered a more prudent measure. Since $H_a(G, \psi)$ is accepted for SK and OK but rejected for GP(G) at a p-value of 0.196, GP(L) adheres more closely to $p$-probability interval groundtruth containment expectations than either SK or OK; and the differences between GP(L) and GP(G) are insignificant. This finding is corroborated by the likelihood score, as $H_a(L, \psi)$ is also accepted for SK and OK but rejected for GP(G) at a p-value of 0.067. Finally, the GP(L) prediction intervals are narrower for all models except GP(G) since $H_a(I, \psi)$ is accepted for both SK and OK.

Collectively, these significance tests indicate that GP(L)/GP-SGS—Gaussian Process Regression using local neighbourhood mean with Sequential Gaussian Simulation—is superior to both simple kriging (SK/SK-SGS) and ordinary kriging (OK/OK-SGS) approaches. The confidence intervals for $X^{\psi} - X^{\text{GP(L)}}$ quantify the margin of superiority, and the evidence from Table 13 is extremely strong against SK/SK-SGS on all scores; very strong against OK/OK-SGS with respect to histogram ($H$), fidelity ($F$) and accuracy ($A$), and moderate with respect to goodness ($G$) and likelihood ($L$). The t-tests also confirm the performance of GP(G)/GP-CRF—Gaussian Process Regression using stationary global mean with Cholesky Random Field simulation—is close to GP(L)/GP-SGS with respect to $H$, $A$, $G$ and $L$. In fact, GP(G)/GP-CRF prediction intervals tend to be narrower. The main reason for preferring GP(L)/GP-SGS is that it achieves higher spatial fidelity based on the $F$ score which is informed by variogram considerations as discussed in Sec. 3.2 and Sec. 5.1.2.

## 5.3 Comparison with in-situ regression

Experiment results for in-situ regression (i.e., performing interpolation instead of extrapolation) were separately compiled. The same procedures were followed, thus the same analysis and graphics seen in Sec. 5.1.1–5.2.7 were produced and included in the supplementary material [note: figures and tables therein carry the S prefix]. Image-based views of the statistics across domains and inference periods are shown in Fig. S.7–S.11. At a high level, similar patterns emerge albeit with greater clarity. The main features can be seen in Table 14 which compares the summary statistics for in-situ regression with future-bench prediction. This table shows the average scores for in-situ regression and expresses differences as percentage changes relative to the average scores for future-bench prediction. [For brevity, standard errors are omitted, these details can be found in Tables S.2–S.6] An insight from the $F$ scores is that the spatial fidelity gaps are smaller between OK/SGS, GP(G)/CRF and GP(L)/SGS for in-situ regression, however, GP(L)/SGS really excels and the gaps widen under future-bench prediction. For the remaining discussion, it is instructive to focus on the last row for GP(L)/SGS in Table 14. The reduction in the histogram and interval scores ($\Delta H \approx -45\%$ and $\Delta I \approx -28\%$) show improvement in mean grade distribution resemblance and contraction in the prediction interval; the latter in particular points to a more confident model. These, together with associated improvements in the fidelity, accuracy, goodness and likelihood scores ($\Delta F \approx +5.2\%$, $\Delta A \approx +19.6\%$, $\Delta G \approx +6.1\%$, $\Delta L \approx +9.6\%$) indicate how much easier in-situ regression is compared with future-bench prediction. The level of difficulty associated with a prediction task is too often omitted from model analysis; this is something to be mindful of.

Significance testing was also carried out on the in-situ regression results. A comparison of Table 13 (the future-bench results) with Table S.7 (the in-situ results) confirms the relative merits of GP(L)/SGS over OK/SGS and SK/SGS remain unchanged. Minor differences exist with respect to the alternative hypotheses $H_a(A, \text{GP(G)/CRF})$ and $H_a(L, \text{GP(G)/CRF})$ which were



**Table 14.** Performance comparison with in-situ regression. Statistical scores for in-situ regression are shown. Parenthesis shows percentage change $\Delta_{\text{future}\rightarrow\text{in-situ}}$ as a general improvement relative to future-bench prediction. Note: The increase in difficulty going from in-situ regression to future-bench prediction is given by $\Delta_{\text{in-situ}\rightarrow\text{future}} = -\Delta_{\text{future}\rightarrow\text{in-situ}}/(1+\Delta_{\text{future}\rightarrow\text{in-situ}})$. Figures are aggregated over all domains and inference periods.

| Family $\psi$ | Histogram $H\!=\!h_{\text{EM}}$ | | Spatial Fidelity $F$ | | Accuracy $A$ | | Precision $P$ | | Interval $I$ | | Goodness $G$ | | Likelihood $L$ | |
|---|---|---|---|---|---|---|---|---|---|---|---|---|---|---|
| SK/SGS | 0.1186 | (-29.3%) | 0.5149 | (+20.5%) | 0.6142 | (+17.1%) | 0.8840† | (+1.84%) | 0.6964 | – | 0.7876 | (+10.1%) | 0.4731 | (+4.50%) |
| OK/SGS | 0.0790 | (-29.8%) | 0.7175 | (+15.0%) | 0.9051 | (+26.1%) | 0.8310 | (-4.17%) | 0.5924 | (-7.55%) | 0.8407 | (+6.85%) | 0.5708 | (+11.1%) |
| GP(G)/CRF | 0.0409 | (-45.6%) | 0.8482 | (+10.5%) | 0.9666 | (+19.5%) | 0.8303 | (-3.86%) | 0.4402 | (-23.9%) | 0.8546 | (+7.17%) | 0.5853 | (+9.07%) |
| GP(L)/SGS | 0.0382 | (-44.7%) | 0.8656 | (+5.16%) | 0.9723 | (+19.6%) | 0.8109 | (-4.71%) | 0.4445 | (-28.0%) | 0.8487 | (+6.12%) | 0.5954 | (+9.60%) |

† Conditional on having an accurate model. Computed using only 61% of the samples in the case of SK.

accepted with p-values of 0.0012 and 0.0137. This suggests GP(L)/SGS has a slightly better accuracy and likelihood scores than GP(G)/CRF at a significance level of 0.05 when the confidence intervals in Table S.7 are taken into account.

### 5.3.1 Effects of simulation

The issue of how sequential or CRF simulation affects the predictive performance of probabilistic models has received little attention in geoscientific literature. This section seeks to provide some answers by asking whether SGS/CRF simulation actually improves the base models and in what way. Table 15 shows the *sample-weighted* performance statistics for future-bench prediction across all domains and inference periods. First, the accuracy of predictive distributions is examined to infer the convergence behaviour of SGS and CRF. The abrupt improvement from $A\!<\!0.17\,(s\!=\!4)$ to $A\!>\!0.85\,(s\!=\!8)$ shows that approximately eight simulation runs is required to produce valid predictions. This can be verified by inspecting the absolute synchronicity columns which show the lower and upper quartiles, $|S|_{.25}$ and $|S|_{.75}$, start exceeding 0.25 and 0.75 respectively when $s \geq 8$. The goodness measure ($G$) achieves its maximum when $s\!=\!8$ whereas the likelihood ($L$) keeps increasing and attains a value superior to the base model after $s\!=\!32$ iterations. This suggests a reasonable number of simulations to choose is between $s\!=\!8$ and $s\!=\!64$. For both GP-CRF and GP-SGS, the accuracy, precision and goodness statistics are comparable to the GP(G) and GP(L) base models after $s = 32$ simulation runs. Hence, GP(L) and GP(G) both generate competent probabilistic predictions. The main reason for preferencing GP-SGS or GP-CRF is that their mean predictions achieve higher spatial fidelity than their corresponding base model, as evident from the $F$ column.

## 6 Discussion

Two take-home messages stem from this work. First, systematic evaluation is crucial for understanding the uncertainty and predictive performance of probabilistic models. Too often, practitioners focus only on global accuracy, neglecting aspects such as uncertainty and local correlation (or vice-versa). This can lead to an incomplete and sometimes flawed understanding of the strengths and deficiencies of models. To address this imbalance, this work advocates a comprehensive approach known



**Table 15.** Sample-weighted performance statistics for future-bench prediction

| Model | Histogram $h_{JS}$ | Spatial Fidelity $F$ | Abs. Synchronicity $|S|_{.25}$ | $|S|_{.75}$ | Likelihood $L$ | Accuracy $A_{\xi=.05}$ | Precision $P$ | Goodness $G$ | Interval $I$ |
|---|---|---|---|---|---|---|---|---|---|
| GP(G)_nst | 0.1260 | 0.4856 | 0.3930 | 0.8229 | 0.5892 | 0.9585 | 0.8111 | 0.8954 | 0.4675 |
| GP(G)_CRF_from_2 | 0.0557 | 0.8617 | 0.0009 | 0.5166 | 0.2795 | 0.0047 | 0.9999 | 0.5578 | 0.5170 |
| GP(G)_CRF_from_4 | 0.0658 | 0.7424 | 0.1343 | 0.7035 | 0.4317 | 0.1440 | 0.9983 | 0.8605 | 0.5508 |
| GP(G)_CRF_from_8 | 0.0796 | 0.6740 | 0.2748 | 0.7643 | 0.5138 | 0.8532 | 0.9481 | 0.9498 | 0.5693 |
| GP(G)_CRF_from_16 | 0.0904 | 0.6303 | 0.3495 | 0.7948 | 0.5608 | 0.9606 | 0.8708 | 0.9280 | 0.5658 |
| GP(G)_CRF_from_32 | 0.0990 | 0.6063 | 0.3931 | 0.8111 | 0.5866 | 0.9768 | 0.8211 | 0.9053 | 0.5687 |
| GP(G)_CRF_from_64 | 0.1052 | 0.5851 | 0.4170 | 0.8208 | 0.6020 | 0.9800 | 0.7909 | 0.8908 | 0.5631 |
| GP(G)_CRF_from_128 | 0.1094 | 0.5745 | 0.4297 | 0.8257 | 0.6103 | 0.9803 | 0.7748 | 0.8831 | 0.5615 |
| GP(L)_nst | 0.0896 | 0.6368 | 0.3935 | 0.8273 | 0.5933 | 0.9618 | 0.8032 | 0.8918 | 0.4748 |
| GP(L)_SGS_from_2 | 0.0576 | 0.8247 | 0.0012 | 0.5177 | 0.2811 | 0.0036 | 0.9999 | 0.5611 | 0.5814 |
| GP(L)_SGS_from_4 | 0.0579 | 0.8499 | 0.1504 | 0.7001 | 0.4369 | 0.1680 | 0.9980 | 0.8705 | 0.6202 |
| GP(L)_SGS_from_8 | 0.0606 | 0.8354 | 0.2909 | 0.7622 | 0.5208 | 0.8662 | 0.9382 | 0.9490 | 0.6323 |
| GP(L)_SGS_from_16 | 0.0687 | 0.8105 | 0.3651 | 0.7915 | 0.5665 | 0.9533 | 0.8596 | 0.9226 | 0.6372 |
| GP(L)_SGS_from_32 | 0.0732 | 0.7924 | 0.4065 | 0.8066 | 0.5914 | 0.9653 | 0.8116 | 0.9006 | 0.6404 |
| GP(L)_SGS_from_64 | 0.0793 | 0.7791 | 0.4304 | 0.8137 | 0.6067 | 0.9741 | 0.7820 | 0.8866 | 0.6426 |
| GP(L)_SGS_from_128 | 0.0815 | 0.7718 | 0.4454 | 0.8197 | 0.6153 | 0.9781 | 0.7651 | 0.8786 | 0.6417 |

This represents a cropped version of Table S.8 in the supplementary material.

as FLAGSHIP—an acronym for fidelity, likelihood, accuracy, goodness, synchronicity, histogram, interval and precision—which assesses the pmf, variogram and uncertainty properties relating to the models. A key benefit with FLAGSHIP is that its statistical scores are standardised and interpretable. For instance, the Jensen-Shannon and Rudzica histogram distances are both

bounded between 0 and 1 and have information- and set-theoretic interpretations. This makes it possible to compute averages for these quantities, and others such as the fidelity ($F$) and likelihood ($L$) scores, meaningfully across geological domains, inference periods and in a variety of settings. More importantly, it facilitates comparison across commodities and mine sites. The interpretation of the FLAGSHIP statistics is universal and independent of geochemistry. In contrast, conventional measures such as MSE can vary considerably depending on the location and target attribute; it does not have a clear meaning standing

on its own. As a case in point, the concentration of copper and molybdenum are measured in units of wt% and ppm which are incompatible, so their MSE cannot be pooled together.[4] However, using the $L$ score, sensible comparisons can be made. This is also the reason for incorporating $\sigma_Y$ normalisation in our definition of interval tightness in (32).

The second point is that FLAGSHIP enables significance testing via large-scale model evaluation, and allows modelling performance to be contextualised. For a well-balanced study, a sufficiently large dataset with varied characteristics (such as

distribution diversity) should be used where possible to minimise selection bias and present challenges to the models. A well recognised problem in the mining industry is that model evaluation takes tremendous time and effort. One novel aspect of

---

[4]When the statistical scores for two sets of observations are not on equal footing, it is possible for one to dominant and masquerade changes (improvement or deterioration) in the other.




this work is the conversion of the variogram from a visual diagnostic tool into a quantitative measure of spatial fidelity. When hypothesis testing is applied to the FLAGSHIP measures, users can establish if there are statistical differences between the models and quantify these differences using confidence intervals as seen in Sec. 5.2.7–5.2.8. Model performance is often re-
ported without much thoughts on how demanding the problem or data is. This is especially true for future-bench prediction, where there are no protocols or standardised measures for articulating how challenging the geology or modelling task is. This opacity is a source of frustration, as it is difficult to assess whether a promising approach would be efficacious in a different situation without some benchmark. In Sec. 5.3, we have shown that it is possible to quantify the decline in model performance, or infer the increase in difficulty, moving from in-situ regression (interpolation) to future-bench prediction (extrapolation). Col-
lectively, these could form the basis of one or more objective measures to help communicate geological modelling difficulties, and by extension, draw attention to challenging areas with a view of deploying additional drilling, sensing or adaptive sam-pling to reduce uncertainty and optimise mining operations in an intelligent, automated and cost effective way (Leung et al., 2023a). In particular, the synchronicity score can generate local distortion maps for probabilistic predictions as demonstrated in Sec. 5.1.6.

## 6.1 Recommendations

A number of guidelines have been devised based on the findings of this study. These recommendations, collectively referred as EUP$^3$M, are collated in Table 16. EUP$^3$M outlines what the authors would consider as current best practice in evaluating the uncertainty and predictive performance of probabilistic models.





**Table 16.** EUP³M recommendations for evaluating the uncertainty and predictive performance of probabilistic models

| | | |
|---|---|---|
| Material | Select a sufficiently large dataset with target attribute diversity to minimise selection bias and challenge the models. | §4.2 |
| Design | Experiments should reflect observational and modelling constraints in practice. For instance, the data available in each inference period defines the scope of our regression/prediction tasks in a manner that it emulates staged data acquisition and progression of mining activities in a real mine. | §4.3 |
| Measures | Compute the FLAGSHIP statistics to investigate the global accuracy, local correlation and uncertainty-based properties of the models relative to the groundtruth. [FLAGSHIP encompasses the spatial **f**idelity, **l**ikelihood, predictive distribution **a**ccuracy, **g**oodness, **s**ynchronicity, **h**istogram distance, **i**nterval tightness and **p**recision scores as defined in (16)–(32)] | §3.1–3.3.6 |
| Analysis | Perform one or more of the following according to needs | |
| | (a) Compute summary statistics to assess group performance: e.g., aggregate values by model family, average over domains or time periods (see Tables 7–12) | §5.2.1–5.2.6 |
| | (b) Observe general trends and variation in individual models: Perform large-scale simultaneous comparison across models and conditions using image-based views of the relevant statistics to identify instances where models may have underperformed (see Fig. 22,23) | §5.2 |
| | (c) Establish statistical significance and confidence interval: e.g., perform hypothesis testing using t-tests and interpret the results using p-values; | §5.2.7–5.2.8 |
| | (d) Contextualise model performance: e.g., compare in-situ regression with future-bench prediction to articulate the difficulty of extrapolation relative to interpolation. Pairwise comparison can also reveal the benefits of modelling with additional data. | §5.3 |

# 7 Conclusions

Although this paper began with a description of geostatistical models, its core contribution and objectives remain firmly on developing measures and novel ways for assessing and comparing the predictive performance of probabilistic models with observational data. Section 2 reviewed the theories that underpin Gaussian Process and Kriging regression and outlined the procedures for Sequential Gaussian and Cholesky Random Field Simulations (SGS and CRF). Section 3 examined three categories of geostatistics which comprises: a) histogram measures that reflect the global accuracy in the mean estimates, such as

the probabilistic symmetric $\chi^2$, Jensen-Shannon, Ruzicka and Wasserstein distances; b) variogram measures that target spatial correlation and local variability in the model predicted mean; c) uncertainty-based measures that assess the performance of probabilistic models using both the mean and standard deviation estimates, $\hat{\mu}(\boldsymbol{x}_*)$ and $\hat{\sigma}(\boldsymbol{x}_*)$, and groundtruth or actual grade, $\mu_0(\boldsymbol{x}_*)$. An example was presented using synthetic data to develop the basic intuition, before the measures were applied to real models and data obtained from a porphyry copper deposit. Section 4 described the geological setting, data attributes,

general considerations and implementation of the experiments. It explained the importance of having diversity in the data, and distinction between future-bench prediction and in-situ regression. Section 5 provided in-depth analysis, focusing initially on





the efficacy of the histogram, variogram and uncertainty measures in two domains within one inference period. Its scope was subsequently expanded to encompass the entire dataset—this includes up to 11 domains and 12 inference periods—to eliminate selection bias and ensure the results would be fair, representative, and statistically significant.

The proposed measures and analytic approach provided insights and clarity. One observation in relation to histogram distance, $H$, is that the JS divergence, Ruzicka and p.s. $\chi^2$ distances are sensitive to discretisation. They may give the false impression that a model is underperforming when few inference points are involved. This confounding effect can be suppressed by using the Wasserstein distance, since it does not involve quantisation and can be computed directly from order statistics. Another benefit of viewing the $H$ statistics as an image is that it focuses attention on difficult cases. Targeted investigation subsequently revealed that instances of poor predictive performance (see light blue pixels in Fig. 22) can generally be explained by a significant mismatch between the training data and groundtruth grade distribution, or insufficient training data for certain domains/periods in this study. In terms of insights, inspection of the variogram curves and automatic determination of variogram ratios had uncovered a general trend, viz., GP-SGS produces results with higher spatial fidelity, $F$, than the GP(L) base model. This finding indicates that while GP(L) can model a random process with non-stationary mean using samples in the local neighbourhood of the inference points, it does not adequately capture mid-range or long-range spatial correlation; so it benefits from sequential simulation which propagates mid-to-long range conditional dependence according to the chain rule in (13).

The lessons pertaining to the uncertainty measures are that Deutsch's accuracy, $A$, is useful for indicating SGS/CRF convergence whilst $P$, $G$ and $I$ convey the conditional precision, goodness and tightness of the model predictive distributions. The synchronicity measure, $S$, was described in connection with the concept of $p$-probability intervals which is used to judge the performance of probabilistic models. The goodness criterion is whether the groundtruth containment intervals live up to expectation, that is, how close the observed proportions, $\bar{\kappa}(p)$, are to the expected proportions, $p$. The proposed likelihood measure, $L$, while related to $G$, is more discerning as it is a decreasing function of the Z score, $(\hat{\mu} - \mu_0)/\hat{\sigma}$. An important reason for computing the synchronicity, $S$, from which the likelihood is derived, is that it can be rendered as a distortion map to identify areas where overestimation or underestimation had occurred. Collectively, the FLAGSHIP statistics provide a standardised approach that is amenable to large-scale simultaneous comparisons between many models. Unlike other measures such as the RMSE, these statistics can be aggregated/averaged meaningfully across spatial and temporal domains, or even compared between different target variables (such as copper and molybdenum). For significance testing, t-tests were applied to establish the superiority of the GP-SGS and GP-CRF models relative to simple and ordinary kriging and their SGS counterparts based on the FLAGSHIP statistics. Finally, the performance gap of future-bench prediction (extrapolation) relative to in-situ regression (interpolation) was quantified to contextualise the increased difficulty of the inference task. In summary, this work described a systematic approach for evaluating the uncertainty and predictive performance of univariate probabilistic models using the FLAGSHIP statistics. This culminated in a set of recommendations (see Table 16) for assessing, comparing and validating probabilistic models to serve a range of needs including standardisation.



*Code and data availability.* An open-source implementation of the algorithms described in this article is available from
**https://github.com/acfr/eup3m** (Leung and Lowe, 2024) and archived in **https://doi.org/10.5281/zenodo.14533140**.
The **eup3m.git** repository provides anonymised test data, Python code for model construction and statistical analysis, a bash script to repli-
cate the experiments and a Jupyter notebook to reproduce key figures. These are further described in Sec. 4.4 and in the README.md file.

*Author contributions.* **Raymond Leung**: Conceptualization, Methodology, Investigation, Formal analysis, Writing - Original Draft, Soft-
ware, Data Curation, Visualization. **Alexander Lowe**: Conceptualization, Methodology, Software, Validation, Writing - Review & Editing.
**Arman Melkumyan**: Conceptualization, Methodology, Writing - Review & Editing, Funding acquisition.

*Competing interests.* The authors declare no conflict of interest.

*Acknowledgements.* This work was supported by the Australian Centre for Robotics and Rio Tinto Centre at the University of Sydney. Rio
Tinto Kennecott Copper is thanked for providing the test data used in this study. The authors would like to acknowledge Adrian Ball and Xin
Tong Wang (IST) for their project management roles and Francisco Rodriguez (RTKC) for his advisory role.



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
