# Peer review of "Evaluating uncertainty and predictive performance of probabilistic models devised for grade estimation in a porphyry copper deposit"

_EGUsphere, 2024_

## Author Comment (AC1)

**Reviewer 1's comments**
https://doi.org/10.5194/egusphere-2024-4051-RC1

**RC1**: 'Comment on egusphere-2024-4051', Anonymous Referee #1, 27 Feb 2025

This manuscript proposes a set of metrics for probabilistic model validation and selection. This approach is applied to a synthetic example and a real case study, with a strong focus on mining applications.

Overall the manuscript is well written, although it could be shortened and the vocabulary is sometimes confusing. Analysis 2 shows some valuable work on metric visualization, but the manuscript lacks a thorough theoretical base and, in the end, the analysis isn't designed in a way that can prove that the metrics - most of which being not new - improve our ability to validate and select models compared to the current best practices. Another objective was to compare kriging and Gaussian processes, but the analysis isn't robust enough to conclude anything there. I think there is potential to make valuable scientific contributions on model comparison and validation based on this application, but this will require a more careful reflection on the study's goals and how to best achieve them.

[**A1**] The authors would like to thank this reviewer for their constructive comments and taking the time to provide such a thorough review. We realize from reading these comments that we have not been sufficiently clear when it comes to stating the motivations and principal objectives of this study. We are committed to reflect and improve on these and other areas in the revised manuscript to avoid misunderstanding (see further comments and how this is addressed in [**A71**]).

Re: "the manuscript lacks a thorough theoretical base…"

[**A1a**] The authors noted that a significant portion of criticism relates to modelling process and formal mechanisms, not the methods or modalities used to assess or compare the modelling results. The latter is the submission category and criteria by which this manuscript should be judged according to the journal guidelines. It is important to correct this misconception, as it sets the tone and expectation that underline much of the feedback which currently focuses disproportionately on the theoretical foundations of models. The authors accept responsibility for the lack of clarity that have contributed to this confusion, and will seek to manage expectations better in the following response and the revised manuscript. We acknowledge that this reviewer has raised some legitimate points (for instance, enforcing consistent neighbourhood definitions in all models for use during probabilistic inference) that would be relevant to and actionable in a *model evaluation* paper. However, there is no need for models to be symmetrically configured in order to be compared in a *methods-for-assessment* paper.

Re: "the analysis isn't designed in a way that can prove that the metrics… improve our ability to validate and select models compared to the current best practices"

[**A1b**] First and foremost, the methods are centred on three aspects, viz., characterising the global accuracy, spatial fidelity of ore grade models, and assessing whether the probabilistic predictions are sound (i.e. well calibrated with probabilities). These are the highest priorities. These aspects, which are usually not jointly investigated in mining literature, form the cornerstone of our investigation. They were examined using histogram, variogram and uncertainty-based measures. Although we observed the Wasserstein measure is more robust than other histogram-based measures such as Jensen-Shannon divergence, the exact choice is not critical to this framework, so long as each of the three categories is fairly represented. For instance, the Wasserstein measure can be replaced by the $R^2$ coefficient of determination if that is what users would prefer. The analysis is not designed to prove the suggested measures are superior to other alternatives because that was never the aim. At most, we would say they are fit-for-purpose. For instance, the variogram ratio statistic is a reasonable indicator for spatial fidelity (over-smoothing) which is something geologists particularly object to.

[**A1c**] We think the main contribution of this paper lies in the approach and value it adds to the analysis. One specific example is the image-based visualisation of measures [that reviewer 1 had also identified] which can help focus attention on identifying and examining situations where models have under-performed. The synchronicity scores can be used to produce maps to reveal locations where under- or over-estimation had occur. This is not something that the uncertainty-based statistics would normally provide. Last but not least, hypothesis testing was applied to suitable measures to compute confidence intervals and check for significant differences between models; a comparison was made between in-situ regression and future-bench prediction to quantify the increased difficulty of extrapolation. This can help inform geological complexity and sufficiency of data gathered in different geological domains. Demonstrating these benefits was the primary motivation behind this study.

[**A1d**] In regard to the comparison of GP with kriging, the aim was to give meaning and substance to the assessments and subject the models to data with varying distributions. The main findings are that their performance were compatible based on the predictive distribution uncertainty-based measures, however, differences in the local neighbourhood definitions (cf. local approximate GP) can lead to differences in terms of spatial fidelity. Whether this constitutes something worthwhile or not is in the eyes of the beholder. We respect the reviewer's opinion if the present analysis is deemed not "robust enough" to draw conclusion. Ultimately, this is probably beside the point, as the manuscript is being considered as a *methods-for-assessment* paper, as opposed to a *model evaluation* paper. According to the journal guidelines, the former is expected to "describe new standard experiments for assessing model performance or novel ways of comparing model results with observational data." It is not expected to settle some debate between models (pitting the semi-variogram approach against length-scales optimisation by maximising the marginal likelihood). The subjects for comparison can indeed be any model and be arbitrarily configured.

**Major comments:**

I find the way you've introduced Gaussian processes (GPs), kriging, and Gaussian simulation confusing, with some comments being factually incorrect. To be clear, kriging and Gaussian processes are both based on the random function concept with Gaussian distributions, so they have the same theoretical basis, and Gaussian process and simple kriging estimate and use their parameters in the same way. This is made clear by Williams & Rasmussen (1995, https://proceedings.neurips.cc/paper_files/paper/1995/ file/7cce53cf90577442771720a370c3c723-Paper.pdf):

"Gaussian processes have also been used in the geostatistics field (e .g. Cressie, 1993), and are known there as "kriging"."

And also by Rasmussen & Williams (2006, https://gaussianprocess.org/gpml/chapters/RW.pdf), which repeat that comment p.30, and introduce GP as a linear predictor p.17 (which corresponds to the usual way kriging is introduced in geostatistics and the way you've introduced it in the manuscript).

The key practical difference between kriging and GPs is the inference of the hyper-parameters:

- In the subsurface we often don't have enough data to robustly estimate a semivariogram model, so for kriging the process is done manually based on an experimental semivariogram. It's been proposed to fit the semivariogram model automatically to the experimental semivariogram (e.g., Pardo-Igúzquiza, 1999, https://doi.org/10.1016/S0098-3004(98)00128-9), but some authors have discouraged it because of risk of biases.
- In machine learning (where GPs have been formalized), the goal is to automate as much as possible, so hyperparameters are optimized based on the negative log likelihood.

So when comparing kriging and GPs, you're not comparing two different approaches, you're comparing two different ways of inferring the hyper-parameters of the same approach (at least for

simple kriging, ordinary kriging with a local neighborhood is different, but equivalent techniques exist in the GP world, see Nearest Neighbor Gaussian Procceses).

[A2] We concur. The comments on their relationship are not in dispute. The authors appreciate the reviewer is dissatisfied with how kriging and GP are presented. We tried to capture the theory from angles more familiar to the geostatistics and ML communities, following the examples of Oliver et al. (2015) and Williams and Rasmussen (2006). On the substantial points, we recognise that we have missed an opportunity to emphasise that kriging and GP indeed share the same conceptual foundations. To be fair though, the authors have re-read Sec. 2.1 and 2.2, and did not find anything that contradicts this. It seems the issue stems from Line 40 or there about. Whether we use the word "ways" or something else, we are in agreement that they are theoretically equivalent, but the computational approaches for inferring the hyper-parameters are different, as the reviewer has eloquently put it, one relies on fitting variograms while the other maximises the marginal likelihood. When different options (such as neighbourhood definitions) are used and the models are configured differently, we believe there is justification for treating them as different models. It is important not to lose sight of the fundamental objectives of this work, which is to present ways to compare model results with observational data, irrespective of what modelling processes are used. Any transparent model is better than a black-box. Even if the models share the same theoretical foundations and are merely configured differently, use inferior approximations or some bias had been introduced; none of these diminish the utility of the assessment methods.

Then comes the question of transforming the data. As long as this transformation is linear (e.g., simple standardization), there's no problem. But when it is non-linear, back-transforming the mean prediction and any confidence interval from kriging/GP can lead to biases, so it shouldn't be done. There are specific techniques to deal with this (e.g., log-normal kriging), but a simple and generic way of doing this is multi-Gaussian kriging (see Deutsch & Journel, 1997, p.81-82, http://claytonvdeutsch.com/wp-content/uploads/2019/03/GSLIB-Book-Second- Edition.pdf, although the original author is probably Verly I think), where kriging is applied on the transformed data, then we simulate multiple realizations and we back-transform them (so not directly back-transforming the kriging mean as you seem to do). This is equivalent to using SGS directly, and taking the mean of the realizations at each location.

[A3] Thank you for the explanation. By in large, this is not an issue. All simulation approaches, particularly, experiments carrying the serials *K-SGS, GP-SGS and GP-CRF already follow this paradigm (i.e., apply kriging/GP to transformed data, simulate multiple realisations, back transform them, then compute the mean and variance of the realisations at each location). One exception is GP(L), where the moments of the transformed random variable is approximated using Taylor expansion (Hendeby and Gustafsson, 2017); an alternative would be via Monte Carlo simulation which we rather avoid. This is a pragmatic modelling decision for an isolated case, that along with numerous other assumptions/ approximations are present in real-world models. We accept the criticism that this approach is an approximation and might be less reliable, but the point of a "methods for assessment" paper is not to restrict comparisons to only the most optimal or perfect models.

Following up on simulation, Gaussian simulation is a way of sampling from the normal distributions of a Gaussian process while preserving the covariance function away from the data. It can be done in one generative step based on LU decomposition or based on a FFT when using a grid, or sequentially based on the chain rule of probability. This rule tells us that the fields generated in one step or sequentially are equivalent. What breaks this equivalency is when we use a neighborhood in the sequential predictions, which is often done in practice for efficiency.

[A4] The authors take this comment as clarification. Based on this description, our "GP(G)/CRF" models use Gaussian Process assuming a stationary/global mean and performs Gaussian simulation using LU (Cholesky) decomposition. All things being equal, "SK/SGS" (simple kriging, followed by sequential Gaussian simulation) should produce more or less the same results, except, the models are configured differently as a point of interest. SK uses an isotropic variogram model, whereas GP uses

an anisotropic Matern 3/2 kernel with possibly different length scales along x, y and z. [This is a more realistic depiction of how we envisage the assessment methods will be used. Instead of enforcing uniformity between the models, practitioners will intentionally apply different settings or change model configurations and assess the impact of these decisions].

Continuing with our GP(L)+SGS models, they perform sequential Gaussian simulation by invoking the chain rule. Here, "(L)" indicates using the local mean during inference, akin to the Nearest Neighbours GP approach which is often done for efficiency reason. Our results also show this provides more accurate regression of the underlying random process.

The entire manuscript needs to be reshaped to properly account for theory. This includes not introducing kriging and GPs as two different methods, being clear on how hyper-parameters are inferred, and not testing cases that we know from theory are not optimal (e.g., using the back-transformed mean of kriging/GP, using too few realizations) or the same (comparing one step generative method with a sequential approach, unless you use a restricted neighborhood in the sequential scheme, but then this needs to be detailed). The methods also need to be clearly explained (e.g., what you're doing in GP(L) isn't explained with enough details, what SK-SGS and OK-SGS mean isn't self-explanatory: are you first using SK or OK then sampling from the distributions using SGS or are you using directly sampling suing SGS with SK or OK? Those aren't quite the same).

[**A5**] The authors will commit to clearing up on how the hyperparameters are inferred and emphasising that kriging and GP share the same conceptual foundations. As it stands, the former was described, although only briefly, in lines 111-115, 150-155.

For reasons mentioned previously in [A1a, A1c]---that this paper is not principally about drawing conclusions on performance of different model techniques, rather it is about adding value to the analysis, for instance, the ability to draw attention to instances where models have under-performed, revealing locations where under-estimation had occurred---optimal model construction is not necessary for this study to stand. There is no reason to exclude a model because it is not theoretically optimal, this flies against the reality that pragmatic decisions are made and approximations are used in practice.

There seems to be a suggestion that taking the "back-transformed mean" is a pervasive problem, and that certain model configurations are redundant, neither of these is true. Just briefly, differences are introduced through the use of isotropic covariance functions (for kriging) and anisotropic kernels (for GP), as well as restricted and different neighbourhoods.

The authors omitted this information and left this to the implementation (open-source code) as we initially felt these details would burden readers, given this paper was submitted as a *methods-for-assessment* paper. In light of this reviewer's comments, we are comfortable adding these technical details (including a clearer description of the coupling of models with Gaussian simulation such as "GP(L)+SGS") in the revised manuscript.

On that note, many references are not the original sources but more recent ones. Beside being unfair to the original authors, I strongly encourage you to have a proper look at the original studies of the methods you're using.

[**A6**] The authors will check and amend references as appropriate in the revised manuscript. This also relates to [A22], [A32].

I also miss a proper literature study and discussion around what has been done on model validation and selection in statistics, geostatistics, and machine learning.

[**A7**] We will work on this as a matter of priority to strengthen the manuscript. In addition, we will add a proper section clarifying the application context and what we seek to achieve.

You state in the abstract that "there are no established protocols for evaluating the uncertainty and predictive performance of univariate probabilistic models", which is a baseless, and even false, claim. A lot has been done on evaluating predictive performance, and scikit-learn has a whole documentation about it:

https://scikit-learn.org/stable/modules/cross_validation.html#cross-validation-of-time-series-data
[**A8**] In hindsight, this was a case of unfortunate wording. It boils down to how "established protocols" is construed. In the writer's mind, established protocols refer to industry standards (such as ISO/IET), official guidelines or procedures observed and/or practised by the mining industry. There is no denying that researchers have contributed many ideas on this front, but these have not been systematically tested or matured to the point where there is common consensus on what is fit-for-purpose for large-scale application. This is an objective fact, at least in the mining geology sector. As humans, we often evaluate situations based on our own experience and perspective. This feedback is a reminder that there can be multiple interpretations and sometimes multiple valid perspectives.

Cross- validation is also the standard method for validation in geostatistics, and is used in Deutsch (1997), which you refer to multiple time in your work. In statistics, you can have a look at the work of Aki Vehtari, who has done many studies on validation, including of GPs. Validation of uncertainty is less often done in practice indeed, but that doesn't mean that no work has been done on it.

[**A9**] The authors have contemplated cross-validation but found it to be unsuitable for the following reasons. First, the data is quite sparse. There are barely enough samples (training data points) gathered for certain geological domains. Partitioning the data would further weaken the analysis. Second, in a k-fold validation set up, the unseen data set aside for testing also belongs to the same region as the training data. This evaluates its in-situ regression performance (essentially, interpolation ability) whereas the main focus of this work is on extrapolation performance, where the model is required to inference into new territories (e.g. to surrounding areas, an adjacent bench or the bench-below).

Using the hold-out method, we already compiled results in Sec. 5.3 that illustrate in-situ regression performance. This task is less demanding compared to future-bench prediction, thus a k-fold validation (hypothetically speaking) that uses the same pool of data would similarly yield optimistic results and not truly reflect model extrapolation performance.

In the end, I remained unconvinced that the proposed FLAGSHIP approach improves our ability to validate probabilistic models. The key problem here is that you don't have any baseline to compare to, so you can't prove that your method improves anything.

[**A10**] To reiterate, the main contribution of this paper lies in the approach and dimensions it adds to the analysis. The analysis is not designed to prove the suggested measures are superior to other alternatives because that was never the aim. Their purpose is to shed light on the global accuracy, spatial fidelity and calibration properties of probabilistic models. Currently, there is no common consensus in the field of mining geology on what procedures should be employed. This paper shows the proposed measures are fit-for-purpose, can be used to evaluate a large number of models/configurations simultaneously across space and time. A useful property is that these statistical measures (FLAGSHIP for short) are standardised, interpretable and amenable to significance testing. They are generally bounded between 0 and 1. Interpretable because measures like fidelity and goodness indicate respectively, how well the variogram (spatial variability) is reproduced compared to the hold-out data/reference, and how well calibrated the predictive distributions are with probabilities. These interpretations do not change with location or the target variable; in contrast, this is not the case for RMSE, it would be misleading to aggregate and average over different geological domains or target variables (e.g. Cu and Zn) or compare model performance based on RMSE at different sites.

These attributes are by no means unique, for instance, $R^2$ would probably be a suitable replacement for the Jensen-Shannon histogram measure. The point remains that the measures we looked at would facilitate such enquiries, i.e., cross-site or cross-species predictive performance comparison, with respect to all three aspects of such models: to evaluate global accuracy, spatial fidelity and how reasonable the uncertainty estimates are.

Returning to the journal's requirements, a *methods-for-assessment* paper is expected to describe new standard experiments for assessing model performance or novel ways of comparing model results with observational data. These criteria were addressed in a number of ways. In Sec. 5.2, image-based visualisation of the FLAGSHIP measures was proposed as a way of focusing attention on identifying and examining situations where models have under-performed, highlighting the sensitivity of certain histogram distance measures to limited test data, and issues such as drifts in the target distribution. In Sec. 5.1.6 (Figs. 20-21), the synchronicity measure, a quantity computed en route to evaluating the goodness of the predictive distributions, enabled us to highlight areas of under- and over-estimation which can reflect modelling deficiencies due to mismatched assumptions or misconfiguration. These observations may lead to modification of the sampling regime, steering future drilling campaigns to capture more meaningful data in uncertain zones. In Sec. 5.2.7–5.2.8, we computed confidence intervals and performed hypothesis testing. This could be a general procedure for establishing if there are significant differences and the extent of these differences between competing models. In Sec. 5.3, we considered the increased difficulty of the extrapolation task relative to in-situ regression in the context of grade modelling in open-pit mining.

I think a simple strategy with a cross-validation (based on group k-fold to assess extrapolation to different domains and intervals), and as metrics R2 to assess predictive performance and expected calibration error for uncertainty (what Deutsch (1997) calls precision basically) would be enough. Those two metrics can also be easily visualized in a similar manner (like in your figure 19), which helps with interpretation and decision making. The log likelihood or negative log probability might be worth testing as a complement. But using too many metrics (including some that can be biased, as you mention in the conclusion) is just counterproductive.

[**A11**] Group k-fold validation is unsuitable for assessing bench-below prediction performance as mentioned earlier in [A9]. There is no barrier to using the $R^2$ coefficient. In our view, the two metrics this reviewer proposes would provide only an incomplete and limited view of model performance. For instance, it may not reveal over-smoothing and the locations where the field is under- or over-estimated.

**Specific comments:**

*Abstract*

Line 2: I'm not sure that "valued" is the right term there, maybe "useful" or "essential" would be more appropriate.

[**A12**] This will be changed to "useful" as suggested.

Line 4: That is a bold statement that statisticians won't enjoy reading. Regarding predictive performance it's just wrong, a lot of work on this has been done on this in statistics and data science, and cross-validation is mentioned in the classical geostatistics textbooks as well (see Deutsch & Journel, 1997, p.94, http://claytonvdeutsch.com/wp-content/uploads/2019/03/GSLIB-Book-Second-Edition.pdf). Regarding uncertainty it is more debatable in the sense that it is less often done in practice, but techniques based on cross-validation also exist for this.

[**A13**] This issue relates to how "established protocols" is interpreted and understood by the reader. As explained in [A8], the authors certainly meant no disrespect. To avoid causing any offence, it might be best changing this to "[established] industry standards".

Line 10: The "with or without sequential or correlated random field simulation" isn't really clear at this stage. I suggest to rephrase this sentence.

[**A14**] We will take up your suggestion when the manuscript is revised.

Line 11: Fidelity to what? What's goodness? How is it different from accuracy or precision? What is synchronicity? How is histogram an evaluation metric? This feels like a somewhat random list of concepts to get a nice acronym, but what it actually means (and whether it is any useful) remains unclear.

[**A15**] We should clarify as part of the third objective that the "spatial fidelity of models" is measured relative to the variogram of the ground-truth/test data. The aspects mentioned by this reviewer (goodness, accuracy etc.) all relate to the model predictive distributions as stated in the text. It is normal for readers to be asking curious questions, such as the concept of synchronicity and how goodness differs from precision, but it would be unusual for an abstract to delve into such details. These questions are answered in Sec. 3 where the measures are described, accompanied by mathematical definitions. To ease the feeling of "randomness", we can bring forward "FLAGSHIP statistics…" and tie these measures with the aim of satisfying the three objectives to assess the quality of the predictive distributions, to make these sentences more connected.

Line 14: Data diversity isn't a property of the experiments in this case, it's a property of the datasets used in the experiments. If you use a single dataset with a single data type, then the data diversity is pretty poor. But the abstract isn't clear on that.

[**A16**] The authors mostly agree. However, we do not think there is much to be gained by drawing this distinction. The experiments are not offered as a predictive service that runs independent of the data. To the contrary, the ability to assess and differentiate models depends to a large degree on how much the models are challenged by the data. Data diversity is therefore a key consideration of these experiments. If the data is too homogeneous; the assessment will likely not reveal any meaningful difference between the models.

*1. Introduction*

Line 27: A single realization isn't enough to quantify uncertainty, you need several ones.

[**A17**] Will modify as "random realisations"

Line 28: "It requires geostatistical simulation of each hydrostratigraphic layer using boundary points specified by geologists." I'm not sure what you mean by that here, nor how it is helpful to understand what you are trying to do.

[**A18**] It is difficult to capture the finer details of a research proposal in a sentence. In engineering geology, boundary points or contact points are understood to be geological interpretations that identify interfaces between different layers such as sandstone and clay.

Line 32: This is the first paragraph of your introduction, yet you start by mentioning studies that are not relevant to your own work? Your goal here is to catch readers' interest, so better go straight to why your specific work is important.

[**A19**] Fair enough. The original thinking was to clarify what is meant by probabilistic models by citing some examples. Then, narrow the focus to ore grade prediction. We will think about how to restructure this and elevate the importance of the objectives for this study.

Line 31: Three studies are a little light for an "intense interest", which only started in the 2020s it seems.

[**A20**] For the very reason you mentioned above, we did not think it was a good idea to cite a lengthy list of works without much explanation, or dedicate more space to describing them, just to clarify what constitutes probabilistic models. We needed to quickly get to the point. The word "intense" can be dropped to address this.

Line 34: "Rapid development"? Interest around probabilistic modeling for the subsurface started in the 1950s, so this statement combined with citations from the 2020s leaves me puzzled. I hope you realize the long history behind probabilistic modeling, both in statistics and geostatistics.

[**A21**] This may have been misconstrued. Recent development extends well beyond classical methods. There are emerging works on Bayesian sparse learning approaches, and compressive sensing (which has roughly a 20 year history if we draw the line under basis pursuit); some in unpublished domains. These directions have not been explored nearly as much in geoscience until recent years. A suitable fix would be using "recent development".

Line 36: It would be much better to cite the original studies on kriging here (so Matheron's work, although Krige's work could also be cited).

[**A22**] This will be done in the revised manuscript.

Line 40: GPs aren't an alternative to kriging, they are the same approach (the only difference comes from what is considered a hyperparameter and how they are estimated).

[**A23**] This has been noted. GP shares the same conceptual foundation as kriging, but their computational approaches are different owing to how their hyperparameters are estimated.

Lines 41-42: Kriging doesn't require a sequential scheme for predictions, this is just a way to alleviate the extra computational cost that comes with more data and larger grids. And a similar scheme has been proposed with GPs (see Nearest Neighbor Gaussian Procceses). Sampling from the posterior distributions (i.e., simulation) is required by both kriging and GPs in case of non-linear transformation (something that is very clear in geostatistics, less so in machine learning unfortunately). And GPs suffer from the exact same limitations as kriging regarding the non-reproduction of the covariance function (again, they are the same method).

[**A24**] Thank you for this comment. This part will be revised to ensure it is factually accurate. The general points we made are consistent with Bai and Tahmasebi (2022) and Ortiz (2020). Without simulation (sampling from the posterior distribution), the predicted mean tends to overestimate small values and underestimates large values; this causes oversmoothing which may be attributed to a deficit in variance, or non-reproduction of the histogram and covariance function. Another issue is that simulation is required to obtain credible uncertainty estimates that are not dominated by sample spacing. Putting computational cost aside, these issues can be mitigated by SGSim (Deutsch and Journel, 1997). Reading your last comment, we appreciate that GPs suffer the same limitations as kriging when all things are equal. However, when the respective models are configured differently and different settings are used (such as the shape of the neighbourhood search and how points are selected to form the local mean), differences can emerge. While these differences might be regarded as a nuisance and discouraged for a *model evaluation* paper, these may be acceptable and even encouraged for a *methods-for-assessment* paper, as these variations reflect how modellers might explore different

options in practice. The authors will seek to make clear in the revision that this is not fundamentally about a contest between kriging and GP, it is about ways of assessing models which happened to share the same theoretical foundations here, but are practically different in terms of how they are computed and configured.

Line 47: At this stage the motivation and goals of the paper are becoming blurry. It would be better to reorganize the introduction to bring the different goals together.

Line 49: The focus on mining is important and should be better highlighted in the introduction and abstract.

Line 63: This is the third time that this objective is mentioned as such in the introduction, which highlights an issue with the flow of ideas and the logic of the introduction.

[A25] Thank you for the advice. On the last three comments, we will give serious thoughts on how these contents can be better organised, and bringing the mining aspect to the fore.

Lines 66: RMSE is just a validation metric. Considering errors as independent or dependent isn't related to the validation metric, it's a modeling decision. And both kriging and GPs assume that errors are independent. If validation shows correlation in the errors, then either there is an issue with the prior model (i.e., the covariance function) or kriging/GP isn't appropriate for the dataset at hand.

[A26] When models are assessed, one consideration is whether the prediction errors are correlated. It seems odd to say independent or dependent errors is a modelling decision, as no model sets out deliberately to produce spatially correlated errors. Perhaps the reviewer is thinking of universal kriging, where a global trend model is first fitted to the data, so that the regression errors would hopefully amount to white noise. As this paper is concerned with assessment methods rather than modelling mechanisms, there is a need to deal with (not dismiss) misbehaving models, and identify where they produce correlated error clusters.

In mining, it is not uncommon for correlated errors to appear in the residuals due to domain delineation errors, unexpected spatial discontinuities, rapid change in the geology, limited data for the models to learn from (which may result in poor estimates of the hyperparameters), or even drifts in the target distribution due to the non-stationary nature of the random process when the available data is extrapolated over large distances.

Line 67: What is the link between errors being dependent and uncertainty quantification? And to any limitations of RMSE as a metric? This whole paragraph is really unclear.

[A27] Fair point. There is no direct relationship between dependent errors and uncertainty quantification. But uncertainty quantification is only one aspect of model assessment. Liemohn et al. (2021) has argued for instance that RMSE by itself may not be enough in judging model performance. It should be accompanied by measures that examine spatial correlation. Why? If the predicted field does not exhibit similar spatial correlation as the reference field, their difference (residual signal) would likely deviate from IID (white noise). A white spectrum has uniform noise power and it implies the errors are uncorrelated. A non-white spectrum implies the errors are correlated and there is spatial structure to the noise. RMSE cannot tell us if the errors are systematic (structured) or random. The spatial fidelity measure, however imperfect, at least goes some way toward capturing spatial correlation as it derives from variogram ratios. It may reveal differences that are transparent to the RMSE.

*2. Geostatistical modelling*

Line 81: That's a fundamental viewpoint of random processes, not necessarily of probabilistic models.

[**A28**] Will revise as appropriate.

Line 112: What past experience? This sounds oddly unscientific, why not do hyperparameter tuning based on a cross-validation, as is standard in machine learning (and should be standard in geostatistics but unfortunately isn't always done)?

[**A29**] Melkumyan and Ramos (2011) investigated this in the context of multi-task GP (covariance functions).

Line 130: Indeed, so why introducing kriging and GPs as two different techniques?

[**A30**] As mentioned in [A2], we tried to capture the theory from two equivalent perspectives using notations familiar to the geostatistics and ML communities, following the examples of Oliver et al. (2015) and Williams and Rasmussen (2006). We will be making changes to emphasise the points raised in your first major comment.

Line 162: It's important to mention there that Gaussian simulation is required to preserve the spatial structure of the covariance function (at least away from the data, as the covariance function is part of the prior and will get updated by the data) but not sequential Gaussian simulation. The former can be done through LU decomposition, which implies building a very large covariance matrix, so the later alleviates computational costs by using the chain rule of probability and building instead multiple smaller covariance matrices. And Gaussian simulation refers to a set of techniques to sample from a Gaussian process, so it isn't independent from kriging in that sense.

[**A31**] Will mention this when Sec. 2.3–2.4 are revised.

Line 163: It would be much better to refers to the original study introducing SGS (I suppose that might be Gomez-Hernandez & Journel, 1993, Joint Sequential Simulation of MultiGaussian Fields).

[**A32**] Will expand citations.

Line 180: That definition was already given, since random process, random field, and random function refer to the same thing essentially.

[**A33**] True. Saying they are the same would probably take just as much space though.

Line 190: At this stage we have no explanation of what SK-SGS and OK-SGS mean, so they require further explanations of what they mean and why picking those options specifically.

[**A34**] Thanks. Will add further explanation and mention some of the key differences in terms of how the models were configured. As mentioned in [A5], these details were omitted and deferred to the source code as the modelling process is not the primary focus for papers submitted in the *methods-for-assessment* category [A1d].

Line 190: How do you condition the mean on the local neighborhood in the GP? The previous sections were quite generic, and in the end give little detail as to what you do exactly.

[**A35**] There is a fair argument for making these details clearer, if not only to make the technical description more complete, it can help readers become more aware of the configuration settings that make the models practically different. These details will be added.

Line 192: Why using an isotropic covariance function for kriging and an anisotropic one for the GP? This creates a bias in your experiments, especially for your goal of comparing the performance of kriging and GPs.

[**A36**] As the authors have acknowledged in [A1a], this is a legitimate criticism that would be valid within the jurisdiction of a model evaluation paper. This being a *methods-for-assessment* paper, the goal of adding value and extra dimensions to results comparison [A1c] overrides other objectives. In particular, there is no need to enforce uniformity, requiring models to be symmetrically configured or bias-free in order to be eligible for comparison. Consistent with this understanding, the models can be arbitrarily configured without defeating the purpose of the analysis [A1d]. There is no reason to exclude a model because it is not theoretically optimal [A5]. On the contrary, pragmatic decisions and various approximations are made in practice, thus a realistic scenario we envisaged in which the assessment methods would be used, would likely involve suboptimal (imperfect) models, where modellers intentionally apply different settings or change model configurations to assess the impact of those decisions on different facets of model performance [A4].

Lines 196-199: This is the wrong procedure when dealing with transformed data, you should sample from the predicted distributions (using SGS for instance), back-transform the realizations, then take the mean. Otherwise your predictions will be biased (see Deutsch & Journel, 1997, p.81-82, http://claytonvdeutsch.com/wp-content/uploads/2019/03/GSLIB-Book-Second- Edition.pdf).

Addressed in [A3].

*3. Geostatistical measures*

Line 202: Before defining the metrics used for validation, it would be better to define the validation strategy: how do you assess the generalization error and the uncertainty quantification? Do you use a holdout validation? A cross-validation? Or do you just do some form of residual analysis? It seems that you're doing a mix, but that's not so clear.

[**A37**] We are thinking of bringing forward mentions of $(\hat{\mu}, \hat{\sigma}, \mu_0)$ and discuss these specifically in the context of "future-bench" prediction and how the experiments are set up. Assessment of model performance on unseen data is accomplished essentially using holdout data. Training is done (hyperparameters are learned for each geological domain) using blasthole assay samples (causal data available) from each domain. For testing, separate grid values collected from the "future-bench" (this could be in the surrounding or adjacent area to the current bench, or from the bench below which the modeller does not have access to at the time) is withheld for validation. This latter source of information serves as the ground truth. For reasons given in [A9], k-fold cross-validation is not used.

Line 208: How are the groundtruth histograms define? In a real case we won't have a ground truth, so how can we translate your method to real cases?

[**A38**] Groundtruth histograms describe the distribution of grade values in the test data, these values are defined at the same locations where prediction (probabilistic inference) is required. Assessing model extrapolation performance---in the sense of predicting into a future bench---must be done retrospectively, since this new territory could be underground thus laid hidden before it gets exposed following excavation. In an open-pit mining work flow, different pits are being worked on concurrently, and the "hold-out" data is always plentiful, as assay samples are taken routinely from a portion of drilled production blastholes before blasting and excavation take place. The sanctioned values determined for the current bench (after assaying, integration with expert knowledge and model refinement are done) serves as test data for the previous bench. This cycle repeats when sanctioned values become available for the next bench, it will be used to assess the bench-below predictive performance of the model trained using data from the current bench. The knowledge gained (such as probability calibration error as measured by the goodness of the predictive distribution, and any

surprises such as clusters of large correlated errors) could inform mine planning, and justify changes to measurement density in geologically complex and uncertain areas. Often, the extracted ore material is stockpiled and there is a significant lag time between various mining processes. Provided there is adequate tracking, there is an opportunity to re-characterise the grade and uncertainty in a stockpile before it is transported or mixed with other material downstream. Outside of the mining scenario, the reference (pseudo-ground truth) would have to come from test data set aside for cross-validation. It is possible to observe generalisation performance, albeit in a weaker (more limited) sense, as the test data derives from the same region as the training data. Basically, this evaluates the model's in-situ regression performance. This was demonstrated in Sec. 5.3 of the manuscript in the manner of hold-out validation.

Line 209: Why just with the mean predictions? Why not with the full distributions?

[**A39**] This is what the uncertainty-based measures described in Sec. 3.3 are for.

Lines 210-211: You need to explain exactly why you picked those approaches, and more importantly why you picked four approaches doing essentially the same thing and not just one. What extra insight do we get by using all those?

[**A40**] The quoted passage "the four chosen measures are motivated by hypothesis testing, information theory, set theory and the Monge-Kantorovich optimal transportation / distribution morphing problem" recognises there are different ways of approaching histogram measures depending on one's background. If the authors had chosen just one measure, others might object and accuse us of cherry-picking. Subsequent analysis (Fig. 17) shows these are highly correlated (interchangeable for the most part), however, the Wasserstein EM histogram distance (see Sec. 5.2.1 and Fig.22) tends to be more robust. For a *methods-for-assessment* paper, these findings are important. Just as well, we decided to investigate several approaches and not eliminate some options too early.

Line 242: It's a loss of spatial fidelity only if you assume that a covariance function (or semivariogram model) can be robustly estimated from the data.

Line 246: The range and nugget would be more appropriate parameters to assess a reduction in spatial variability. So why using the sill?

[**A41**] There are caveats and possible alternatives to just about any proposed measure. Applying the logic in the previous remark, the range and nugget may be considered only if they can be robustly estimated. As a general notion, equation (21) considers the variogram ratio over all distances, $d$. It is not the sill that this ratio is confined to. In particular, the median ratio is used in (22). Depending on the choice of the variogram model, the nugget value can be quite variable. It also represents an intrinsic component of variability inherent in the geology that is irreducible. This is something that any model would struggle to predict, and not particularly indicative of the average loss in spatial variability over all distances, $d$.

Equation 22: Is r_model(d) defined anywhere? What is that spatial fidelity supposed to be measuring? And what's its theoretical justification?

[**A42**] r_model(d) is defined in eq. (21). Spatial fidelity measures the reduction in spatial variability in the model predictions relative to the test data based on variograms. We plan on making this clearer in the next revision. As stated in line 242, this proposed measure stems from the widespread use of semi-variograms in geostatistics, which itself is an expression of the covariance between two points. It is basically equivalent to the autocorrelation function which measures the decay in signal power with increasing distance.

Line 256: There we touched upon how to use those metrics in practice, but this is too little too late. We need clearer explanations on the overall validation methodology and its theoretical justification.

[**A43**] This material will be reorganised, with extra explanations added on the overall model assessment strategy in line with [A38], including the objectives as explained in [A1b, A1c].

Line 260: Better to stay consistent with the notation than creating an unnecessary source of potential confusion.

[**A44**] There should be no confusion. We are merely using $(\hat{\mu}, \hat{\sigma}, \mu_0)$ instead of $(\hat{\mu}(\mathbf{x}_{*j}), \hat{\sigma}(\mathbf{x}_{*j}), \mu_0(\mathbf{x}_{*j}))$. We believe this has been used consistently throughout.

Line 305: Why not use the negative log probability of the target under the mode as a metric that can also capture the accuracy of the mean prediction (see Rasmussen & Williams, 2006, p.23, https://gaussianprocess.org/gpml/chapters/RW.pdf)?

[**A45**] Indeed, the negative log probability of the target under the model, $-\log p(y_*|D, \mathbf{x}_*) = \frac{1}{2}\log(2\pi\sigma_*^2) + \frac{(y_* - \bar{f}(\mathbf{x}_*))^2}{2\sigma_*^2}$, as a loss measure is a viable alternative to our synchronicity utility measure ($S$) defined in (24) for evaluating the predictive distribution given the predictive mean and variance. There are couple of reasons for choosing the latter: (1) $S$ is bounded; (2) it may be used to derive other uncertainty-based measures, to compute the goodness of the estimated distribution ($G$) for instance.

Line 314: What happens when you don't have a ground truth and a small, potentially biased validation set?

[**A46**] This was partially addressed in [A38]. These challenges are not unique to a specific assessment method. For the application we envisaged, ground truth is plentiful in an open-pit mining environment subject to time lag, however, the reference points (test data) do not come in equal amounts across all geological domains. If this is what is meant by bias, under-sampling is real. So, it is instructive to examine model performance on all (or at least multiple) domains. If by bias, you mean the withheld data also comes from the same spatial region as the training data, and therefore may not indicate model predictive performance into new territories, then yes, the assessment will be more limited (mainly restricted to in-situ regression performance), if test data from the bench-below is not available to assess forward prediction capability.

Line 333: "Model 1 is simulated using a uniform distribution": What does that mean exactly?

[**A47**] In Fig. 3, the smooth curve represents the true target value, $\mu_0(\mathbf{x}_*)$. As explained in line 332-334, "in blue and black, we have two noisy [synthetic] models. They differ in terms of how much their mean predictions gravitate toward the actual mean, $\mu_0(\mathbf{x}_*)$. Model 1 is simulated using a uniform distribution, so its estimated means are more spread out. Model 2 is simulated from a normal distribution, so its mean predictions, $\hat{\mu}(\mathbf{x}_*)$ tend to be concentrated around $\mu_0(\mathbf{x}_*)$ but its tail values extend further out." In other words, for model 1, $\hat{\mu}(\mathbf{x}_*)$ is drawn from a uniform distribution specified by $\mu_0(\mathbf{x}_*) + U([-b/2, b/2])$. For model 2, $\hat{\mu}(\mathbf{x}_*)$ is drawn from a Gaussian, specified by $\mu_0(\mathbf{x}_*) + bias + const \times N(0,1)$.

Figure 3: The figures are hard to read unless one zoom in a lot. It would be better if the font size of the figure was closer to the font size of the text. Also shouldn't there be an uncertainty estimate?

[**A48**] We will endeavour to increase the font size for legibility. In this synthetic example, as mentioned in line 330-331, each vertical bar signifies $\pm\hat{\sigma}$. This length is somewhat arbitrary; its purpose is to emphasize that we have a predictive distribution. The current choice, $[\hat{\mu}-\hat{\sigma}, \hat{\mu}+\hat{\sigma}]$, corresponds to $[Q_{(1-p)/2}, Q_{(1+p)/2}]$ where $p \approx 0.68$.

Line 340: So here you're doing a residual analysis, is that correct? That doesn't tell you anything about generalization error and about the ability of the models to extrapolate though, and the ability of your metrics to reflect that.

[**A49**] Here we are evaluating the predictive distributions using uncertainty-based measures. Since the predictive mean and variance values are contrived (they are synthesised rather than a product of some real modelling process), generalisation performance and the ability to extrapolate are irrelevant. The point is to show if we have a biased (insert other adjectives), conservative, or over-confident model, that captures more or fewer true target values in a given probability interval---which we can quite easily see based on the proportion of overlap of the bars with the target function (smooth curve), we can check these behaviours and see they are reflected in these measures. The aim is to show the numbers (e.g., interval tightness and precision/goodness of the predictive distribution) are in accord with intuition.

Table 2: Having so many non-standard metrics is just counterproductive in my opinion, because figuring out what each one means gets complicated, so comparing the models gets complicated. Having just two or maybe three well-chosen metrics would be much more efficient.

[**A50**] We appreciate this comment and understand where this reviewer is coming from. Where our views probably differ is that we are less interested in establishing the superiority of models per se, and more interested in the methods for assessing models, which is the expectation for our submission category after all. This gives greater scope for considering the accuracy, precision, goodness metrics described by Deutsch et al. In the end, using a subset of these might be sufficient, we won't know if we don't find out. As mentioned earlier, in relation to the histogram distances, the Wasserstein EM distance is shown to be more robust than several alternatives such as the Jensen-Shannon divergence. Returning to the uncertainty measures, the synchronicity measure has the added advantage of showing over- and under-estimation and can be rendered as maps. This is useful in grade model spatial comparison.

*4. Experiments*

line 374: So here you're looking at extrapolation, although the previous synthetic example wasn't. Being consistent between the synthetic case study where we have a ground truth for comparison and the real case study would make the analysis of the proposed method more robust.

[**A51**] In the synthetic example, the estimated values were contrived. The attention was purely on $(\hat{\mu}, \hat{\sigma}, \mu_0)$ and the uncertainty-based statistics. It would not have mattered whether it represents extrapolation or not, as no actual training data was used by the fictitious models. The uncertainty-based statistics are used consistently across the board in all subsequent experiments. The only difference is that the ground truths (hold-out test data, $\mu_0(\mathbf{x}_*)$) are given at inference locations $\mathbf{x}_*$ that require non-trivial extrapolation in the real experiments (typically projecting into adjacent benches or the bench below), whereas in the synthetic example, we have perfect knowledge of $\mu_0(\mathbf{x}_*)$---the target values on a smooth curve.

Line 383: I would add a link to the repository here.

[**A52**] An URL to the repository was included in the initial submission. The authors were subsequently asked by editorial staff to remove it to comply with journal regulations.

*5. Results*

[**A53**] As a general comment, we intend to revise Sec. 5.1 Analysis 1 with a view of simplifying or removing some of the illustrations that are deemed obvious.

Line 412: That part is incredibly confusing, because it states basic properties of simulations (the more simulations, the closer to the kriging/GP predictions), but also suggest that fewer simulations get you closer to the ground truth, which is wrong, this is just an artifact from using too few simulations.

[**A54**] This criticism relates to observations and remarks that pertain to Fig. 7. We do not believe this is a fair or accurate representation of what the text is actually saying. The principal reasons for conducting conditional simulations according to (Deutsch and Journel, 1997, p.4) are to (1) obtain uncertainty estimates of the field at unsampled locations altogether in space; (2) address the issue of oversmoothing seen in kriging/GP (Olea and Pawlowsky, 1996); and (3) reproduce the histogram and restore spatial correlation between samples and inference grid points (Ortiz, 2020). Our first comment in relation to the smoothing seen in SK and GP(G) is a reflection of (2). Our second comment that "simulations can improve the spatial fidelity of the predictions" in the sense that the spatial variability matches more closely with the groundtruth is merely a restatement of (3).

If we take "closeness to the groundtruth" to mean global accuracy as measured by histogram distance, then certainly we expect this error to fall in line with baseline GP for sequential simulations upon convergence since it honours the data and the disturbances have zero mean (this can be seen in the first column of Table 15). If we consider the spatial fidelity of the predictions in the sense defined in (3), then sequential simulations does improve local accuracy and help replicate the spatial variability observed in the test data (groundtruth). The second column of Table 15 provides evidence of this. We do not dispute that the simulated results would contain significant fluctuations which this reviewer refers to as artefacts if one relies on using a few random realisations. As the accuracy column in Table 15 shows, this does not make good sense (help ensure the uncertainty estimates are sound). Any suggestion of the benefits is limited to the extent that sequential simulation injects correlated noise that helps minimise over-smoothing inherent in kriging/GP, and the effects can be seen early even after a few simulations.

J.M. Ortiz, Introduction to sequential Gaussian simulation, Queen's University, Annual Report, Queen's University, Ontario, Canada, 2020. URL: https://qspace.library.queensu.ca/server/api/core/bitstreams/3f7458da-0662-46be-822b-974c4094ea46/content

Figures 7 & 9: The problem here is that the first row is the wrong way of proceeding, while the bottom row is the right way. But this is nothing new, it's basic geostatistical knowledge (see Deutsch & Journel, 1997, p.81-82, http://claytonvdeutsch.com/wp-content/uploads/2019/03/GSLIB-Book-Second- Edition.pdf). I suppose "from_32" means from 32 realizations, but in general you need a few hundred realizations to have a robust estimate of the mean, and a few thousand for the standard deviation.

[**A55a**] We appreciate this reviewer is well versed in geostatistics. As authors, we have a responsibility to cater for the needs of a wider audience. Following the mathematically heavy sections, these figures illustrate the predictive distribution in a mining geology context, and help readers comprehend spatial variability and over-smoothing, and visualise differences between different models/configurations.

[**A55b**] We respect this reviewer's comment. We would rather refrain from casting judgment, as it is unhelpful for understanding the rationale of this work [A1a-A1d]. For a *methods-of-assessment* paper, there is no reason to exclude a model simply because it is suboptimal. Seen in their true light, the differences that exist between models may reflect pragmatic decisions, hyper-parameters tuning, various approximations, or model settings that modellers might apply intentionally in practice to investigate the impact of those decisions on different facets of model performance [A36]. If a few hundred/thousands of realisations is needed to obtain a very reliable estimate of the target standard deviation, it would be a fair question to ask can we get a reasonable (probabilistic sound) estimate with fewer realisations? The configuration changes between models are made in this spirit to reflect what

might happen in a real-world model assessment scenario, and such variations (not strictly limited to number of realisations) is what the vertical axis in many of the plots (in Fig. 22-26) truly represent.

Figures 8 & 10: I'm not sure what is the point of those figures, they illustrate a very basic property of simulations.

Answered in [A55] above.

Line 421: Estimation of the variance in GP is the same as in kriging, so it isn't based on the observed values either. Yet there is a difference between the uncertainty quantification from kriging and from the GP, why is that?

[**A56**] This most likely is due to different neighbourhood definitions used in kriging and GP, respectively. The former (based on the code we inherited) uses isotropic covariance functions and spherical nearest neighbour search, whereas the latter uses a kernel with separate length scale parameters for each axis and an ellipsoid search.

Line 425: Any heterogeneity you see there comes from the insufficient number of realizations that you're using. It's a bias, not a feature.

[**A57**] Graininess is an artefact when the number of realisations is small. This is a distraction if anything. The substantial point we try to make is that the kriging variance measures only the uncertainty prevailing at a given location $\mathbf{u}$; it does not provide any indication about the joint uncertainty prevailing at several locations (Deutsche and Journel, 1997, p.19).

To elaborate, the kriging/GP variance, being independent of data values, largely reflects the geometric data configuration, not the local estimation accuracy (Journel, 1986). Conditional simulation effectively trades off minimum estimation variance for the reproduction of the covariance to restore spatial variability. Under the multivariate Gaussian assumption, the simple kriging mean and variance convey the same information as the posterior ccdf (cumulative distribution function conditioned on an information set of neighbouring data). This univariate ccdf "measures only the uncertainty prevailing at a given location $\mathbf{u}$; it does not provide any indication about the joint uncertainty prevailing at several locations" (Deutsch and Journel, 1997, p.19). The significance is that we can make probabilistic inference for block averages not exceeding a threshold within some spatial support ($B$) using simulation/stochastic images, but this cannot be derived from the set of univariate ccdfs by considering multiple points $\{\mathbf{u}_j\}$ in $B$.

Line 434: At this stage it remains unclear to me what does it mean to couple SGS or CRF with a GP. Are you sampling from the GP distributions using SGS or CRF?

[**A58**] Our approach is consistent with (Deutsch and Journel, 1997, p.126). For each random realisation $s$, we generate a sequence of simulated values $\{z_*^{(s)}(\mathbf{u}_i), i = 1, \ldots, m\}$ by sampling from a univariate ccdf $F(\mathbf{u}_i; z|(n + i - 1))$ using the posterior mean and variance estimates from kriging/GP for Z($\mathbf{u}_i$). The conditional implies using $n$ data points plus ($i$-1) previously simulated values. In practice, a spatial neighbourhood is imposed to mitigate computation cost. These details will be made clear in the revised manuscript as hinted in [A5].

Table 4: This is quite hard to understand in the end. Some kind of bar plot would already be an improvement.

[**A59**] We will reassess the need for a bar graph. Overall, readers are not expected to scrutinise individual numbers, as the histogram distances are already sorted in the 'rank' column. For guidance, the main point is summarised in line 432-433: although the trend varies somewhat depending on the

domain, a common observation is a general improvement in the histogram rank (reduction in histogram distances) when simulation (SGS/CRF) is used.

Line 447: How did you infer the semivariogram model parameters though? That needs to be explained in the method section.

**[A60]** As indicated in eq. (21), variograms are computed for the model and reference using the posterior mean function evaluated at the predicted locations $\{\mathbf{x}_*\}$ and the unseen test data (groundtruth for the future-bench), respectively. The SciKit package `skgstat.Variogram` is used to fit the semi-variograms. The variogram model and parameters are user configurable, and the Matheron estimator and Matern model are used by default.

Line 460: This is a basic property of kriging and GPs. You will observe the same for the simulations if only you simulate enough realizations. The only difference will be that you'll do the back-transformation correctly, which should impact the results, although I'm not sure to what degree.

**[A61]** We agree that this is what conditional simulations are meant to do. In regard to the second part on back-transformation, we believe this has been discussed in [A3].

Line 469: A key question is do you need to do so? Reproducing semivariogram models per se isn't particularly valuable, since a model can be wrong anyway. Assessing generalization error is much more valuable.

**[A62]** In mining geology modelling, global accuracy (e.g. kriging reproduction of global trend) is not the only aspect that people would care about. Geologists are particularly sensitive to texture (i.e. patterns of local variability) in model appraisal; hence a characterisation of how well spatial variability is preserved (using variograms to measure spatial correlation) matters a lot. By design, all our experiments consider the generalisation error (e.g. comparing the histogram distance between the predicted values and test data). Thus, we can detect over-fitting. A more important question is what constitutes the test data. If it comes from the same region as the training data, we are assessing in-situ regression performance. This is not the best indication of forward inference capability where a model is required to predict into future benches, so this is where the test data should be sourced [A38].

Line 475: What do you mean by "32 iterations"? I've assumed realizations so far, but now I'm confused. If it's indeed realizations, then you shouldn't cherry pick a number of realizations because you get a better value on your metrics: more realizations mean a more accurate approximation of the full predicted distribution, that predicted distribution might be wrong, but that issue won't be solved by sampling less.

**[A63]** We generated 128 realisations and considered the prefix-set based on the first 32 realisations. As a *methods-for-assessment* paper, we are not casting judgement on whether one way is better or worse than another, or dictating what people should do. All model enquiries are valid. If a modeller wants to limit their simulation to $s=N_S$ realisations, and find out how well spatial variability is reproduced with respect to the test data, they can observe how spatial fidelity ($F$) varies with $s$, without inspecting stochastic images manually. Here, $F$ is computed for the ensemble average of the first $s$ realisations for a given $s$ as all instances are treated as equal.

Correction: lines 472 and 474 should be referencing Table 5 not Table 6.

Table 6: This is also tricky to decipher. In the end figure 19 is much easier to interpret.

**[A64]** Will reassess usefulness of the table and the comments around it in the next revision.

Line 505 and figures 20 & 21: I don't see what's so special about the synchronicity. In the end you could get the same insights from a map of mean error.

[**A65**] The mean error is fine for single analysis, if the evaluation is restricted to a specific variable at a specific site. If the interest expands to multiple target variables (e.g. applying the same modelling technique to predict Cu, Zn etc.) and at several different locations, it would be not be meaningful to average across them, or compare a mean error of -0.5 for A vs -0.05 for B. Plus, the synchronicity is computed (for free) en route to evaluating uncertainty-based measures, such as the goodness of a predictive distribution. So, it indicates the local bias and has a consistent probabilistic interpretation that depends on the prediction-interval.

Figure 22: I would put the two subfigures in a single column and increase their size, including font size. Otherwise readability is quite low. It took me some time to figure out what the labels on the horizontal axis meant, I would suggest to move the axis labels to the left-hand side of the axis, with "inference period" above "domain label".

[**A66**] Layout is more a matter for the publication office. Regarding font size, the text is probably as large as it can be to fit within the space available. Regarding moving x-label to the left-hand side, the concern is that it will get confused with the y-labels, but I will consult with my co-authors. Most readers can magnify figures at will from a PDF.

Figure 23: I get what you want to achieve with the dotted cells, but they stand out so much that they attract attention too much, whereas you want people to still focus on the values of your metrics. Maybe filling out the cells in light grey would work better?

[**A67**] Time permitting, this will be changed as suggested in the revised manuscript.

Line 677: A reasonable number of realizations is as high as possible, since your approximation of the full distribution improves with the number of realizations.

[**A68**] That would be ideal, but in many instances impractical. A more pertinent question is "can we get a reasonable (probabilistic sound) estimate with fewer realisations?"

Line 680: If you didn't use a neighborhood in the SGS, then this is just what theory predicts, hence why it has received little attention.

Noted

Line 694: This is why people in statistics and machine learning usually use R2. What Deutsch called accuracy and precision is the same as the expected calibration error in machine learning. While you're right that this isn't as used as it should be, this isn't something new.

Noted

What I'm missing at this point is actual proof that your proposed FLAGSHIP gives us more insight than those standard metrics.

[**A69**] As discussed in [A10], proving these measures are better than others is not the goal of this study. We are not suggesting a replacement, merely a suite of measures that would be fit-for-purpose for large scale studies. In the field of mining geology, we have not seen probabilistic models being evaluated simultaneously with respect to global accuracy, spatial variability and calibration errors (in the ML sense) in a systematic and automated way. The FLAGSHIP measures help shed light on these aspects, but we acknowledge alternatives such as $R^2$ exist. Beyond summation statistics, the

synchronicity measure can produce maps to identify local bias which is prevalent in grade modelling for reasons mentioned in [A26] and the image-based visualisation can help focus attention on situations where models misbehave. These require the measures to be standardised and interpretable independent of modelling conditions.

Line 703: The medical literature has shown that hypothesis testing isn't as robust as we'd like to think, and p-hacking is real. In the end plotting the mean value of your metric and the standard error for each model, and looking at overlap between the standard errors of different models, is a simpler but no-less robust way of assessing the value of each model.

[A70] Yes, we agree. That's why we included a t-test CI to show the overlap in Table 13.

*6. Discussion*

Table 16: It's still not clear how your procedure relates to the best practices in statistics, machine learning/data science, and geostatistics. The absence of any literature on those in the entire manuscript is really problematic. From that perspective, the discussion isn't really a discussion (there's no comparison to what has been suggested in the literature, nor discussions on the strength and limitations of your approach and of your study, nor perspectives for improvement or future work) but more a conclusion. I'm also missing a discussion on kriging vs GP, although that was mentioned as an objective of this work.

[A71] We believe a lot of the confusion stems from not having a section that provides a focused discussion of the motivation, objectives and application context upfront. The open-ended nature of the manuscript had drawn a lot of criticisms that could have been avoided. Based on this reviewer's opening remarks, the authors have reflected on the goals of this study and how best to achieve them. In the revised manuscript, we have reframed this paper and emphasised the actual purpose of this work. The first objective is to develop an approach that supports highly automated model assessment at scale, focusing on three aspects and using standardised measures that can be meaningfully compared across domains and target variables. The second objective is to provide a more complete understanding on different facets of model performance—this includes identifying situations where models misbehave, visualising error clusters, and testing for statistical significance between models. Both the introduction and discussion sections have been rewritten to make this study more coherent and cognisant of other possibilities mentioned in the literature. The section on kriging/GP has been rewritten to rectify issues this reviewer has identified. It is worth reiterating that the real focus is on the methods of assessment, not establishing which model (implementation or configuration) is best. We have eliminated any such suggestion in the revised manuscript so that readers can see this study in its true light. — Return to section [A1]

*7. Conclusions*

Line 735: That's highly debatable. Personally, I find that your approach has too many metrics, which blurs the analysis and leads to less clarity.

Lines 736-737: Then why using those metrics? How are they helping to get clarity if they can be biased and another, single metric can do better?

[A72] We feel there is a misapprehension of what this paper is about. The ultimate goal is not to establish which model works best by taking the shortest path, rather the interest is in using different measures, modalities and adding extra dimensions to the analysis to compile a more complete view of model performance, irrespective of what the modelling processes are used. Examining several methods for assessment is actually the point, we won't know, for instance, which histogram distance (or global accuracy) measure is more robust if we don't find out.

Lines 745-746: You're mixing things up here, the sequential scheme won't improve anything, the final predictions will be the same (which is exactly what the chain rule is saying).

[**A73**] We thank the reviewer for the observation and agree that a large number of simulations will recover the overall statistics in the case of a global kriging/GP. However, in the case of a local kriging/GP (where the estimation at each point depends only on its own neighbourhood) the non-linearities arising from the local neighbourhood will result in the SGS and GP(L) having different statistics over a large number of simulations. We are happy to produce an example to demonstrate this point if the reviewer considers it beneficial.

Line 759: You would need a more robust analysis to make such a general conclusion. You can only say it's true for your particular case study and setting.

[**A74**] Agree. This will be revised.

---

## Author Comment (AC2)

Following the handling editor's advice, the authors will respond to the key concerns raised in this review, particularly the fundamental issues that have been highlighted, rather than engaging with every minor point at this stage. Dr. Poulet has instructed us to consider the overarching critiques and how they impact the manuscript's core contributions.

**Reviewer 2's comments**
https://doi.org/10.5194/egusphere-2024-4051-RC2

**RC1**: 'Comment on egusphere-2024-4051', Anonymous Referee #2, 19 Mar 2025

This manuscript proposes a suit of probabilistic prediction validation measures named "FLAGSHIP" intended for use in evaluation of interpolations in the mining industry.

The presentation and writing are both very polished and I personally could not find any spelling or grammatical errors. There is a significant number of experiments on both real and synthetic data which are all adequately documented. I think some of the concepts here, such as comparing histograms and variograms, have promise, but the execution is poor. Overall the work lacks a principled approach to justifying performance metrics, the experiments are not set up in a way that can lead to insights, there are major theoretical mistakes, and the literature review on predictive performance metrics and paradigms is lacking.

[**B0**] The authors would like to thank this reviewer for the constructive comments and highlighting the positive aspects of this manuscript. In reference to poor execution, we acknowledge that we have not been sufficiently clear when it comes to stating the motivations and principal objectives of this study. Because the actual purpose of the study was not clearly articulated, it was open to interpretation and potential misunderstanding. This, along with the criticism that it "lacks a principled approach" and "the experiments are not set up in a way that can lead to insights" can be addressed by reflecting on the goals, and providing better guidance to manage readers' expectation. Indeed, work was already underway to reframe and restructure the manuscript, to address similar concerns (esp. clarity) from reviewer 1 before this critique was received. We note in particular that the issues regarding kriging/GP have been rectified, and new references have been added in the revised manuscript to strengthen the background section and discuss other approaches/paradigms in relation to this work.

[**B0a**] There are a few points we need to clarify to correct any misconception. First, this study focuses mainly on evaluating extrapolation performance (not so much interpolation) where models are required to "forward predict" into new territories. The application context and implications for the test data and validation approach are explained in the rewritten introduction. Second, this manuscript was submitted as a *methods-for-assessment* paper rather than a *models evaluation* paper. This distinction is important as it sets the tone and expectations for what is to come. Finally, many of the issues identified in the major comments have been addressed in the revised manuscript; see also claims refuted in [B4].

[**B0b**] To elaborate on the second point – According to the journal guidelines, it should describe "new standard experiments for assessing model performance or novel ways of comparing model results with observational data". Instead of comparing models to determine which is superior, the emphasis is on developing a balanced approach that adds value and insights to the analysis. In this study, we seek to develop a more complete approach that (a) meets the relevant requirements (illuminating on the three pillars of performance – global accuracy, local accuracy/spatial variability, and calibration properties of the predictive distributions for ore grade estimation) and (b) supports highly automated model assessment at scale, using standardised and interpretable measures that can be meaningfully compared across domains and target variables. An associated objective is to provide a richer understanding on different facets of model performance—this includes identifying situations where models misbehave, visualising error clusters, and testing for statistical significance between models. Regarding insights, this can come from functional enhancement, e.g., what information/patterns users are able to access, observe and interpret; not merely lessons or conclusions that can be drawn on specific models.

[**B0c**] In relation to "justifying performance metrics", the revised introduction makes clear some of the deficiencies in current practice. In Sec. 3.2, we have given further reasons for why a spatial fidelity measure is needed. It is worth pointing out that we are not suggesting the proposed measures are necessarily better than any existing alternatives; they are merely fit-for-purpose.

To do this topic justice I advise that the authors do more focused work on a smaller subset of measures and analyze them more thoroughly. I also suggest not focusing on the krigging vs GP comparison, but to instead compare variogram/kernel model choices and fitting methods withing each paradigm separately.

[**B0d**] We thank the reviewer for the advice. In relation to the first part, pursuant to [B0c] we are not arguing that the suggested measures must be used to the exclusion of everything else. In the new discussion section, we have mentioned a few potential alternatives, and indeed one can fairly represent the three pillars of performance [B0b] using a subset of the FLAGSHIP measures. Had we started with a much "smaller subset of measures" in the beginning, we would not have discovered the Wasserstein EM measure is more robust than the Jensen-Shannon histogram distance, for instance, and would subject ourselves to accusation of selection bias or cherry-picking.

[**B0e**] Regarding "not focusing on kriging vs GP comparison", we agree that it may be unhelpful reading too much into any findings regarding kriging or GP. The study should still stand irrespective of the chosen models as far as the objectives in [B0b] are concerned. It is not about what conclusions can be drawn for specific models, rather it is about the kind of observations or insights one can attain from having a richer set of assessment tools at one's disposal. We added a new section (Sec. 2.4) to clarify that it is not just differences in hyper-parameter estimation per se, but differences in kernel and neighbourhood definitions also contribute to differences between the kriging/GP models (paradigms).

This is a *methods-for-assessment* paper after all, so it would be helpful to recognise these differences for what they truly represent, as an embodiment of parameter/ configuration changes that reflect how modellers might explore different modelling options in practice. The authors also feel that shifting the focus to the modelling mechanisms (comparing variogram/kernel choices etc. beyond what is covered in Sec. 2.4) would be counterproductive as the study would stray from its original objectives [B0b].

Major Comments:

I am recommending that this work be rejected for several reasons, listed here in order of importance:

1. The experiments lack purpose. For any experiment, including computer algorithm experiments, there needs to be some prior concept of what the potential outcomes are and what different conclusions would be drawn in each case. In this manuscript we are simply presented with different measures applied to different interpolations and are told one method performs better than anther. Some principle needs to be defined for how metrics are assessed and what the experiments are meant to contribute to our understanding.

[**B1**] We acknowledge the lack of clarity has been problematic as it obfuscates the purpose of this study. The manuscript has been thoroughly revised to make its objectives much clearer (as mentioned in [B0]). To prepare readers with "some prior concept of what the potential outcomes are", we have reinforced what we seek to achieve in the introduction (Sec. 1, last paragraph) and Sec. 5 (first paragraph) before the results was presented. Further guidance is provided in the discussion (Sec. 6).

On the suggestion of aimless meandering, the measures are organised based on three principles and grouped based on three categories: the histogram, variogram and uncertainty-based measures each targets one aspect of performance, viz., the global accuracy, local accuracy (spatial variability) and calibration properties of various model predictive distributions. Regarding "[being] told one method

performs better than another], the aim is to illustrate fitness-for-purpose in the context of grade modelling for forward extrapolation in mineral deposits; not that one measure outperforms another.

For a *methods-for-assessment* paper, the goal is less about drawing conclusions for specific models, it is more about demonstrating the kind of insights one can attain from having a richer set of assessment tools at one's disposal. Regarding "what the experiments [or results] are meant to contribute to our understanding", we have now clarified what outcomes the readers can expect in line 553-556 (highlighted in paragraph 3 in Sec. 5 of the "diff").

2. There is no theoretical justification of the metrics proposed. Performance metrics are supposed to be abstract proxies for what is valued in the real world (e.g., cost of recovering minerals). There is no discussion here of how the metrics relate to the real world setting or objective.

[**B2**] The principal motivation (or empirical justification) is that the three main aspects of performance (see B1) are seldom jointly investigated and evaluated in ways that are amenable to large-scale automated processing. The measures considered in this work would support this and provide meaningful cross-site comparisons irrespective of the domain and target variable.

The variogram-based spatial fidelity measure (F) acts as a proxy for spatial variability. It can detect over-smoothing which geologists particularly object to when evaluating ore grade models. This is discussed in Sec. 3.2. More generally, the utility of these metrics, i.e., how these relate to the application setting or objectives is considered in Sec. 1 (paragraph 3).

3. The literature review on predictive performance measures is inadequate. There is an enormous amount published on this topic. The authors need to broaden their search outside of goescience and geostatistics. Consider literature in machine learning, statistics, Bayesian methods, meta-science, and philosophy of science. For things specific to spatial predictions, I know there is a lot in environmental science.

[**B3**] The literature survey has been strengthened, and relevant works are discussed in relation to this study in Sec. 1 (paragraph 1). We strived to strike the right balance cognisant of the fact that (a) this is not actually a survey article; (b) not all approaches (e.g. k-fold cross validation) are necessarily well suited to open-pit mining. For instance, partitioning the data and setting aside test points that come from the same region as the training points has its limitations. It is not going to set up a situation representative of "future-bench prediction" and will bias the results when forward extrapolation performance is of interest. The performance gap is considered in Sec. 5.3.

4. Trading off different metrics is not discussed. Some of these measures (e.g. accuracy) can be trivially maximized by simply making the prediction standard deviation as wide as possible. Others (e.g.fidelity) can be maximized by over-fitting. How are the different properties to be traded off against each other? The problem is not even acknowledged. There are plenty of existing metrics, such as cross-entropy, which have the tradeoff built in. I get that you may want to measure parts of the objective separately, but a validation framework that does not even acknowledge the trade-off issue can only mislead.

[**B4**] This is **not true**. In lines 445 and 456 (revised manuscript), we mentioned that "in general, both $G$ [goodness] and $I$ [interval tightness] need to be taken in account when assessing probabilistic models, because uncertainty cannot be artificially reduced at the expense of accuracy." This is what the $I$ metric is for. In relation to spatial fidelity $F$, we acknowledged in line 389 that "a ratio that increases far beyond 1 is also undesirable as it signifies noise amplification." Hence, the expression $F(R)$ in eq. 22 penalises absolute deviation from 1, particularly the case where $R > 1$ which represents over-fitting. As an aside, over-fitting can be a problem in neural networks, but it is not prevalent in kriging and GP regression as they behave like weighted moving average filters.

5. The distinction between a predicted distribution and a sample from such a distribution is not reflected in the experiments. For example, one cannot simply compare the mean image obtained from OK directly with a single sample or finite sample set obtained from OK_SGS from_n (where n is small) directly, as one is a distribution and the other a realization (or set of realizations) from that distribution. Similarly, comparing a real data histogram to a histogram of a predicted mean is not meaningful. The same also applies to variogram comparison.

[**B5**] We appreciate the difference between a distribution and a realisation. This comment highlights the risk of misapprehension and not seeing the forest for the trees. The general applicability of the assessment methods (e.g. image-based visualisation of various statistics in Figs. 18-22) is NOT dependent on what models are used. It would not have mattered if the vertical axes in Figs. 18-22 are replaced with dummy labels. Although the vertical axis currently portrays variation in the number of simulation runs, it is meant to represent more broadly any modelling approaches, configuration changes (including hyperparameters tuning) that modellers might want to investigate, to assess the impact of various approximations/decisions on predictive performance. This is now mentioned in the discussion (around line 847 in the "diff"). In this sense, there is no practical limit to what users might choose to compare, whether the model interpretation is theoretically sound or not. In practice, there is good reason for not keeping hundreds or thousands of simulation runs around (especially when modelling is done at scale), one reductionist approach is to retain only the mean and variance. So, the mean and stdev are all they have at the end of the day. It would be impractical to insist users would have to regenerate hundreds of simulations again, in order to examine the cumulative distribution function of some statistic (e.g. goodness) across $S$ simulations. It would not meet the ease-of-use criterion. It should not be misconstrued that a single (or several) realisations are on an equal footing with a proper simulation. That is not the point of these illustrations. The collection of models represents a metaphor for arbitrary variations that give rise to the different models that user would like to compare and contrast.

6. The first reviewer has already detailed the equivalence between kriging and GPs. I will agree here that the comparing them is effectively a comparison of how the variogram is fitted and how the posterior distribution is calculated/approximated. I believe much of the conclusions draw comparing these two methods are over-generalizations resulting from this lack of theoretical understanding.

[**B6**] This has been addressed elsewhere. The issues have been fixed in Sec. 2, and the observations are implementation dependent not universal (this is now indicated in the conclusion).

Specific Comments:

Lines 41: Kriging also provides a predicted mean and variance and the covariance can be calculated as well. The apparent over-smoothing is likely due to specifics of how the variograms are fitted and will be sensitive to details of how hyper parameters are fit. Note, for both kriging and GPs there are multiple methods.

Yes, absolutely.

Line 160: [*] The statement that kriging does not reproduce variability between pairs of test points seems wrong to me. [**] A mean predicted image will necessarily be smoother than the true image. The roughness of the mean prediction should not be directly interpreted as the expected roughness of the truth. The variance between two test points can be calculated simply by applying the variogram to their relative lag, thus any bias towards underestimating variance would be due to the variogram fitting process or its restrictions (e.g. stationarity and isotropy), or due to the true process being significantly non-Gaussian.

Re: [*] The kriging variance is estimated at two predicted locations (test points) independent of each other, with no spatial continuity constraint imposed between the two. This is mentioned in Prof. Michael Pyrcz's video https://www.youtube.com/watch?v=3cLqK3lR56Y&t=13m35s

Re: [**] We agree the roughness of the mean prediction should not be directly interpreted as the expected roughness of the truth. From the point of view of a *methods-for-assessment* paper, to facilitate automated evaluation at scale, the proposed methods should be model-agnostic. All model comparisons are fair game. It is not always possible to know what modelling techniques are used to create a model (proprietary models created using third party software often behave like black-boxes). Often, one only has access to just the mean prediction (and the variance estimate if lucky). So, it is a fair question to ask how different are a bunch of models in terms of spatial variability. The end-use ranges from visual interpretations to mine planning, including probabilistic inference and potentially stochastic optimisation if uncertainties are available. So, for some use cases over-smoothing matters.

Line 180: This definition is for a finite field. In general, spatial processes are defined over an infinite number of variables.

Line 191: Both kriging and GP can use isotropic or non-isotropic variograms/kernels. The way it is written here suggest that these are limitations of the methods and not of the specific implementations.

You are right. The revised manuscript now portrays these as specific implementation decisions.

Line 248: Fidelity feels like the wrong word here given that accuracy has no effect on it. Also this measure can be easily maximized by simply over-fitting thus there needs to be some discussion on how it is to be traded-off with other measures.

Line 264: This sentence makes no sense to me. What does "once the validation measure is revealed" mean? Conditioning is done on random variable outcomes, not measure types. What does "likelihood of that the model is correct mean", likelihood is the probability of data given a model as a function of the model, the shaded area is proportional to the probability of data being in that interval given the model as an assumption.

We acknowledge this was poorly written, and the interpretation of "observation likelihood given a model" is entrenched in Bayesian reasoning. In the revised manuscript, we have replaced *likelihood* with *local consensus* to reflect the consensus between model prediction and true measurement.

Line 267: The rational behind 'S' as a measure is not clear to me at all and needs elaboration.

Line 271: Likelihood is the probability density of the known data assuming the candidate model. Why is it defined as cumulative density here?

Section 3.3: I think the kappa statistics are interesting but without discussion on how they can be traded-off against each-other there is no clear way to use them. There is no discussion on what it is you are looking for from them. There is no mention of the fact that some can be easily maximized by arbitrarily over-fitting or under-fitting predictions.

As mentioned in [B4], there is an interval tightness measure to restrain (or indicate) over-fitting.

Line 418: Estimated variance does depend on the data as the variogram is estimated from it. The uniformity you see is due to the regular spacing of the data used combined with the stationarity assumption. Questions relating legitimacy should be about the stationarity assumption, which is not strictly needed for kriging or GPs.

We agree with the points regarding "regular spacing" and "stationarity". As the paper is submitted to the category of *methods-for-assessment*, we do not intend to delve more deeply into modelling processes and questions, as these take attention away from the core objectives expressed in [B0b].

Analysis of the appropriateness of the stationarity assumption should precede the interpolation.

The observed higher spatial variability of GP over SK and OK are simply due to it inferring a different variogram with higher variance at short lags.

I do not understand the logic in comparing nst with SGS or CRF.
To illustrate over-smoothing in kriging/GP and contrast with SGS/CFR; putting this beyond words.

Variance estimates from single samples are not comparable to whole posterior distributions. The apparent differences are due to misinterpreting what they produce.
Figures removed.

Line 430:  The ground truth has thicker tails because it is a realization of the random variable which is being compared here to mean predictions. Again, the mistake here is to expect distribution means to have the same properties as realizations from those distributions. The correct thing to do here would be to convolve each prediction mean with its predicted standard deviation Gaussian kernel to obtain a correct posterior expected histogram.

This viewpoint is interesting and different. Essentially, it treats each prediction as a Gaussian distribution centred at the predicted mean, and takes a sum of these Gaussians over all test locations to produce the posterior expected histogram. This concept is now mentioned in the discussion in the revised manuscript.

Sections 5.1.2:  Again, properties of distributions are being compared to properties of samples. SK_SGS_single is a sample from the SK distributions. Their variograms are not comparable. One could compare the SGS_single variogram with the ground truths, or the SK_SGS_from_highestnumberthatispractical with SK, but not across those groups.

We appreciate this point from a theoretical perspective. However, the techniques used to produce the models are not always known or documented in practice, so it's not possible to treat models differently or decline comparison on these grounds. That is why our methods are completely model-agnostic.

More broadly, the variation in the number of simulation runs (see vertical axis in Fig. 22) is meant to represent arbitrary changes in the model parameters or configuration. So, it would not be particularly helpful seizing on one embodiment of the idea, as users are free to choose any model they want and the context could change completely.

I am leaving out my notes for the remaining sections because they are all about the same point: properties of distributions and samples from those distributions should not be expected to be the same and are not directly comparable.

---

## Author Comment (AC3)

**Revised GMD manuscript: List of changes for egusphere-2024-4051 (March 31, 2025)**

**Evaluating uncertainty and predictive performance of probabilistic models
devised for grade estimation in a porphyry copper deposit**

**Raymond Leung, Alexander Lowe, Arman Melkumyan**

‡ Section and page references for each item are hyperlink enabled.

| | Section | page | line | Notes |
|---|---|---|---|---|
| 1 | Abstract | 1 | | Minor changes (word choices). Rephrasing to avoid ambiguous interpretation. |
| 2 | §1 Intro | 1-2 | 21-48 | Revised introduction to make the overarching objectives much clearer. Paragraph 1 defines the problem and explains what the paper is about. It identifies deficiencies in current practice, citing references that have influenced this work. The scope of assessment (three pillars of performance) and value proposition are mentioned explicitly. |
| 3 | | 2-3 | 49-89 | Paragraph 2 provides a snapshot of recent work in probabilistic modelling and clarifies what the writers have in mind. |
| 4 | | 3 | 90-121 | Paragraph 3 emphasises the motivation and application context, explaining its unique characteristics and how forward extrapolation would shape the validation approach. |
| 5 | | 4-5 | 124-149 | Paragraph 4 recaps the main points and outlines the paper's organisation. |
| 6 | §2 | 5 | 156-175 | Rewritten modelling section to address issues identified by reviewer: citing seminal works and pointing out kriging/GP share the same conceptual foundations. |
| 7 | §2.3 | 8 | 246-260 | New section on conditional simulations. Point out that SGSim and Cholesky random field simulation are both examples of this; and the reasons for running such simulations. |
| 8 | §2.4 | 9-10 | 292-331 | New section on model configuration, conditional dependence and approximations. Point out implementation-specific features that contribute to differences in the kriging and GP models. Explain why these variations are reasonable for a methods-for-assessment paper. |
| 9 | §3.2 | 12 | 374-379 | Improve its flow. Provide reasons for having variogram-based spatial fidelity measure. |
| 10 | §3.3 | 13 | 404,410 | Renamed *likelihood* as *local consensus* to eliminate other connotations. |
| 11 | §5 | 19 | 539-545 | Restating the purpose of this study and focusing readers' attention on the key contributions. Addressing the critique that readers are not sufficiently informed about the organisation of the section and understanding the rationale and potential outcomes, lines 550-556 now highlights this clearly and provides a roadmap for what is to come. To avoid mental overload, non-essential figures have been removed (as indicated by the red text on p.20-21) |
| 12 | §6 | 36 | 841-902 | Rewritten the discussion section, linking the illustrated outcomes to study objectives. Pointing out the models considered currently depict variation in the number of simulation runs, but this metaphor can represent more broadly any modelling approach, configuration changes or variation in parameters that modellers may want to investigate, to assess the impact of various approximations/decisions on model performance. The discussion also mentions why correlated error clusters are not uncommon in mining, and some alternatives to the suggested measures. |
| 13 | §6.1 | 38 | 907-909 | The recommendations are given in the spirit of filling a gap in the absence of industry standards, to support highly automated model assessment performed at scale and across multiple sites, and to provide a richer understanding of model/data deficiencies in a potentially complex geological (grade modelling) environment. |
| 14 | §7 | 39 | 951 | Edited conclusion to indicate finding is implementation-dependent. |

Note: This file has been truncated as the authors have been asked by the copernicus system NOT
to submit the revised manuscript as a supplemental file attached with our comments.
The revised manuscript is available for upload if we are advised of the proper way for doing this.